# Structure and functioning of epipelagic mesozooplankton and response to dust deposition events during the spring PEACETIME cruise in the Mediterranean Sea

Guillermo Feliú[1], Marc Pagano[1], Pamela Hidalgo[2], François Carlotti[1]

[1] Aix Marseille Univ., Universite de Toulon, CNRS, IRD, MIO UM 110 , 13288, Marseille, France
[2] Department of Oceanography and Millennium Institute of Oceanography, Faculty of Natural Science and Oceanography, University of Concepcion, P.O. Box 160 C, Concepción, Chile

*Correspondence to*: François Carlotti (francois.carlotti@mio.osupytheas.fr)

**Abstract.** The PEACETIME cruise (May-June 2017) was a basin scale survey covering the Provencal, Algerian, Tyrrhenian and Ionian basins during the post-spring bloom period and was dedicated to track the impact of Saharan dust deposition events on the Mediterranean Sea pelagic ecosystem. Two such events occurred during this period, and the cruise strategy allowed to study the initial phase of the ecosystem response to one dust event in the Algerian basin (during 5 days at the so-called 'FAST long-duration station'), and a latter response to another dust event in the Tyrrhenian basin (by sampling from 5 to 12 days after the deposition). The present paper documents the structural and functioning patterns of the zooplankton component during this survey, including their responses to these two dust events. The mesozooplankon was sampled at 12 stations by combining nets with 2 mesh sizes (100 and 200 µm) mounted on a Bongo frame for vertical hauls within the 0-300 meter-depth layer.

Algerian and Tyrrhenian basins were found quite similar in terms of hydrological and biological variables, which clearly differentiated them from the northern Provencal Basin and the eastern Ionian Basin. In general, total mesozooplankton showed reduced variations in abundance and biomass values over the whole area, with a noticeable contribution of the small size fraction (< 500 µm) of up to 50 % in abundance and 25 % in biomass. This small-size fraction makes a significant contribution (15 to 21 %) to the mesozooplankton fluxes (carbon demand, grazing pressure, respiration and excretion) estimated using allometric relationships to the mesozooplankton size spectrum at all stations. The taxonomic structure was dominated by copepods, mainly cyclopoids and calanoids, and completed by appendicularians, ostracods and chaetognaths. Zooplankton taxa assemblages, analyzed using multivariate analysis and rank frequency diagrams, slightly differed between basins in agreement with recently proposed Mediterranean regional patterns.

However, the strongest changes in zooplankton community were linked to the dust deposition events. A synoptic analysis of the two dust events observed in the Tyrrhenian and Algerian basins and based on the rank frequency diagrams and a derived index proposed by Mouillot and Lepretre (2000) delivered a conceptual model of a virtual time series of zooplankton community responses after a dust deposition event. The initial phase before the deposition event (state 0) was dominated by small-size cells consumed by their typical zooplankton filter feeders (small copepods and appendicularians). Then, the

disturbed phase during the first five days after the deposition event (state 1) induced a strong increase of filter-feeders and grazers of larger cells and the progressive attraction of carnivorous species, leading to a sharp increase of the zooplankton distribution index. Afterward, this index progressively decreased from day 5 to day 12 highlighting a diversification of the community (state 2). A three weeks delay was estimated to get the index returned to its initial value, potentially indicating the recovery time of a Mediterranean zooplankton community after a dust event.

To our knowledge, PEACETIME is the first in situ study allowing observation of mesozooplankton responses before and soon after natural Saharan dust depositions. The change in rank-frequency diagrams of the zooplankton taxonomic structure is an interesting tool to highlight short-term responses of zooplankton to episodic dust deposition events. Obviously dust-stimulated pelagic productivity impacts up to mesozooplankton in terms of strong but short changes in taxa assemblage and trophic structure, with potential implications for oligotrophic system such as the Mediterranean Sea.

## 1 Introduction

The Mediterranean Sea is a semi-enclosed basin connected to the Atlantic Ocean and the Black Sea. It is composed of two major sub-basins, the Eastern and Western Mediterranean, connected by the Sicily strait (Skliris, 2014). The Mediterranean Sea can be considered as a model of the world's oceans (Bethoux et al., 1999; Lejeusne et al., 2010) because of its characteristics, such as the unique thermohaline circulation pattern and the deep water formation process. In addition, it is considered to be oligotrophic with an excess of carbon, a deficiency of phosphorus relative to nitrogen (MERMEX Group, 2011) and a decreasing west-east gradient in chlorophyll-*a* (i.e. Siokou-Frangou et al., 2010).

For the last two hundred years, numerous investigations have documented the pelagic zooplankton community inhabiting the Mediterranean Sea (Saiz et al., 2014), including long-term time series (i.e. Fernández de Puelles et al., 2003; Mazzocchi et al., 2007; Molinero et al., 2008; García-Comas et al., 2011; Berline et al., 2012) and a succession of oceanographic surveys covering wide transects at different time periods of the year (Kimor and Wood, 1975; Nowaczyk et al., 2011; Donoso et al., 2017; Siokou et al., 2019). The regular monitoring of the zooplankton community is essential when considering the high sensitivity of the Mediterranean Sea to anthropogenic and climate disturbance (Sazzini et al., 2014). Some of those disturbances may alter the structure and functioning of the pelagic ecosystem, and this is critical considering that marine ecosystems are being altered by anthropogenic climate change at an unprecedented rate (Chust et al., 2017).

Dust deposition is a major source of micro- and macro-nutrients (Wagener et al., 2010) that can stimulate primary production (Ridame et al., 2014), accelerate carbon sedimentation and possibly aggregation of marine particles (i.e. Neuer et al., 2004; Ternon et al., 2010; Bressac et al., 2014). Large amounts of Saharan dust can be transported in the atmosphere throughout the western and eastern Mediterranean Sea and then deposited on the sea surface by wet or dry deposition. The PEACETIME oceanographic cruise, carried out between May 10 and June 11 of 2017, was designed to study *in situ* the processes occurring in the Mediterranean Sea after atmospheric dust deposition and their impact on marine nutrient budget and fluxes, and on the biogeochemical functioning of the pelagic ecosystem. Thus, the survey strategy was designed to be

flexible in order to be able to change the sampling area depending on atmospheric events (Guieu et al., accepted).
Consequently, the survey sampling program realized consisted in 14 oceanographic stations in the central and western parts of the Mediterranean Sea.

The aims of the present contribution to the PEACETIME project are 1) to document the zooplankton abundance, biomass and size distribution along the survey transect, with special attention to small-sized zooplankton; 2) to analyze the relationship between zooplankton structure and environmental variability, including dust deposition; 3) to estimate the
bottom-up (nutrient regeneration) and the top-down (grazing) impact of zooplankton on phytoplankton stock and production by estimating its ingestion, respiration, ammonium and phosphate excretions using allometric models.

These objectives will serve to test the following hypotheses: whether the Saharan dust events impact the zooplankton community structure following deposition (H1), and if so, whether the effect would be immediately observable or after a lag time (H2). Finally, whether changes in zooplankton structure driven by dust deposition exceed regional differences under
oligothropic conditions (H3).

## 2 Material and methods

### 2.1 Study area and environmental variables

The PEACETIME cruise survey was conducted in May/June 2017 in the Western Mediterranean Sea (Figure 1) on board
R.V. *Pourquoi pas?*. Among the 12 stations studied, 10 were sampled once for zooplankton (the short-duration stations ST1 to ST9, and the long-duration station TYR), whereas two long-duration stations ION and FAST, lasting 3 and 5 days respectively, were sampled three times. The station positions along the transect were planned before the cruise in order to sample the principal ecoregions (see Figure 4 in Guieu et al., accepted), with the exception of FAST, an opportunistic station to monitor a wet dust deposition event which occurred on June 5 a few hours after the first sampling date (Table 1). A dust
event occurred over a large area including the southern Tyrrhenian Sea starting on May 10 which could have impacted the samples at ST5, TYR and ST6 which were sampled on May 16, 19 and 22, respectively (pers. comm. C. Guieu).

Hydrological variables (temperature, density, salinity) were measured on vertical profiles using a CTD. Dissolved oxygen was measured using a SBE43 sensor and chlorophyll-*a* concentration was determined from Niskin bottle samples by HPLC following the protocol of Ras et al. (2008), and with a Fluorescence sensor coupled with the CTD. Primary production was
measured with the [14]C-uptake technique, following the methods detailed in (Marañón et al., 2000). The depth of the mixed layer (MLD) was computed using the density difference criterion $\Delta\sigma_\theta = 0.03\ kgr^{-3}$ defined in de Boyer Montégut et al. (2004).

### 2.2 Ancillary data on dust deposition events occurring during the PEACETIME survey

Guieu et al. (introductory paper) detailed how they used three regional dust transport models to identify major dust events during the PEACETIME cruise. Two major wet dust events occurred during the period (Table 2). The first concerned the whole southern Tyrrhenian basin, with predicted flux > 1g m$^{-2}$ (Desboeufs et al. in prep.), and started on May 10, several days before the arrival of the vessel in this area. The dust event was confirmed by aluminium, iron and lithogenic Si measured in sediment traps at TYR 6 to 9 days after the event with a cumulated (4 days) lithogenic flux of 153 mg m$^{-2}$ at 200 m and 207 mg m$^{-2}$ at 1000 m (Bressac et al., in prep.). The second was located in the area between the Balearics and the Algerian coast and occurred from 3 to 5 June, with predicted flux of 0.5 g m$^{-2}$ (Guieu et al., accepted) after the arrival of the vessel in this area (station FAST). The dust event was confirmed by on-board atmospheric dust deposition samples (Desboeufs, in preparation this special issue), water column observations (nutrients, trace metals) (Tovar-Sánchez et al. 2020) and tracers of dust deposition in sediment traps with a cumulated (5 days) lithogenic flux of 50 mg m$^{-2}$ at 200 m and 70 mg m$^{-2}$ at 1000m (Bressac et al., in prep). Lithogenic flux values at TYR and FAST are likely underestimated considering that traps were placed with a time delay after the dust event (6 and 1 days respectively), thus the reported values could represent only a fraction of the total fluxes. The highest aerosol mass concentrations (around 25 μg m$^{-3}$) with the highest iron content (245 ng m$^{-3}$) were measured at FAST between 1 and 5 June, and subsequently the highest trace metal concentrations in the surface micro-layer were measured on 4 June (Co: 773.6 pM; Cu: 20.1 nM; Fe: 1433.3 nM; and Pb: 1294.7 pM) (Tovar-Sánchez et al 2020). The chemical composition of rain samples at FAST confirmed wet deposition of dust reaching a total particulate flux of 0.012 g m$^{-2}$ (Fu et al., in prep.). The Ionian basin was the only southern area not impacted by dust deposition during the PEACETIME cruise, and results obtained at the long-duration station ION will be considered (for comparison) as a non-recently impacted area.

## 2.3 Zooplankton sampling and sample processing

A total of 16 zooplankton samples were collected at 12 stations (Table 1) using a Bongo frame (double net ring of 60 cm mouth diameter) equipped with 100 μm and 200 μm mesh size nets (noted $N_{100}$ and $N_{200}$ below) mounted with filtering cod-ends. At all sampling stations, the Bongo frame was vertically towed from 300 m depth to the surface at a constant speed of 1ms$^{-1}$. Sample volume was estimated based on the ring diameter and the towed cable length. The sampling was mostly performed during the morning, except for ST7, ST9 and TYR, and night tows were also performed for the long-duration stations FAST and ION. The samples were preserved in 4% borax-buffered formalin immediately after the net was hauled back onto the deck.

The samples were processed using FlowCAM (Fluid Imaging Technologies Inc. Series VS-IV, Benchtop model) and ZOOSCAN (Gorsky et al., 2010). One of the goals of this study was to achieve determination of the complete size structure of the zooplankton community by combining different plankton mesh size nets and analysis techniques (FlowCAM and ZOOSCAN) in order to optimize the observed size spectrum. The formalin preserved samples were rinsed with tap water to remove the formalin. For net $N_{100}$, the sample was then split into 3 size fractions: < 200 μm (noted below $N_{100}F_{<200}$), 200 μm

– 1000 µm (noted below $N_{100}F_{200/1000}$), and > 1000 µm (noted below $N_{100}F_{>1000}$). For net $N_{200}$, the sample was split into two
size fractions: < 1000 µm (noted below $N_{200}F_{<1000}$) and > 1000 µm (noted below $N_{200}F_{>1000}$).

To determine the complete size spectrum, different combinations of size fractions from the two nets and analytical techniques were tested using two-way ANOVA. Taking into account the two mesh sizes, ($N_{100}$, $N_{200}$), the limits of the size spectrum were defined from the fraction $N_{100}F_{<200}$ for the lower limit and from the fraction $N_{200}F_{>1000}$ for the upper limit. Considering that our FlowCAM does not detect particles larger than 1200 µm of ESD and our ZOOSCAN does not detect particles smaller than 300 µm of ESD, $N_{100}F_{<200}$ was analyzed by FlowCAM and $N_{200}F_{>1000}$ by ZOOSCAN. The intermediate size fractions $N_{100}F_{200/1000}$ and $N_{200}F_{<1000}$ were both analyzed with ZOOSCAN and FlowCAM. These analyses delivered abundance and biomass values for successive ESD size classes: <200 µm (noted $C_{<200}$); 200-300 µm ($C_{200-300}$); 300-500 µm ($C_{300-500}$); 500-1000 µm ($C_{500-1000}$); 1000-2000 µm ($C_{1000-2000}$); > 2000 µm ($C_{200-300}$). The challenge was to choose the best net-analysis technique combination for the intermediate size fractions ($C_{200-300}$, $C_{300-500}$ and $C_{500-1000}$). The abundance of each class for the two nets and the two treatments was statistically compared. Parts of the spectrum corresponding to fractions $C_{200-300}$ and $C_{300-500}$ from $N_{100}$ measured with FlowCAM, and to the fractions $C_{500-1000}$ from $N_{200}$ measured with the ZOOSCAN have significantly higher abundances than other net-analysis technique combinations (P<0.000). Consequently, we combined data for $N_{100}F_{<200}$ and $N_{100}F_{200-1000}$ measured with FlowCAM to compute ESD size classes <500 um (Figure 2a) and data for $N_{200}F_{<1000}$ and $N_{200}F_{>1000}$ measured with ZOOSCAN to compute ESD size classes >500 um (see Figure 2b). The combination of these data enabled us to compute the final size spectrum (Figure 2c), that was used to estimate abundance, biomass and metabolic rates for each ESD size class, and then for the whole sample (sum of all the size classes) and for the total mesozooplankton (sum of the size classes $C_{200-300}$, $C_{300-500}$, $C_{500-1000}$ and $C_{1000-2000}$).

For the FlowCAM analyses, the sample was concentrated in a given water volume. Then, an aliquot of each sample was analyzed using FlowCAM in auto-image mode. For the fraction $N_{100}F_{<200}$, a 4X magnification and 300 µm FOV flow cell were used and the analysis was carried out up to 3000 counted particles. For the fraction $N_{100}F_{200-1000}$ a 2X magnification and 800 µm FOV flow cell were used and the analysis was carried out up to 1500 counted particles.

The digitalized images were analyzed using the VisualSpeadsheet® software and classified manually into taxonomic categories. Considered living organism groups for the FlowCAM were copepods, nauplii, crustaceans, appendicularians, gelatinous, chaetognaths and other diverse zooplankton groups (polychaeta, ostracods etc.). Non-organism particles were classified as detritus. Duplicates and bubbles were deleted.

To calculate the number of particles in the sample, the following equation was used.

$$A = \frac{p_a \times V_c}{V_a \times V_s}$$

where A is the abundance (ind m$^{-3}$); $P_a$ is the number of particles in the analyzed aliquot; $V_c$ is the given volume in the concentrated sample and $V_a$ is the volume of the analyzed aliquot and $V_s$ is the volume of sea water sampled by the zooplankton net (m$^3$).

For the ZOOSCAN analyses, the sample was homogenized and split using a Motoda box until a minimum of 1000 particles were obtained. Then, for the digitalization, the subsample was placed on the glass slide of the ZOOSCAN and the organisms were manually separated using a wooden spike to avoid overlapping. After scanning, the images were processed with ZooProcess (version 7.32) using the image analysis software Image J (Grosjean et al.,2004; Gorsky et al., 2010). Particles were classified automatically into taxonomic categories using the Plankton Identifier software (http://www.obs-vlfr.fr/~gaspari/Plankton_Identifier/index.php, last access: November 2019). Then the classification was manually verified to ensure that every vignette is in the correct category. Considered living groups of organisms for the ZOOSCAN were copepods, nauplii, crustaceans, appendicularians, gelatinous, chaetognaths and diverse zooplankton (polychaeta, ostracods etc.). Non-organism particles were classified as detritus. Blurs and bubbles were deleted.

## 2.4 Normalized biomass size spectrum

The size spectra were computed for each station using combined FlowCAM and ZOOSCAN data, following Suthers et al. (2006). Firstly, the data were classified in size categories of 0.1 mm of ESD from 0.2 to 2.0 mm. Zooplankton biovolume ($mm^3$) was estimated for each category following the equation:

$$Biovolume = \frac{1}{6} \times \pi \times (ESD)^3$$

with ESD expressed in mm. The X-axis of the normalized biomass size spectrum (NBSS) was calculated by dividing the biovolume by the abundance of each category and transformed into Log10. For the Y-axis, the biovolume of each category was divided by the difference in biovolume between two consecutive categories and transformed into Log10. NBSS slope and intercept were determined using linear regression model. The slope of the NBSS reflects the balance between small and large individuals, a steeper slope corresponding to a higher proportion of small individuals (bottom-up control) and a flatter slope corresponding to a higher proportion of large individuals (top down control) (Donoso et al., 2017; Naito et al., 2019).

## 2.5 Zooplankton carbon demand, respiration and excretion rates

The zooplankton carbon demand (ZCD in mg C $m^{-3}$ $d^{-1}$) was computed based on estimates of biomass from ZOOSCAN and FlowCAM samples and on estimates of growth rate:

$$ZCD = Ration \times B_{zoo}$$

where $B_{zoo}$ is the biomass of zooplankton in mgC $m^{-3}$, calculated using the area-weight relationships from Lehette and Hernández-León (2009) and converted to carbon assuming that carbon represent 40% of the total body dry weight (Omori and Ikeda, 1984). Ration ($d^{-1}$) is defined as the amount of food consumed per unit of biomass per day calculated as:

$$Ration = g_z + \frac{r}{A}$$

where $g_z$ is the growth rate, r is the weight specific respiration and A is assimilation efficiency. $g_z$ was calculated following Zhou et al. (2010):

$$g_z(w, \text{T}, \text{Ca}) = 0.033 \left( \frac{Ca}{Ca + 205e^{-0.125T}} \right) e^{0.09T} w^{-0.06}$$

as a function of sea water temperature (T, °C), food availability (Ca, mgC m$^{-3}$), estimated from Chl-a, and weight of individuals (w, mg C). We consider here that food is phytoplankton following Calbet et al. (1996). Following Alcaraz et al. (2007) and Nival et al. (1975), values of r and A were 0.16 d$^{-1}$ and 0.7 respectively. ZCD was compared to the phytoplankton stock, converted to carbon assuming a C:Chl/a ratio of 50:1, and to primary production to estimate the potential clearance of phytoplankton by zooplankton.

Ammonium and phosphorus excretion and oxygen consumption rates were estimated using the multiple regression model by Ikeda et al. (1985) with carbon body weight and temperature as independent variables

$$\ln Y = a_0 + a_1 \ln X_1 + a_2 X_2$$

Where lnY represent the ammonium excretion, phosphorus excretion or oxygen consumption. $a_0$, $a_1$ and $a_2$ are constant (see Ikeda et al. 1985), $X_1$ is the body mass (dry weight, carbon, nitrogen or phosphorus weight) and $X_2$ is the habitat temperature (°C).

Contribution to nutrient regeneration by zooplankton was estimated using the values of primary production and converted to nitrogen and phosphorus requirement using Redfield ratio. Respiration was converted to respiratory carbon lost assuming a respiratory quotient for zooplankton of 0.97 following Ikeda et al. (2000) and used as carbon requirement for zooplankton metabolism.

**2.6 Data analysis**

Spatial patterns of the environmental variables were explored using a principal component analysis (PCA). We considered temperature, salinity, dissolved oxygen and Chl-*a* values from a fluorescence sensor coupled with CTD, using mean values of the 0-300 m layer depth, plus the estimated MLD. The data were normalized prior to analyses performed using PRIMER v7 software (Anderson et al., 2008).

Differences in zooplankton abundance and biomass between size classes and areas were tested using two-way ANOVA. One-way ANOVA with Scheffé post-hoc analysis was applied to compare mean values between areas for total zooplankton and within each size class. Prior analyses data were log-transformed and tested for homogeneity. Dunnett's test was used in case of non-homogeneity. Potential association between univariate zooplankton and environmental data were tested using Spearman's rank-correlations. These analyses were performed with Statistica 7 Software. The 100 µm sample of station TYR was discarded from these data analyses due to poor state of preservation of the sample.

For studying the spatial patterns of zooplankton communities, a taxonomic group-station matrix with the abundance values was created and then square-root transformed to estimate station similarity using Bray Curtis similarity. The similarity matrix was then ordinated using Nonmetric Multidimenstional Scaling (NMDS). The contributions of significant taxa to the

similarity or dissimilarity between stations and areas was tested using SIMPER. Then the BIOENV algorithm was used to select the environmental variables best explaining the spatial pattern observed for the zooplankton communities. PERMANOVA was used to test the differences between areas based on environmental or zooplankton multivariate data. All these analyses were performed using PRIMER v7 software (Anderson et al., 2008).

The relationships between the biological and the environmental variables were also studied by coupling multivariate analyses of two datasets. The first dataset featured the abundances of all the zooplankton taxa identified from the 200μm net samples, and the second recorded environmental variables (the same as for the PCA analysis). A factorial correspondence analysis (FCA) and a principal component analysis (PCA) were performed on these two data sets, respectively. Then the results of the two analyses were associated through a co-inertia analysis (Doledec and Chessel, 1994) performed using ADE-4software (Thioulouse et al., 1997). Prior to the analyses, the data were log-transformed to tend towards the normality of the distributions.

Rank frequency diagrams (RFD) were created using the data from $N_{200}$ to see differences in taxonomic composition between the samples. In order to improve the interpretation of the RFDs, first we used a method derived from Saeedghalati et al. (2017) based on the ordination of normalized rank abundance distribution. Rank-abundance matrix was created with the data standardized by the total abundance. Resemblance was measured with Bray-Curtis similarity and a cluster was created using the complete linkage criterion. Secondly, a rank abundance distribution index was estimated following Mouillot and Lepretre (2000). The RFD for each station was separated into three portions: first the ranks with relative abundance <0.5 % were discarded (rare taxa, between 0 and 30% of the taxa according to all stations; by taking <1% we would discard between 18 and 49% of the taxa) and then the two parts were fitted with a linear regressions. One part with 4 highest ranks (see Mouillot and Lepretre for the justification) and the remaining portion with the following ranks (between 15 and 23 taxa, depending on the station). The slope for both upper and lower RFD portion was calculated (p1 and p2 respectively), then the p1/p2 ratios were estimated to quantify the differences between the RFDs of all the stations.

## 3 Results

### 3.1 Spatial patterns of environmental variables

The Principle Component Analysis (PCA) on environmental data explains 90.3 % of the total variance in the first two axes and delivers three clusters of oceanographic areas plus two distinct stations (Figure 3). The first axis (62 % of the variance) is mostly influenced by temperature and dissolved oxygen, as shown by their high correlations with the scores of the sampling points on this axis (r= 0.95 with p=0.000 and r=0.92 with p=0.000, respectively), whereas the second axis (28.3 %) is mostly influenced by MLD (r=-0.75, p=0.01), salinity (r=-0.75, p=0.001) and Chl-*a* (r=-0.57, p=0.022) (Supplementary Table 1).

The cluster of western stations in the Algerian Basin (AB) includes ST3, ST4, ST9 and FAST which are characterized by low temperature, salinity and MLD values. The cluster located in the Tyrrhenian Basin (TB) comprises (stations ST5, ST6 and TYR) and is very close to the first group, but with lower chlorophyll-a concentrations and higher values of temperature and salinity. Eastern stations (stations ST7, ST8 and ION ) located in the Ionian Basin (IB) are characterized by the highest temperature and salinity values and the lowest dissolved oxygen concentrations found during the survey. Stations 1 and 2 on the Provencal Basin (PB) do not cluster with any of the other stations due to deeper MLD and higher chlorophyll-a concentrations.

## 3.2 Spatial patterns of zooplankton structure

Zooplankton abundance (Figure 4a) during the PEACETIME cruise ranges between 265 and 583 x $10^3$ ind m$^{-2}$, with an average of 372 x $10^3$ ± 84 x $10^3$ ind m$^{-2}$, and biomass (Figure 4b) from 1160 to 2170 mgDW m$^{-2}$, with an average of 1707 ± 333 mgDW m$^{-2}$. The highest abundances are found in PB and AB, and the highest biomass in AB. The averaged total biomass in PB is lower than in AB, due to the very low contribution of the size classes $C_{1000-2000}$ and $C_{>2000}$, but size classes from $C_{<200}$ to $C_{500-1000}$ present higher biomass values than in AB. In TB, total biomass values decrease between ST4 and ST6, the latter presenting the lowest biomass value of the whole survey. Note that the biomass of TYR is obtained only for the size classes above 500 µm ESD, and the corresponding abundance is comparable to those obtained in ST5 and ST6 for these larger size classes. In IB, total biomass and abundance are lower than in AB and with low variability between stations. Detritus estimated for all analyzed classes by FlowCAM and ZOOSCAN represents between 14.6 to 39.1% of the total biomass. The $C_{200-300}$ ESD size class has the highest averaged contribution (42.9 %) to the total zooplankton abundance, followed by $C_{300-500}$ (28.5%), $C_{<200}$ (17.8 %), $C_{500-1000}$ (8,9 %), $C_{1000-2000}$ (1.7 %) and finally $C_{>2000}$ (0.22 %). In terms of biomass, $C_{500-1000}$ has the highest averaged contribution (25.3%), followed by $C_{1000-2000}$ (23.8 %), $C_{300-500}$ (21.3 %), $C_{>2000}$ (15.5 % ), $C_{200-300}$ (11,9 %), and finally $C_{<200}$ µm fraction (2.1 %). There is no correlation between total zooplankton abundance or biomass and integrated Chl-a, but $C_{300-500}$ biomass is negatively correlated with Chl-a (r=-0.52, p= 0.044). Total abundance is negatively correlated with temperature (r=-0.67, p= 0.006) (Table 3).

Copepods are the most abundant taxonomic group at all stations (Figure 5), representing 40 to 79 % of the abundance and 32 to 85 % of the total biomass. Abundance of zooplankton smaller than 300 µm is dominated by cyclopoid and calanoid copepodites. In $N_{200}$, 51 taxonomic groups are found of which 34 are copepod genus. The adult stages of the copepod community are dominated by the genus *Para/Clausocalanus* spp. (28.7 %), *Oithona* spp. (13.7 %), *Corycaeus* spp. (6.2 %), *Oncaea* spp. (4.1 %) and undefined calanoid copepods (7.0 %). The most abundant non-copepod groups are appendicularians (5.1 %), ostracods (4.8 %) and chaetognaths (3.6 %). The highest contributions of copepods to abundance and biomass are found in PB, and then this proportion tends to decrease southwards where the abundance and biomass of the other groups such as chaetognaths and gelatinous zooplankton increase. The ratio between copepods with length smaller than

1 mm and larger than 1mm (Figure 5) ranges from 2.8 to 8.3 (5.1 on average), with maximum mean values found in TB and minimum in IB.

The two-way ANOVA shows that the PB basin is characterized by significantly lower abundance and biomass in the upper size classes (1000-2000μm and >2000μm) compared to the other areas (p<0.05). One-way ANOVA results show that both total zooplankton and mesozooplankton present significantly higher abundance in PB than in IB, whereas their total biomass

was not significantly different between the areas (p>0.05). Significant differences in abundance and biomass between areas were found in the size classes C300-500, C1000-2000 and C>2000 and the biomass of C<200 (P<0.05) (Table 4 and Supplementary Figure 1).

NBSS is calculated for each station as shown in Figure 6 taking ION1 as an example. During the PEACETIME survey, the NBSS slopes (Figure 7) range between -0.60 and 1.27, with an average value of -0.80. The most negative slopes are found in

PB, whereas the IB area has the fewest negative slopes. At the long-duration stations FAST and ION, strong variations in slope values appear depending on the sampling time, with steeper slopes in the samples collected during the daytime indicating higher contributions of small zooplankton compared to large ones, and potentially linked to daily migration of larger forms deeper than 300 m.

The NMDS analysis (Figure 8) on the mesozooplanktonic taxa abundances based on $N_{200}$ delivers a distribution pattern for

the stations rather similar to that of the PCA on environmental variables. ST1 and ST2 on PB are the most dissimilar stations due to the higher abundance of copepods, especially *Para/Clausocalanus* spp. at ST1, which is twice as high as at ST2, and between 5 to 13 times higher than the rest of the transect (Figures 8a and 5). Similarly, *Centropages* spp. abundance is 10 times higher at ST1 and ST2 than at other stations of the survey. In contrast, abundances of *Oithona* spp. and *Corycaeus* spp., are respectively 6 and 10 times lower at ST1 and ST2 than at other stations. The zooplankton community in AB is

slightly different from those in TB and IB due to appendicularians and unidentified calanoid copepods being more abundant in AB and to *Haloptilus* spp. being more abundant in TB and IB. Within TB and IB, the three sampling dates (ION1, ION2, ION3) at station ION form a unique cluster, whereas, ST7 and 8 are grouped with station TB in another cluster. This differentiation of ST7 and 8 from the ION sampling dates in the NMDS analysis is mainly due to differences in relative abundance of *Mesocalanus* spp. (more abundant), ostracods (less abundant), *Clytemnestra* spp. (absent in ION) and

*Pontellidae* spp. (absent at ST7 and 8).

The SIMPER analysis shows that the lower average similarity between the stations is in PB (64.79 %) mainly due to *Para/Clausocalanus* spp. The rest of the basins share a higher internal similarity 78.43 %, 79.79 % and 78.03 % for AB, TB and IB respectively. Another interesting point highlighted in the SIMPER analysis is the lower average dissimilarity between TB and ST7 and ST8 from (20.25 %), this dissimilarity increases when the comparison is made between TB and the rest of

the stations included in IB (29.04 %); this is in agreement with the NMDS analysis (Figure 8) that related ST7 and ST8 with TB rather than with the stations in their basin.

**3.3 Relationship between environmental variables and zooplankton community**

Results of the PERMANOVA analysis on the environmental variables and on diversity on taxa are summarized on the following in Table 5. Interestingly, based on the zooplankton diversity of TB and IB, their difference is more significant when ST7 and ST8 are removed from IB and placed on TB (based on the NMDS cluster, Figure 8), whereas it is not the case when considering environmental variables (see Table below). This suggests that the similarity between st7 and st8 and the TB stations is not linked to the environmental context.

The BIOENV results show that salinity and chlorophyll were the environmental variables best explaining the overall spatial distribution of zooplankton community (BIOENV; Rs = 0.657).

The first factorial plane of the Co-inertia analysis (Figure 9) explained 96% of the total variance, with 79 % due to the first axis. On both spaces ('Environment' and 'Zooplankton'), the first axis opposes the IB stations associated with high temperature and salinity values and several zooplankton taxa (namely Echinoderm larvae and some copepod taxa, ie Pontellidae, *Rhincalanus* spp., *Haloptilus* spp. and *Phaena* spp.) to the PB and AB stations characterized by higher chlorophyll concentrations and by some copepod taxa (mainly *Pseudodiaptomus* spp., *Tortanus* spp. and *Pleuromama* spp.). On this axis, TB stations have an intermediate position, close to the coordinate zero. The second axis opposes northern (st1 and 2 of PB) and southern (AB) stations sampled in the Western Mediterranean basin. On this axis, PB stations are characterized by higher chlorophyll and salinity and deeper MLD, compared to AB and by the association with *Pseudodiaptomus* spp., whereas southern AB stations are associated with the copepods *Heterorhabdus* spp., *Labidocera* spp. and *Euterpina* spp. As in the preceding multivariate analyses, we note that St 8 and 9 from the IB tend to be closer to the TB stations than to the Ion station on the first factorial plane, particularly in the 'Zooplankton system'. The association between the environmental context and the zooplankton community is high with good correlation between the normalized scores of the stations (R2=0.844 and R2=0.820 for X1 and X2 axes, respectively), and by the positions of the plots of these stations close to the equality lines (i.e. X1 zooplankton = X1 Environment or X2 zooplankton = X2 Environment).

**3.4 Zooplankton community changes linked to dust deposition events during the PEACETIME survey**

The zooplankton community changes were analyzed using the variations of RFD between samplings. The RFDs for stations TYR, ST5, ST6, ION and FAST are presented separately in Figures 10a to 10d, and grouped in Figures 10e and 10f. As only one sample was done at station TYR, nine days after a large dust deposition event in the Southern Tyrrhenian Sea, RFDs of ST5 and ST6 also sampled in TB (six and twelve days after the dust event, respectively) are added for comparison (Figures 10a and 10b). At all three TB stations, RFDs are characterized by high dominance of filter-feeding zooplankton *Para/Clausocalanus* spp. and *Oithona* spp. in 1st and 2nd position with a strong drop in abundance for the following ranked taxa (undefined calanoid copepods or *Corycaeus* spp.). Appendicularians drop from the 4[th] position at ST5 and TYR to the 10[th] position at ST6. The shapes of RFDs change more between ST5 and TYR than between TYR and ST6. At station ION that was not impacted by dust deposition, RFD shapes are similar at both sampling dates (ION1 and ION3) with the

community dominated by *Para/Clausocalanus* spp. (Figure 10c). *Corycaeus* spp. changes from the 2$^{nd}$ position to the 4$^{th}$, calanoid copepods from 3$^{rd}$ to 6$^{th}$ and *Oithona* spp. from 4$^{th}$ to 2$^{nd}$. Appendicularians occupy a very similar position in both RFDs (6th and 7th rank at ION1 and ION3 respectively). At station FAST, the taxonomic composition is dominated by copepods (Figure 10d), but the rank order of the most dominant species changes between the two sampling dates (FAST1 and FAST3). *Oithona* spp. and *Para/Clausocalanus* spp. have the 1$^{rst}$ and 2$^{nd}$ ranks during FAST1, but this order is reversed at FAST 3. The 3$^{rd}$ place on both days are occupied by calanoid copepods. Appendicularians present one of the most significant changes, with their rank dropping from 4$^{th}$ to 14$^{th}$ between the two dates. It is remarkable that the RFDs change from a convex shape at FAST1 to a more concave one at FAST2, influenced by the high dominance of *Para/Clausocalanus* spp. at the first rank (Figure 10d). The comparison of the standardized RFDs for all the stations (Figure 10e) highlights that the greatest change in shape is visible at FAST, whereas it stays moderate at ION and negligible at TB. Figure 10f is similar to Figure 10e, but without ION, to visualize changes in zooplankton community composition at different time lags after a dust event, and will be commented on in more detail in the Discussion section. RFDs for all stations are shown in the Supplementary Figure 2.

### 3.4 Estimated zooplankton carbon demand, grazing pressure, respiration and excretion rates

Zooplankton carbon demand ZCD (Table 6) varies between 145.9 and 280.1 mgC m$^{-2}$ d$^{-1}$ at ST6 and FAST1, respectively. Assuming phytoplankton as the major food source, zooplankton consumption potentially represents 15 % of the phytoplankton stock on average per day and 97 % of the primary production (see Table 6). ZCD follows the zooplankton biomass pattern with higher values in AB and lower values in TB, and does not increase with primary production (r= -0.18, p>0.05). The average respiration (mean: 83.1 mgC m$^{-2}$ d$^{-1}$ and range between 62.9 and 112.2 mgC m$^{-2}$ d$^{-1}$) corresponds to 36.4 % of the integrated primary production. Almost half of this zooplankton respiration is due to organisms smaller than 500 μm of ESD. Mean ammonium excretion is 12.3 mg NH4 m$^{-2}$ d$^{-1}$ (range between 9.1 and 17.7 mg NH4 m$^{-2}$ d$^{-1}$), and mean phosphate excretion 1.7 mg PO4 m$^{-2}$ d$^{-1}$ (range between 1.3 to 2.3 PO4 m$^{-2}$ d$^{-1}$). The potential contributions of excreted nitrogen and phosphorus to primary production are respectively 31.5 % (range between 19.9 to 42.6 %) and 26.3% (range between 19.9 to 42.6%). Zooplankton size classes smaller than 500 μm of ESD contribute 45 % and 47 % of the total ammonium and phosphate excretion respectively. Estimated values for all zooplankton size classes of grazing, respiration and excretion rates and of their impact on the phytoplankton stock and production along the PEACETIME survey transect are presented in the Supplementary Table 2

## 4 Discussion

### 4.1 Methodological concerns and the importance of the small zooplankton fraction

This methodology combining two nets ($N_{100}$ and $N_{200}$) and two sample treatments (FlowCAM and ZOOSCAN) enables us to deliver a more accurate mesozooplankton community size spectrum (200-2000 μm), whereas size classes $C_{<200}$ and $C_{>2000}$ at the edges of the spectrum range remain under-sampled and require other equipment for proper sampling (respectively bottles and larger mesh size net). The length:width ratio of mesozooplankton organisms is quite variable, from 1 for the nearly round-shaped organisms such as nauplii or cladoceran, to more than 10 for long organisms such as chaetognaths (Pearre, 1982) or some copepods such as *Macrosetella gracilis* (Böttger-Schnack, 1989), with an average value between 3 and 4 for copepods (Mauchline, 1998). If we consider that organisms with a length:width ratio of 6 caught by the 200 μm mesh size will present an ESD of at least 490 μm, it is consistent that this net quite correctly samples organisms having an ESD above 500 μm ESD. For these organisms (> 500 μm ESD), ZOOSCAN is the most appropriate tool to deliver the size spectrum. Similarly, the 100 μm mesh size net allows small organisms of width just below 100 μm to pass through, but most of them might have an ESD up to 200 μm because for these smaller sizes, the length:width ratio is mostly below 4 (Mauchline, 1998). Due to the threshold of ZOOSCAN at 300 μm ESD, FlowCAM is the best tool to process organisms in the fraction below 500 μm.

Several authors have already highlighted the limitation of the 200 μm mesh size to catch small zooplankton individuals. Comparisons of different zooplankton mesh size nets comprised between 60 and 330 μm have systematically shown a decrease in abundance with increasing mesh size (Turner, 2004; Pasternak et al., 2008; Riccardi, 2010; Makabe et al., 2012; Altukhov et al., 2015). When the goal of the study is to achieve a full understanding of the complete mesozooplankton community structure and functioning, the size selectivity of the sampling nets is an important issue: clearly, a large fraction of organisms of ESD between 200 and 500 μm is undersampled using a single 200 μm mesh size net. Pasternak et al. (2008) reported that a 220 μm mesh can lose up to 98% of the abundance of *Oithona* spp. and 80% of copepodite stages of *Calanus* spp. Riccardi (2010) found that a classical 200 μm net catches only 11% of the abundance and 54 % of the biomass compared to a 80 μm mesh size, leading also to differences in observed species composition in the Venice lagoon. During the PEACETIME survey, the small size classes ($C_{200-300}$ and $C_{300-500}$) of mesozooplankton have been optimally sampled using a 100 μm mesh size net ($N_{100}$). Consequently, these size classes represent very large percentages of the total abundance (respectively 52.3 and 34.8 %) and a significant contribution to the total biomass (respectively 14.5 and 25.9 %). These reliable estimations have direct consequences for the estimated fluxes (see below).

### 4.2 Differences in abundance, biomass and zooplankton community structure in relation to regional environmental characteristics

A review of the most relevant information available on zooplankton biomass and abundance in different regions of the Central and Western Mediterranean Sea (Table 7) shows a wide range of variation that can be attributed to location,

sampling seasons and/or sampling methods (net mesh size, depth of tow, etc), and in general, the values during the
PEACETIME survey are of the same order of magnitude, although most of other studies were performed with a 200 μm
mesh size net and often over a shallower surface layer. However, during this post-bloom period, no clear regional patterns in
abundance and biomass were found, unlike other descriptions showing a north-south and west-east decrease in zooplankton
stocks (Dolan et al., 2002, Siokou-Frangou, 2004). In PB, Donoso et al. (2017) and Nival et al. (1975) highlighted a strong
variability which is consistent with the strong gradient found between ST1 and ST2 during PEACETIME (see Figure 4). In
AB, abundance and biomass values obtained during the survey are similar to those recorded in late spring by Nowaczyk et
al. (2011), whereas Riandey et al. (2005) found lower abundance and higher biomass values. However, the latter study
focused on high resolution of a mesoscale eddy highlighting an important fine-scale variability of abundance and biomass
values. For TB, the data are difficult to compare due to different sampling conditions (net mesh size, depth of tow and
sampling season). In IB, all biomass values presented in Table 7 are of the same order, but abundances found by Mazzocchi
et al. (2003, 2014) are three times lower than those observed during PEACETIME, probably due to a high contribution of
$C_{<200}$ and $C_{200-300}$ obtained with $N_{100}$ (see Figure 4). In general, the better sampling of small size classes with $N_{100}$ should lead
to higher abundance values. However, the comparison of data in Table 7 shows that regional and temporal variability of
these values partially masks this benefits.

In PEACETIME, clear regional differences are found both in terms of environmental variables and zooplankton taxonomic
composition. ST1 and ST2 are clearly differentiated from all the others with deeper MLD, higher chlorophyll-*a*
concentrations and a zooplankton community dominated by typical herbivorous copepods of PB (*Centropages*,
*Para/Clausocalanus*, *Acartia,* etc), as mentioned by Gaudy et al. (2003) and Donoso et al. (2017), and characterized by a
scarcity of thaliaceans which normally occurs in ephemeral and aperiodical patches (Deibel and Paffenhöfer, 2009). AB and
TB are very closely related to each other in terms of hydrological features and chlorophyll-*a*, but slightly differentiated in
salinity and zooplankton taxonomy, probably because they are both strongly influenced by the Modified Atlantic Water
(MAW) and its associated  mesoscale features (Millot and Taupier-Letage, 2005). In AB, 17 days separated the sampling of
ST3 and ST4 with that of ST9 and FAST, but despite this time gap, they are very close in terms of hydrological features,
chlorophyll-*a* level and zooplankton community structure. IB is clearly differentiated from these groups in terms of
environmental parameters (see Figure 3) due to higher salinity and lower chlorophyll-*a*, but in terms of zooplankton
community the western Ionian stations (ST7 and ST8) present more analogy with TB than with the ION station (see Figure
8). During PEACETIME, the station ION appears clearly separated from ST7 and ST8 located further westwards by a north-
south jet (ADCP and MVP observations, Berline et al., in preparation), which might correspond to the Mid-Mediterranean
Jet (Malanotte-Rizzoli et al., 2014, their Figure 5). The location of ST7 and ST8 within anticyclonic structures of the portion
of the Modified Atlantic Water (MAW) flowing through the Sicily Channel could explain their similarity to TB stations in
terms of zooplankton assemblages, as TB is directly influenced by the main part of the MAW flowing through the Sardinia
Channel. Ayata et al. (2018) also classified the Tyrrhenian Sea as heterogeneous due to complex circulation patterns
including transient hydrodynamic structures in the south, which could also explain the similarity of ST7 and ST8 to TB

stations in terms of zooplankton assemblages during PEACETIME. This visited area of the IB during PEACETIME certainly represents a transition area between the eastern and western Mediterranean basins (Siokou-Frangou et al., 2010; Mazzocchi et al., 2003).

These regional differences highlighted both in terms of environmental characteristics and zooplankton taxa assemblages are in agreement with the regionalization of the Mediterranean basin by Ayata et al. (2018) based on historical biogeochemical, biological and physical data of the epipelagic zone. For example, ST1 of PEACETIME characterized by high Chl-*a*, high zooplankton abundance and dominance of small copepods is clearly located in the 'consensual Ligurian Sea Region' *sensu* Ayata et al. (2018), identified as the most productive of the Mediterranean due to intense deep convection events. Among AB stations, stations 3, 4 and 9 are clearly in the 'consensual Algerian region' (Ayata et al., 2018), whereas station FAST corresponds to the 'western Algerian heterogeneous region'. Among the IB stations, the separation of stations 7 and 8 from the ION stations in terms of zooplankton communities and, to a lesser extent, of environmental variables, also corresponds to the distinction between the 'consensual North Ionian' region and the western part of the 'Ionian Sea region', considered as a heterogeneous region (Ayata et al., 2018).

## 4.3 Estimated zooplankton-mediated fluxes during the PEACETIME survey

By using allometric relationships relating zooplankton grazing and metabolic rates to size structure, zooplankton impacts (top-down vs. bottom-up) on primary production have been investigated. We are aware that using constant conversion factors may limit the analysis of the spatial variation, since these factors may display temporal and geographical variations (Minutoli and Guglielmo, 2009). However, our sampling strategy based on a limited number of stations sampled did not enable us to consider temporal and spatial variations accurately, and our main goal was to have rough estimations of the epipelagic zooplankton mediated fluxes at the scale of the PEACETIME cruise.

ZCD estimations show that zooplankton required 15 % of the daily phytoplankton stock, with narrow variations over the whole area (between 9.5 to 19.3), which are twice lower than the values estimated by Donoso et al. (2017) during the spring bloom in the North-Western Mediterranean Sea. However, estimated grazing rates are of the order of the estimated primary production, which corresponds to the highest range of the values summarized by Siokou-Frangou et al. (2010) for the whole Mediterranean Sea (from 14 to 100 %). Just estimating ZCD on the basis of mesozooplankton alone certainly leads to overestimation of its top-down impact on phytoplankton. In the Mediterranean Sea, the primary production is consumed by a "multivorous web" including microbial and zooplankton components (Siokou-Frangou et al., 2010). Mesozooplankton simultaneously grazes on phytoplankton and heterotrophic prey, such as heterotrophic dinoflagellates (Sherr and Sherr, 2007) or ciliates (Dolan et al., 2002), and might be quite flexible in its feeding strategy depending on the composition and size of prey as well as on environmental variables such as turbulence (Kleppel, 1993; Yang en al., 2010). On one hand, a large part of the primary production can be consumed by ciliates (Dolan and Marrasé et al., 1995), but on the other hand mesozooplankton can consume almost the entire ciliate production (Pitta et al., 2001; Pérez et al., 1997; Zervoudaki et al., 2007), potentially explaining the wide variations of standing stock of ciliates over the Mediterranean Sea (Dolan et al., 1999;

Pitta et al., 2001; Dolan et al., 2002). The extensively described east-west pattern of decreasing grazing impact (Siokou-Frangou et al., 2010) could not be observed during this study as only one station (ION station) was typical of the Eastern Mediterranean Sea.

Estimated $NH_3$ and $PO_4$ excretion rates by mesozooplankton during PEACETIME are consistent with the few observations collected in the Mediterranean Sea (Alcaraz, 1988; Alcaraz et al., 1994; Gaudy et al., 2003) and with those obtained at similar latitudes (see review in Hernández-León at al., 2008). From our estimation, zooplankton excretion would contribute respectively to 21 - 44 % and 17 - 38 % of the N and P requirements for phytoplankton production. In the NWMS, Alcaraz et al. (1994) estimated a zooplankton nitrogen excretion contribution to primary production > 40%, whereas Gaudy et al. (2003) reported 31-32 % and 10-100 % N and P contributions. This impact on phytoplankton production can be even greater in proximity to the DCM where zooplankton tends to aggregate fuelling regenerated production (Saiz and Alcaraz, 1990) and enhancing bacterial production (Christaki et al., 1998). Zooplankton grazing impact and nutrient contribution to primary production are higher in the western basin than in the Ionian Sea, mainly linked to variations of zooplankton biomass.

Mean carbon released through zooplankton respiration represents 36 % of the primary production during PEACETIME, which is higher than previous measurements in NWMS (by Alcaraz, 1988 and Gaudy et al., 2003) from onboard incubation experiments on zooplankton collected with a 200 μm mesh size net.

Metabolic estimations clearly show that the size fractions < 500 μm (optimally captured with the 100 μm mesh size net) make a significant contribution to the whole mesozooplankton estimated fluxes: 14.9 % of the ZCD is due to organisms <300 μm, and this size class contributes 21 % and 20 % of the total ammonium and phosphate excretion, respectively.

### 4.4 Impact of dust deposition on the zooplankton community

In the past years, responses to Saharan dust inputs in marine systems have been mostly studied in microcosm and mesocosm experiments, but more rarely observed *in situ*. Most studied responses to dust are focused on the microbial biota and are generally marked by an increase in metabolic rates rather than by standing stock changes (probably due to trophic transfer along the food-web) (Ternon et al., 2011; Guieu et al., 2014; Ridame et al., 2014; Herut et al., 2016). In mesocosms, changes in zooplankton stocks are strongly dependent on the initial conditions, and cannot really reflect what could occur in natural waters within the Mediterranean "multivorous planktonic food-web" (Siokou-Frangou et al., 2010). Pitta et al. (2017) found an increase in mesozooplankton biomass 9 days after the beginning of a mesocosm experiment, probably as a result of an earlier increase of prey (flagellates, ciliates and dinoflagellates). Tsagaraki et al. (2017) described an increase in productivity after an artificial dust deposition that was transferred to higher trophic levels by the classical food web, resulting in an increase of copepod egg production 5 days after the beginning of the experiment. Very few *in situ* studies have documented mesozooplankton responses to Saharan dust. Abundance increase was observed by Thingstad et al. (2005) in the Eastern Mediterranean Sea, and by Hernández-León et al. (2004) in Atlantic waters close to the Canary Islands one week after the deposition. In this latter area, Franchy et al. (2013) detected increases of zooplankton grazing and zooplankton biomass after

another event. Thus, the PEACETIME survey dedicated to the tracking of such events was an opportunity to observe real *in situ* zooplankton responses in the epipelagic layer (0-300 m).

At station FAST (an opportunistic station after a Saharan dust deposition event), an increase in nitrate (from 50 nM to 120 nM) and phosphate concentrations (from 8 nM to 16 nM) occurred in the mixed layer (pers. comm. C. Guieu), which led to an increase in primary production from FAST1 to FAST3, but with no visible changes in phytoplankton biomass (see Table 2). For zooplankton, the total abundance slightly decreases but the community composition presents obvious changes, mainly a decrease of appendicularians and an increase of *Para/Clausocalanus* spp. and of carnivorous taxa (*Candacia* spp.,

chaetognaths, siphonophores) (see Figure 10d). The sharp decrease of appendicularian abundance (four-fold decrease) and rank position (see Figure 10d) could potentially be linked either to food limitation or to predation. Size and species composition of the phytoplankton community in FAST suggest a change toward larger cells (Table 2) poorly edible by appendicularians and inducing filter clogging. There were also potential increases in food competition with *Para/Clausocalanus* spp. (Lombard et al., 2010) and/or in predation by chaetognaths and siphonophores (Purcell et al.,

2005). Although total zooplankton biomass remains relatively stable at FAST, the contribution of the size classes $C_{500-1000}$ and $C_{1000-2000}$ increase relative to the smaller size classes (see Figure 4b) inducing variations on the NBSS slope from -0.76 to -0.63 (see Figure 6). This 15% increase in biomass is mainly due to large migrating taxa such as copepods *Eucalanus* spp., *Rhincalanus* spp. and *Candacia* spp., chaetognaths and siphonophores. The daily observation of sediment traps at 200 and 500 meters over five days between FAST1 and FAST3 (pers. comm. C. Guieu) shows a relative increase of swimmers

collected at 500 m versus those collected at 200 m, also suggesting increasing numbers of migrants. An obvious planktonic transition occurred during this period but it is difficult to conclude which of the bottom-up (changes in primary producers) or top-down (increase of carnivorous migrants) effects was dominant. The change in the RFDs (Figure 10d), from a convex shape at FAST1, indicating a more stable system with no dominance of the first taxonomic groups, to a more concave shape at FAST3 influenced by the high dominance of *Para/Clausocalanus* at the first rank, could reflect a disturbance effect (*sensu*

Pinca and Dallot, 1997) of the dust deposition on the zooplankton community.

A synoptic analysis of the RFDs linked to the dust events observed in the Tyrrhenian basin and at station FAST offers a basis for proposing a conceptual model of a virtual time series of zooplankton community responses after a dust deposition event (Figure 10f): the first sampling is carried out before the event (FAST1), and several other samplings are done with a time-lag of five days (FAST3), six days (ST5), nine days (TYR) and twelve days (ST6) after the event. FAST1 represents an initial

steady state (state 0) with no dominance in the first taxa ranks, while FAST3 and ST5 represent a disturbed state of the community (state 1) with strong dominance of the first taxa and the collapse of the following ones. TYR and ST6 represent the beginning of recovery towards a stable system (state 2) with the move up of the second rank. State 0 before the dust event is characterized by oligothropic conditions with low nutrients, low phytoplankton concentration dominated by small-size cells and their typical zooplankton grazers (e.g. appendicularians and thaliaceans), leading to a convex RFD shape (like

FAST1, Figure 10f) reflecting a mature community (*sensu* Frontier, 1976). State 1 is characterized by a nutrient input linked to the dust event stimulating larger phytoplankton cells and their herbivorous grazers (copepods) and attracting carnivorous

migrants leading to a more concave RFD shape (like FAST3, ST5 and TYR, Figure 10f) typical of a disturbed community (*sensu* Frontier, 1976). State 2 is characterized by the diversification of herbivorous taxa leading to changes in RFD towards a convex shape (like ST6, Figure 10f).

The cluster analysis on the RFDs (Figure 11a) is in agreement with this succession of the time series (Figure 10f) by grouping the stations according to impact level of the wet dust deposition. It separates the initial condition (FAST1) from the most disturbed state (stations FAST3 and ST6) and identifies a transition phase before (FAST2) and after (TYR and ST6) the peak disturbance. The changing trends in p1/p2 ratios (Figure 11b) show an interesting development, with a sharp increase until day 5 after the dust deposition and a progressive decrease towards the end of the virtual time series. The linear

regression suggests that the community structure will deliver a p1/p2 ratio value similar to the initial value of the time series after 22 days. Is interesting to note that this delay corresponds to an average generation time of zooplankton organisms for this region. Cluster analysis on the RFDs and p1/p2 ratio for all stations are shown in the Supplementary Figures 3 and 4 respectively. Interestingly, in the Co-inertia analysis (see Figure 9), the stations impacted by the dust (FAST and TB stations) are grouped on the left side of the relationship between X2 axis of environment and zooplankton. In addition, their

succession in this graph is consistent with the sequence observed in the virtual time series of RFD (with FAST1 as the initial station before the dust deposition and TYR and St6 corresponding to day 9 and 12 after the dust event) showing the coupled impact of dust on both environment and zooplankton.

## 5 Conclusion

To our knowledge, PEACETIME was the first study in the Mediterranean Sea that managed to collect zooplankton samples before and soon after natural Saharan dust deposition events and to highlight *in situ* zooplankton responses in terms of community composition and size structure. Our study suggests that a complete understanding of the mesozooplankton community response to a single massive dust event would require continuous observation over two to three weeks, from an initial state just before the event to a complete process of zooplankton community succession after the event. To identify

such a succession, the rank-frequency diagrams of the zooplankton taxonomic structure appear to be a more practical and sensitive index than observable changes in stock (abundance and biomass) or in metabolic rates, and should be further tested. Particularly the changes of the p1/p2 ratio might characterize the response of the zooplankton community to a pulse of dust (or any massive disturbance) and its resilience capacity after the forcing event.

This approach requires a complete overview of mesozooplankton size spectrum and community composition which was

achieved in our study by combining data from two mesh size nets (100 and 200 μm) and two analytical techniques (FlowCAM and ZOOSCAN). In our study, this strategy also enabled us to show the importance of small forms (< 500 μm of ESD) both in terms of stocks and fluxes.

## Acknowledgments

This study is a contribution to the PEACETIME project (http://peacetime-project.org), a joint initiative of the MERMEX and ChArMEx components supported by CNRS-INSU, IFREMER, CEA, and Météo-France as part of the programme MISTRALS coordinated by INSU. The PEACETIME cruise (https://doi.org/10.17600/17000300) was managed by C. Guieu (LOV) and Karine Desboeufs (LISA). We thank the PEACETIME project coordinators and scientists on board, especially Nagib Bhairy who did the zooplankton sampling. Zooplankton analyses were realized on the Microscopy and Imaging platform of MIO, partly funded from European FEDER Fund under project 1166-39417.

Thanks to E. Maranon and M. Perez-Lorenzo for the PP data and to Julia Uitz, Céline Dimier and the SAPIGH analytical service at the Institut de la Mer de Villefranche (IMEV) for onboard sampling and HPLC analysis. Thanks to Cécile Guieu, Elvira Pullido, France Van Wambeke, and Julia Uitz for critical reading and advice on the draft, and to Michael Paul for correcting the English. We would like to thank the two reviewers for their constructive comments and suggestions which stimulated a substantial revision.

G. Feliú was supported by a Becas-Chile PhD scholarship by the National Agency for Research and Development (ANID), Government of Chile.

## Data availability

All data and metadata will be made available at the French INSU/CNRS LEFE CYBER database (scientific coordinator: Hervé Claustre; data manager, webmaster: Catherine Schmechtig). INSU/CNRSLEFE CYBER (2020)

## Authors contribution

GF, MP and FC wrote the paper with contributions by PH. GF participated in the sample treatment. GF, FC, MP and PH participated in the data analysis

## Competing interests

The authors declare that they have no conflict of interest.

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

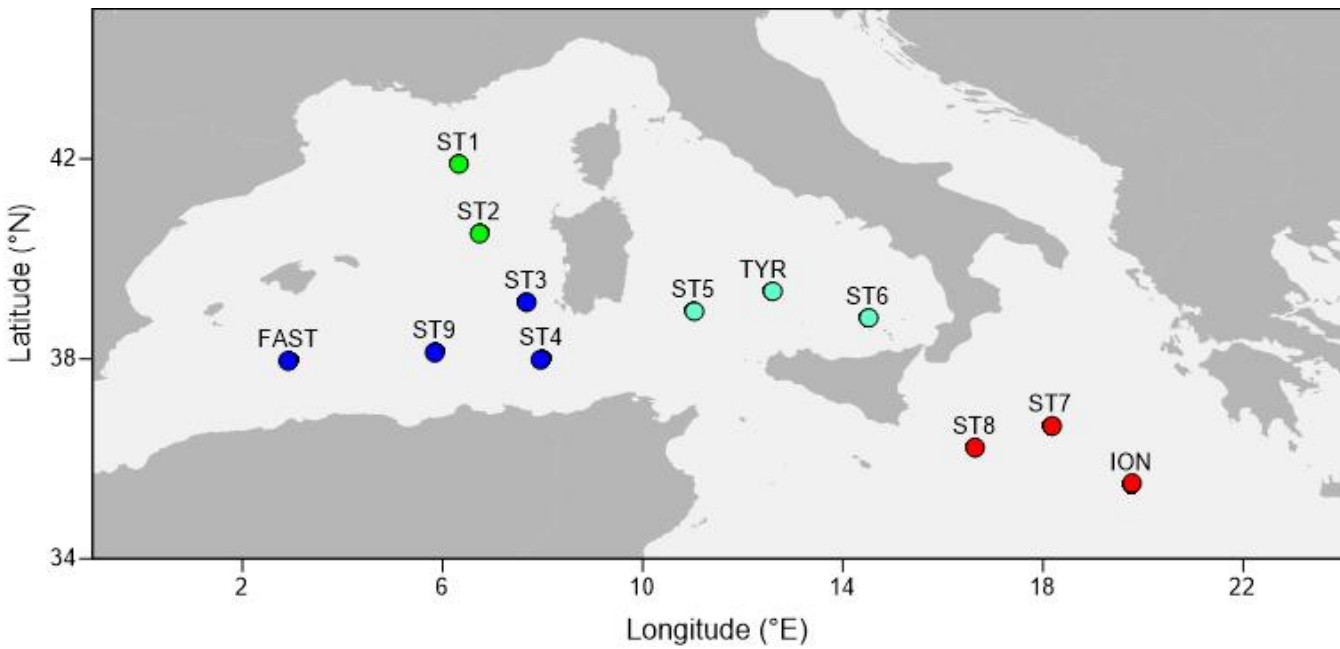


**Figure 1: Map with the sampling points during PEACETIME cruise 2017. The colours of the points indicate the different areas considered in the course of the study. Green dots: Provencal Basin (PB); Dark blue dots: Algerian Basin (AB); Light blue dots: Tyrrhenian Sea (TB); Red dots: Ionian Basin (IB).**

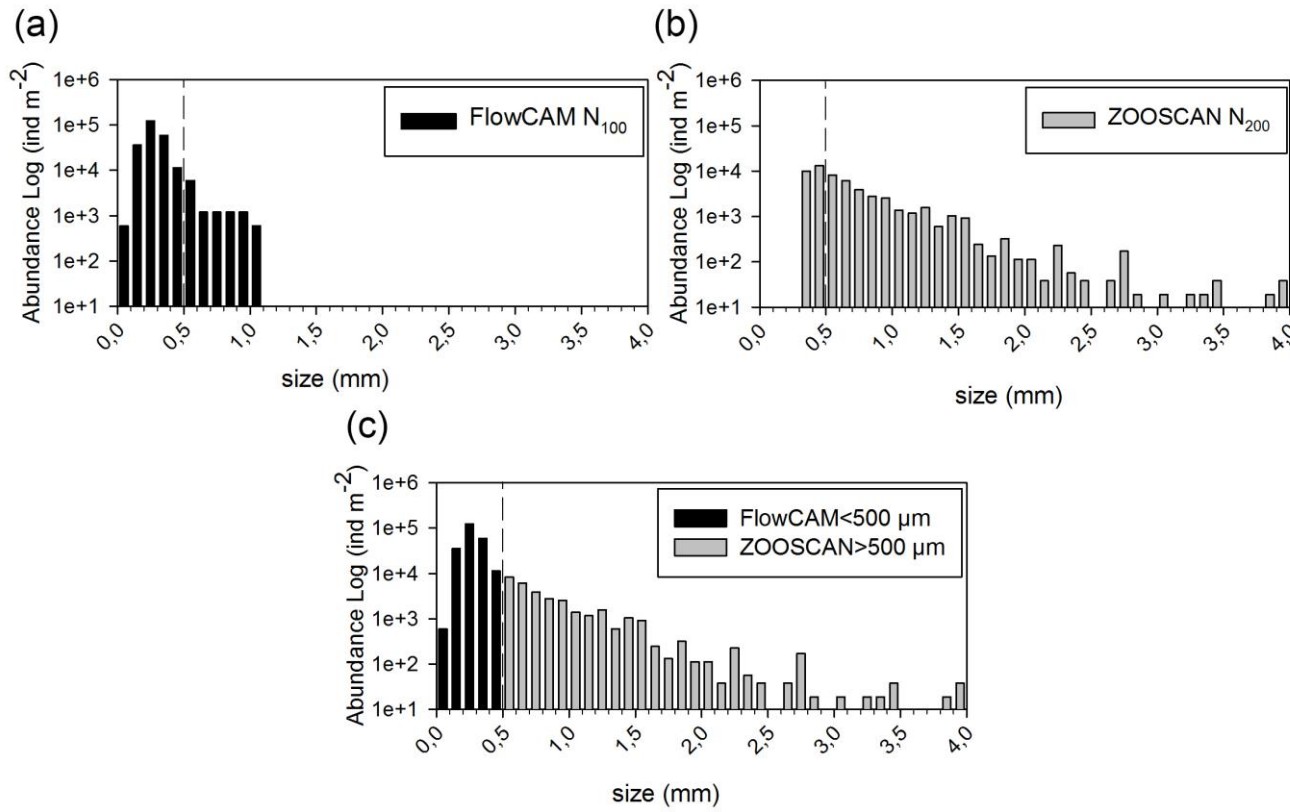


Figure 2: Size spectrum of ST ION1 as an example obtained by: a) FlowCAM ($N_{100}$), b) ZOOSCAN ($N_{200}$) and b) combination of FlowCAM ($N_{100}$ counting only zooplankton smaller than 500 µm of ESD) and ZOOSCAN ($N_{200}$ counting only zooplankton bigger than 500 µm of ESD)

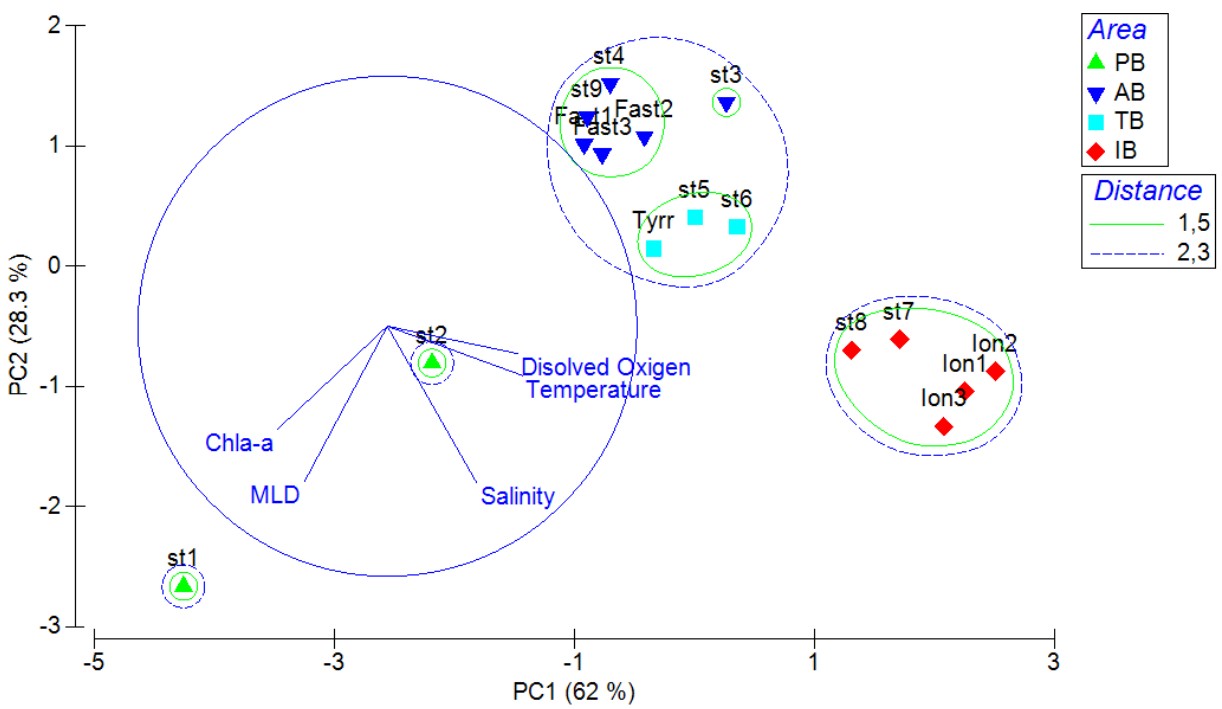


**Figure 3: Principal Component Analysis (PCA) ordination of five environmental indicators: Mixed/layer Depth (MLD), Integrated values of Chl-a concentration, mean values on the upper 0/300 m of temperature, salinity and dissolved oxygen. AB: Algerian Basin, PB: Provencal Basin, TB: Tyrrhenian Basin, IB: Ionian Basin.**


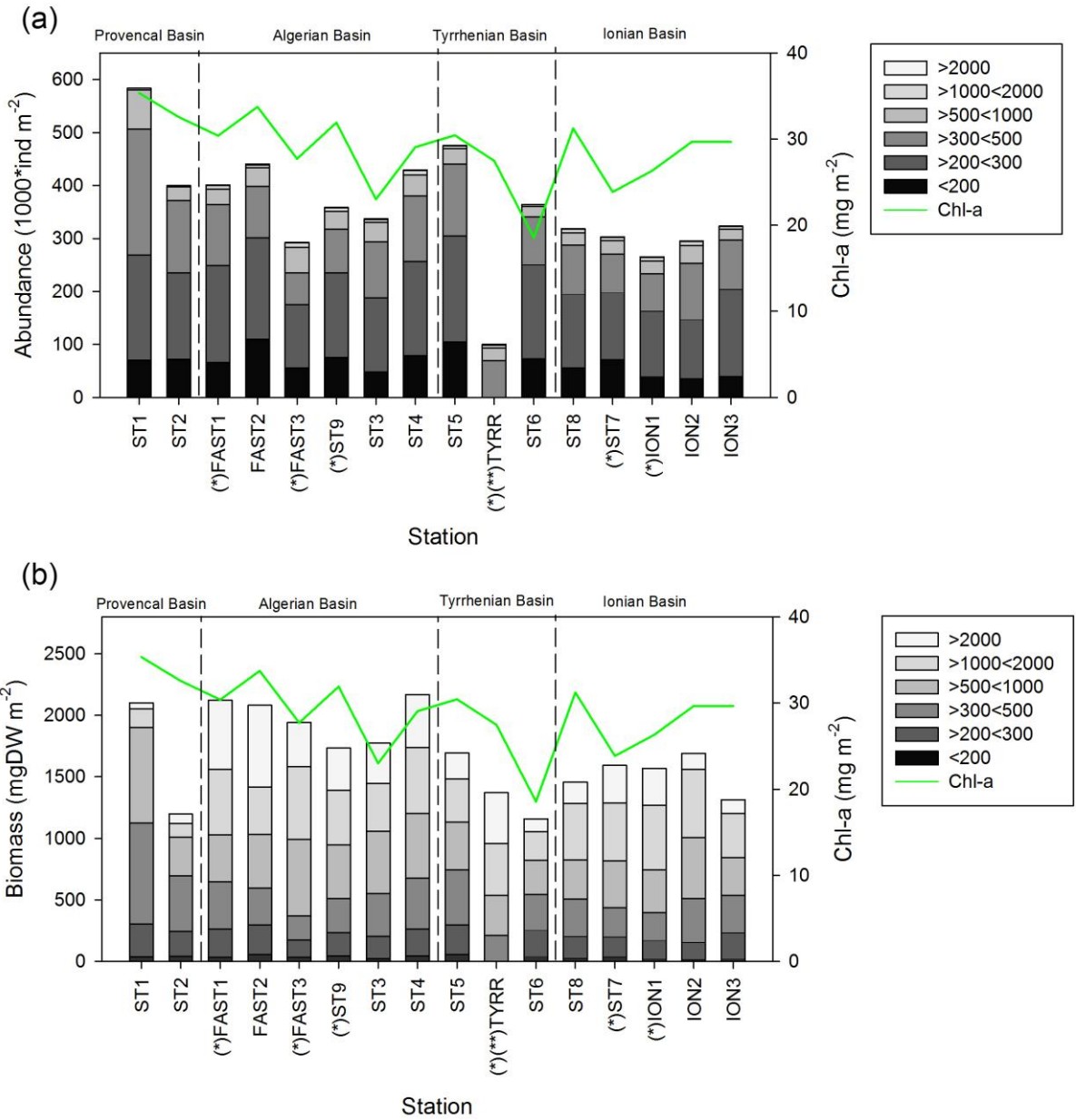

**Figure 4: Values of zooplankton abundance (a) and biomass (b) cumulated by ESD size classes across different stations of the PEACETIME cruise. Integrated Chl-a concentrations (green line). (*)Stations sampled during the night. (**) At station TYR, only the abundance and biomass values above 300 μm are presented.**


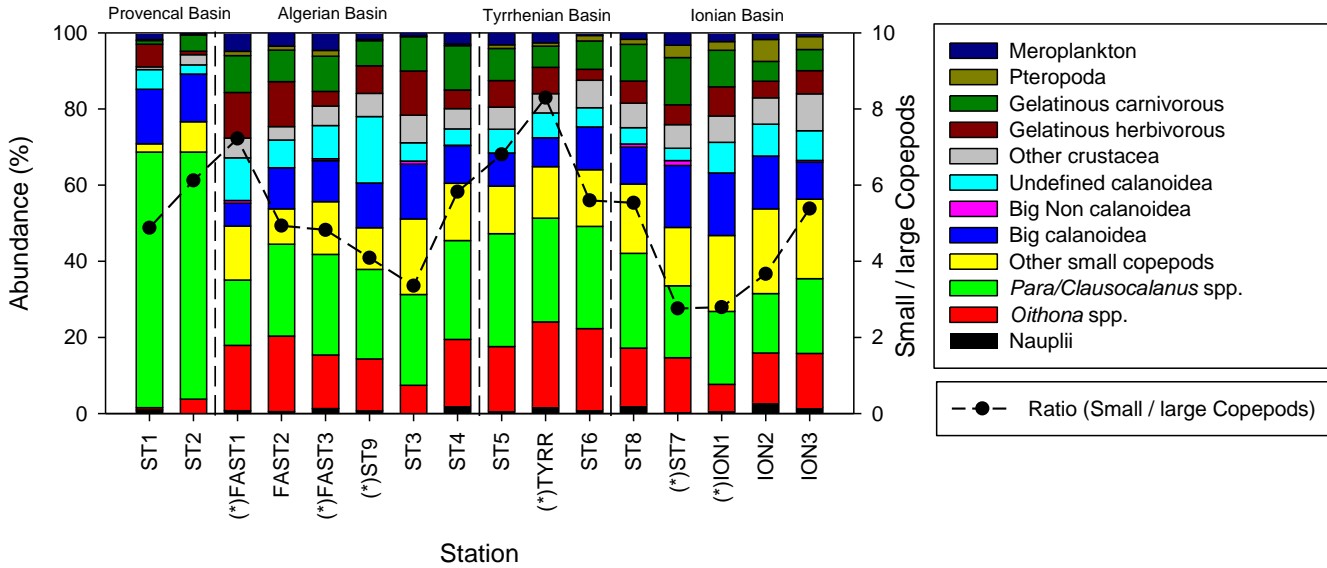

**Figure 5: spatial variation of taxonomic groups (stock bars) and small (length< 1 mm)/large (length> 1 mm) copepod ratio (dashed line). (*)Stations sampled during the night.**

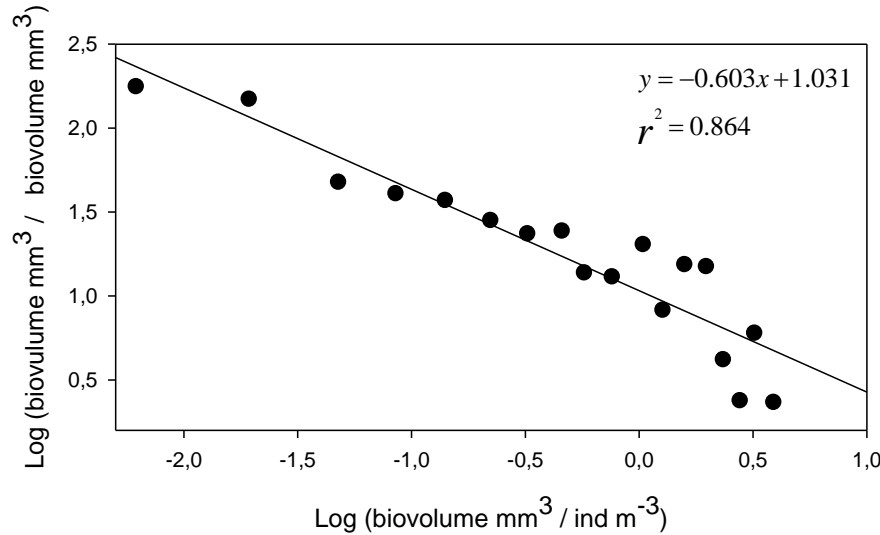


**Figure 6: Normalized biomass size spectrum (NBSS) of mesozooplankton at Station ION1. Normalized biomasses in the successive size classes (black dots) and lineal regression (straight line) giving the slope value.**

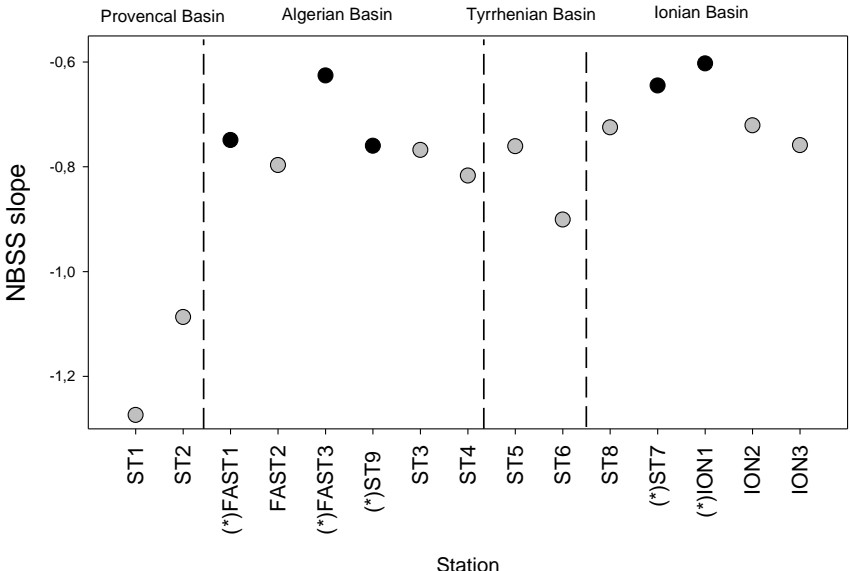

**Figure 7: NBSS slope values of mesozooplankton obtained for all stations during the PEACETIME survey. Black dots (night samples) and grey dots (day samples)**

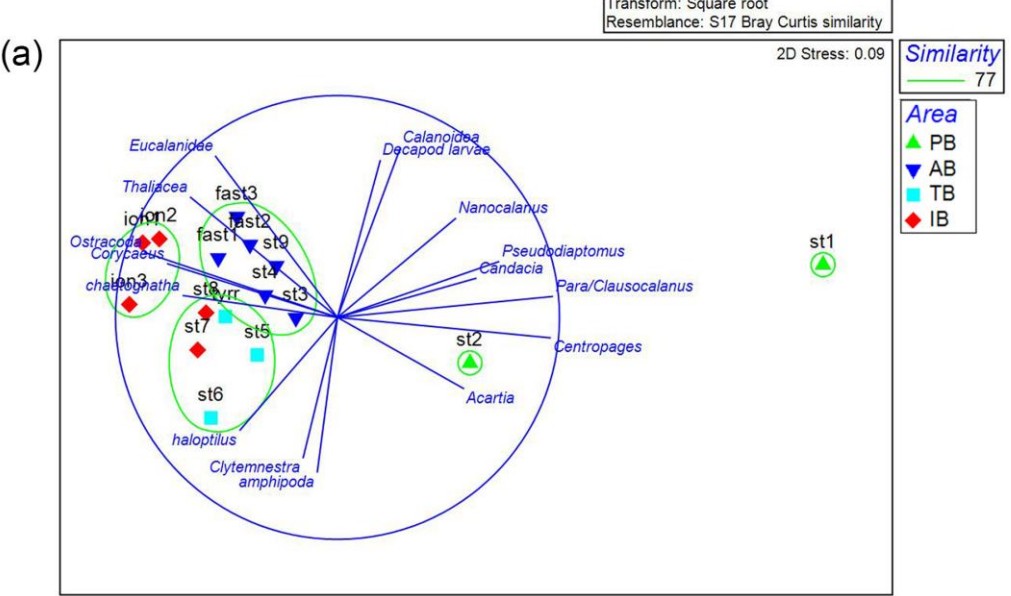

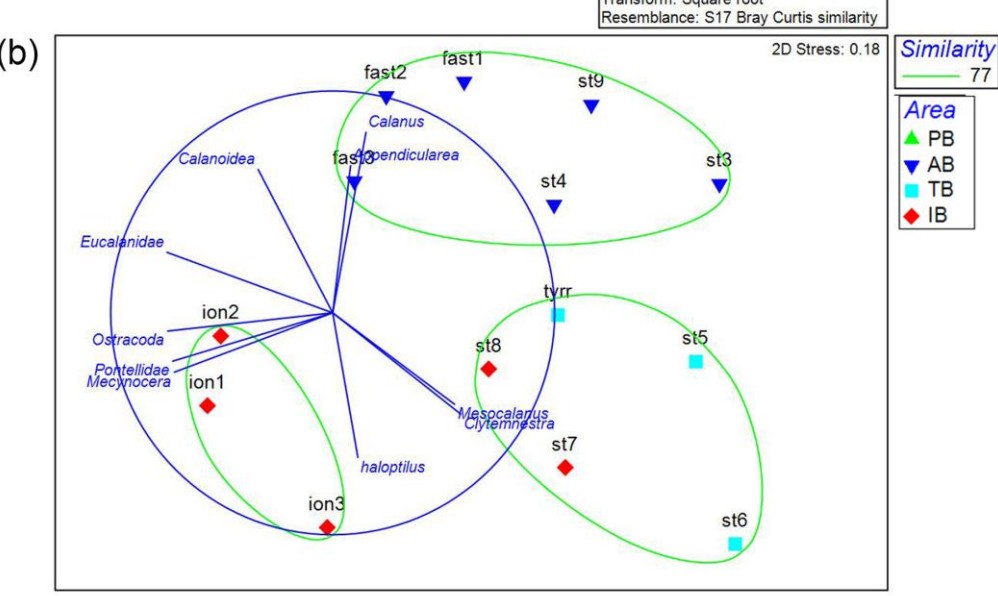


**Figure 8: NMDS analysis of the zooplankton taxa for all stations (a) excluding ST 1 and ST2 (b): plot of the stations and the taxa correlated at >0.65 with the axes. Colour of the stations represents the areas identified by the PCA in the environmental analysis (see Fig. 2). This analysis was performed on the zooplankton collected with the data from $N_{200}$. PB: Provencal Basin, AB= Algerian Basin, TB =Tyrrhenian Basin, IB = Ionian Basin.**


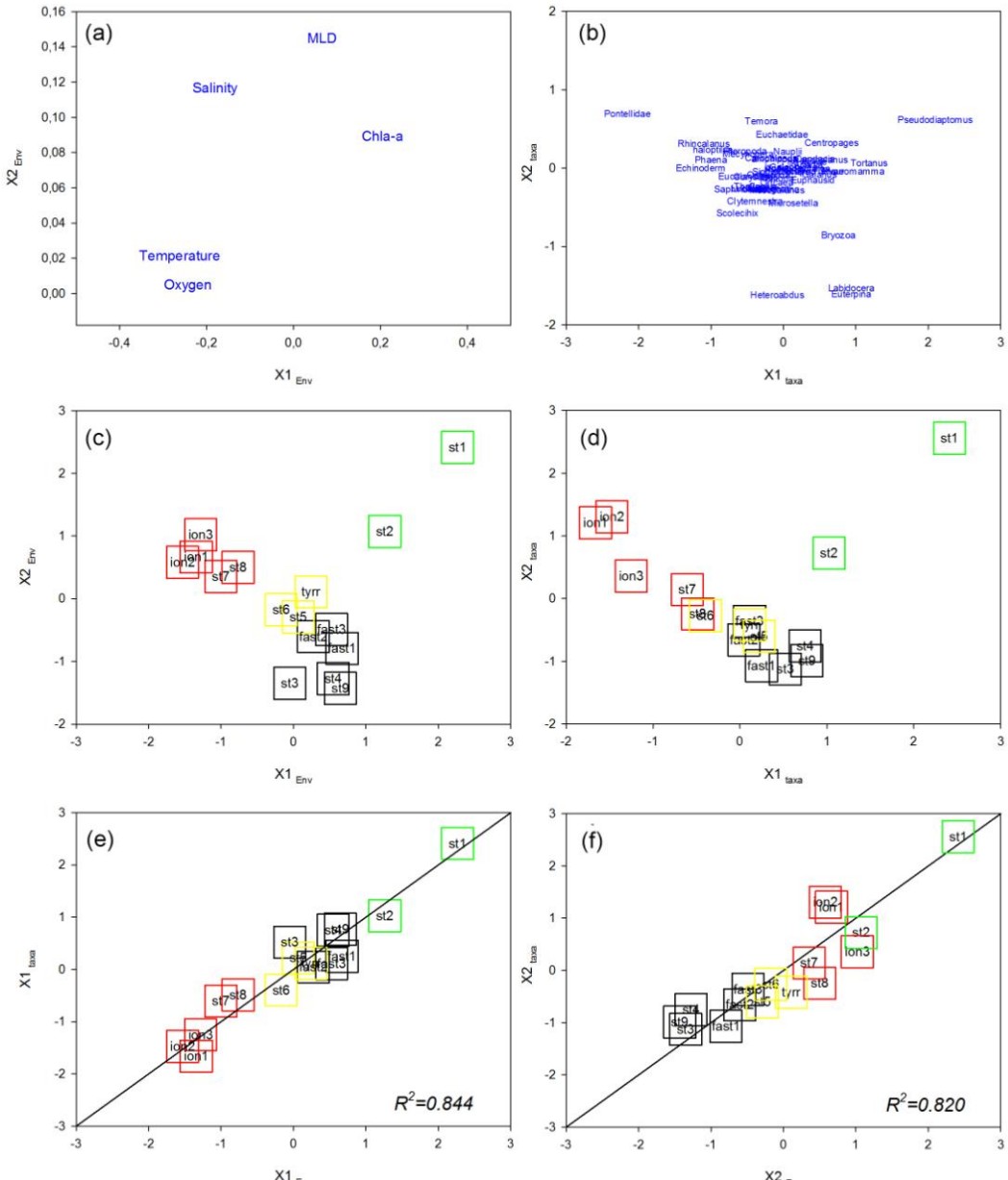

**Figure 9: Co-inertia analysis.** Ordination on the plans (1, 2) of the environmental variables (a) and the abundance of the zooplankton taxa (b) and of the stations in the 'Environment system' (c) and in the 'Zooplankton system' and plots of the stations on the first (c) and second (d) axes of the two systems. The line represents the equality between the coordinates on the two systems. Coloured squares identify the different regions: green = PB, black = AB, yellow = TB and red = IB.

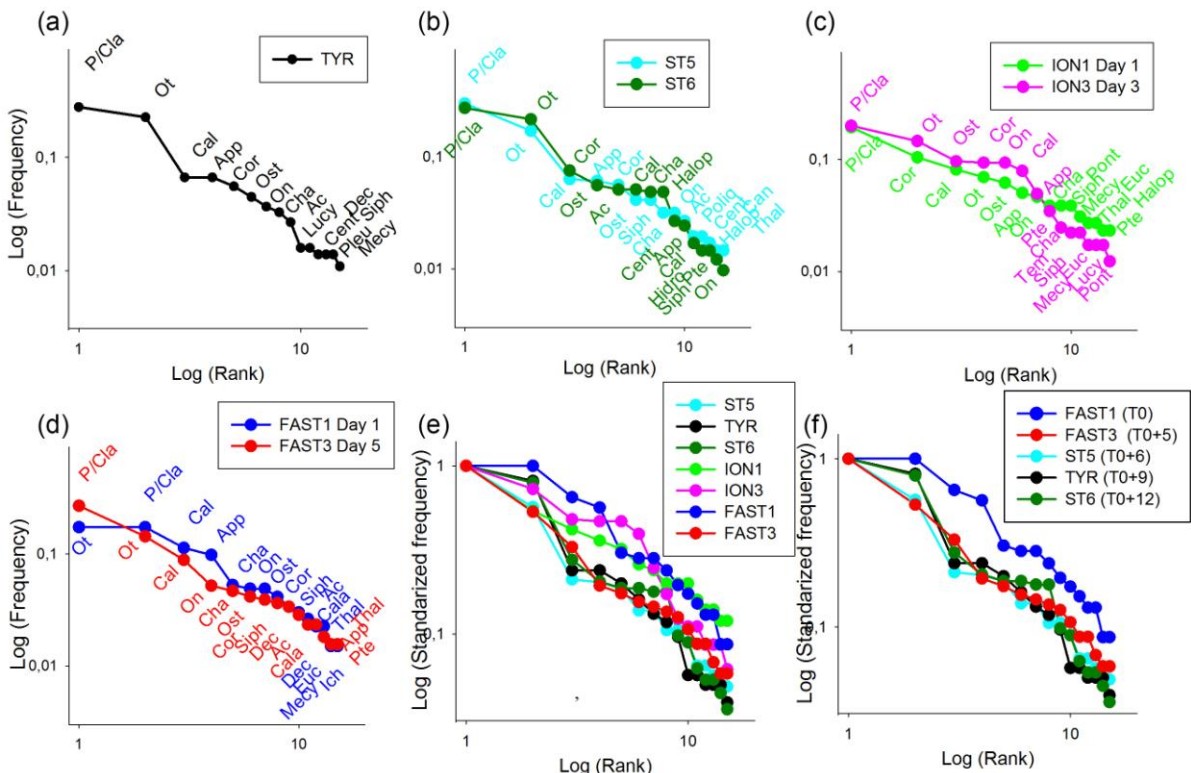

**Figure 10: Rank frequency diagram at stations TYR (a), ST5 and 6 (b)ION (c) FAST (d) and Log standardized frequency for all stations (e) and stations influenced by dust deposition (f). Ac:** *Acartia* **spp.; Cal: Calanoid copepods; Cala:** *Calanus* **spp.; Cent:** *Centropages* **spp.; Cor:** *Corycaeus* **spp.; Euc:** *Eucalanus* **spp.; Halop:** *Haloptilus***spp; Luci:** *Lucicutia* **spp.; Mecy:** *Mecynocera* **spp.; On:** *Oncaea* **spp.; Ot:** *Oithona* **spp.; P/Cla:** *Para/Clausocalanus* **spp.; Pleu:** *Pleuromamma* **spp.; Pont:Pontellidae; Tem: Temora** *spp***.; App: Appendicularia; Cha: Chaetognatha; Dec: Decapods; Hydro: Hydrozoans; Ich: Ichtyoplankton; Ost: Ostracods; Poly: Polychaeta; Pte: Pteropods; Siph: Siphonophores; Thal: Thaliaceans.**



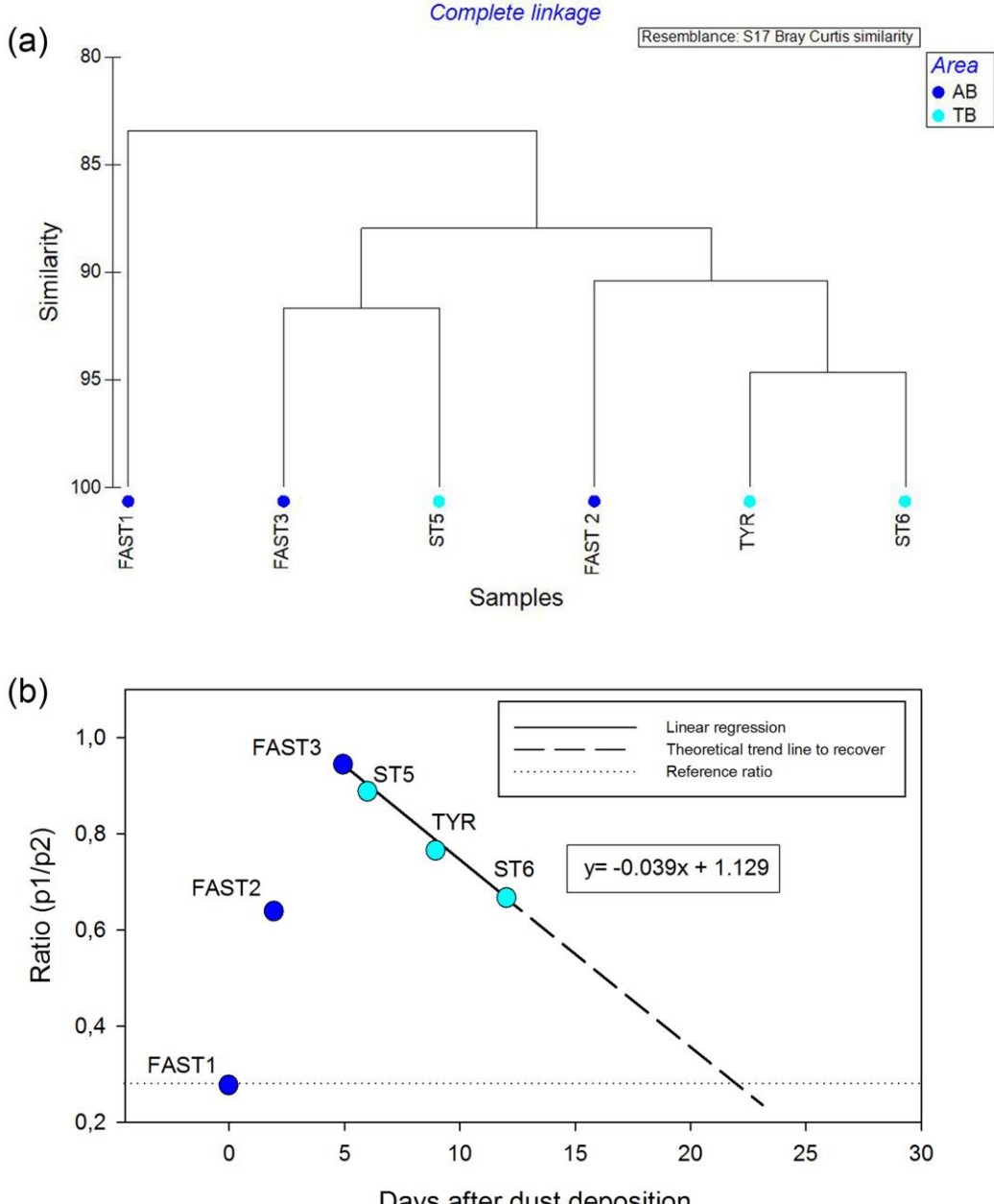

**Figure 11. Cluster analysis on rank frequency diagrams (a) and changing trends in the p1/p2 ratio (b) on the stations impacted by wet dust deposition.**


**Table 1. Sampled stations during the PEACETIME survey: geographical information, date and time of zooplankton net sampling. AB: Algerian Basin; PB: Provencal Basin; TB: Tyrrhenian Basin; IB: Ionian Basin.**

| Station ID | Area | lat (N) | long ( E) | Date (DD/MM/YYYY) | Time (HH:MM) |
|---|---|---|---|---|---|
| St1 | PB | 41°53,51 | 6°20,00 | 12/05/2017 | 11:30 |
| St2 | PB | 40°30,37 | 6°43,79 | 13/05/2017 | 9:30 |
| St3 | AB | 39°8,00 | 7°41,01 | 14/05/2017 | 9:15 |
| St4 | AB | 37°58,99 | 7°58,61 | 15/05/2017 | 9:15 |
| St5 | TB | 38°57,19 | 11°1,40 | 16/05/2017 | 7:05 |
| TYR | TB | 39°20,39 | 12°35,57 | 19/05/2017 | 23:00 |
| St6 | TB | 38°48,46 | 14°29,98 | 22/05/2017 | 10:15 |
| St7 | IB | 36°39,49 | 18°9,29 | 24/05/2017 | 2:00 |
| ION1 | IB | 35°29,38 | 19°46,51 | 26/05/2017 | 21:59 |
| ION2 | IB | 35°29,38 | 19°46,51 | 27/05/2017 | 8:50 |
| ION3 | IB | 35°29,38 | 19°46,51 | 28/05/2017 | 8:45 |
| St8 | IB | 36°12,62 | 16°37,86 | 30/05/2017 | 9:05 |
| St9 | AB | 38°8,08 | 5°50,45 | 01/06/2007 | 23:00 |
| FAST1 | AB | 37°56,81 | 2°54,99 | 04/06/2017 | 22:15 |
| FAST2 | AB | 37°56,81 | 2°54,99 | 06/06/2017 | 9:50 |
| FAST3 | AB | 37°56,81 | 2°54,99 | 08/06/2017 | 23:45 |


Table 2. Overview of the main characteristics of the wet dust events occurring during PEACETIME. Zooplankton sampling was carried out very close to a CTD cast except at FAST2 where the sampling was done between two casts respectively 9 hours after the first cast [a] and 16 hours before the second [b]. [*] Value measured on 17-05-2017


| Stations impacted by dust and cruise visit duration | Cruise strategy with regard to dust events | Dates, geographical characteristics and intensity of the dust events predicted by the model and by observations | Zooplankton sampling Date | Iron in aerosol ng m$^{-3}$ | Nutrients below the nutricline NO3 (n mol/l) PO4 (n mol/l) | Surface Primary production mg C m$^{-3}$ d$^{-1}$ | Water column (0-250) average Chl-a concentration mg m$^{-3}$ | Depth range of the DCM strata (m) | Mean concentration of Chl-a on DCM strata mg m$^{-3}$ | Ratio fluorescence phytoplankton $F_{micro}$:$F_{nano}$:$F_{pico}$ within the DCM strata |
|---|---|---|---|---|---|---|---|---|---|---|
| Wet dust event Tyrrhenian 16 to 22May | TB stations schedule before the cruise. Model predicted a dust event 6 days before the arrival | From 10 to 12 May; Impacted area: whole southern Tyrrhenian sea; predicted flux from models: >1 g m$^{-2}$ (Desboeufs et al. in prep) Dust event was confirmed by alumimium, iron and lithogenic Si measured in sediment traps at TYR over 4 days: cumulated lithogenic flux of 153 mg m$^{-2}$ at 200 m and 207 mg m$^{-2}$ at 1000 m (Bressac et al., in prep.) | ST5: 16-05-2017 | 57.3 | 841 148 | 1.68 | 0,12 | 70-80 | 0,55 | 21:48:30 |
| | | | TYR: 19-05-2017 | 162.3 | 54 [*] 127 | 1.77 | 0,11 | 70-80 | 0,61 | 33:40:27 |
| | | | ST6: 22-05-2017 | 189.8 | 488 136 | 1.66 | 0,07 | 70-80 | 0,36 | 7:44:49 |
| Wet dust event FAST 02 to 07 June | Station FAST schedule and position determined on board according to meteorological event | From 3 to 5 June; Impacted area: between Baleares and Algerian coast; predicted flux from models: 0.5 g m$^{-2}$ (Guieu et al., accepted, Supp Info figure SI5); On-board atmospheric dust deposition observations confirmed a weak wet dust deposition of 0.012 g m$^{-2}$ (Guieu et al., accepted). Cumulated lithogenic fluxes in sediment traps over 5 days: 50 mg m$^{-2}$ at 200 m and 70 mg m$^{-2}$ at 1000m (Bressac et al., in prep). Water column observations (nutrients, trace metals) (Van Wambeke et al., in prep, Tovar-Sánchez et al. 2020, Bressac et al., in prep) show a clear imprint of the atmospheric deposition. | FAST1: 04-06-2017 | 245.3 | 224 246 | 2.44 | 0,12 | 60-90 | 0,42 | 27:45:28 |
| | | | FAST2: 06-06-2017 | 266.0 | 808 239 | 2.85 | 0,14[a] 0,18 [b] | 60-100 [a] 70-90 [b] | 0,38[a] 0,86 [b] | 25:43:32 [a] 50:30:20 [b] |
| | | | FAST3: 08-06-2017 | 44.9 | 135 113 | 2.04 | 0,10 | 70- 90 | 0,42 | 20:49:31 |
| References of the data | Dulac (pers.com) Desboeufs et al. (in prep) Guieu et al. (accepted) . | Desboeufs et al. (in prep) Guieu et al. (accepted) Bressac et al. (in prep) Tovar-Sánchez et al. (2020) van Wambeke et al. (in prep) | Tovar-Sánchez et al. (2020) | | Van Wambeke (pers. comm.) | E. Maranon and M. Perez-Lorenzo | J.Uitz, C. Dimier | J.Uitz, C. Dimier | J.Uitz, C. Dimier | J.Uitz, C. Dimierl |

**Table 3.** Summary table of Spearman rank correlations. T°= temperature; Sal= salinity; Chl-a= Chlorophyll; MLD= Mix layer depth; pp= primary production. Bold characters indicate significant Rs value (p<0.05).

| | | | Correlation coeff. | | | |
|---|---|---|---|---|---|---|
| Abundance | T° | Sal | Chl-a | DO | MLD | PP |
| $C_{<200}$ | -0,49 | -0,43 | 0,32 | **-0,61** | -0,16 | -0,37 |
| $C_{200-300}$ | **-0,58** | -0,37 | 0,48 | **-0,58** | 0,08 | -0,24 |
| $C_{300-500}$ | -0,51 | -0,19 | **0,52** | -0,45 | 0,21 | -2,28 |
| $C_{500-1000}$ | **-0,56** | -0,50 | 0,23 | -0,49 | -0,06 | 0,05 |
| $C_{1000-2000}$ | 0,29 | 0,01 | -0,28 | 0,33 | -0,34 | 0,35 |
| $C_{>2000}$ | **-0,12** | **-0,53** | -0,15 | 0,08 | -0,50 | -0,16 |
| Total abundance | **-0,67** | -0,44 | **0,56** | **-0,68** | 0,08 | -0,28 |
| Biomass | | | | | | |
| $C_{<200}$ | **-0,61** | -0,48 | 0,42 | **-0,71** | -0,08 | -0,36 |
| $C_{200-300}$ | **-0,52** | -0,29 | 0,52 | -0,51 | 0,12 | -0,14 |
| $C_{300-500}$ | -0,49 | -0,18 | **-0,53** | -0,46 | 0,19 | -0,27 |
| $C_{500-1000}$ | -0,45 | -0,43 | 0,17 | -0,41 | -0,11 | 0,14 |
| $C_{1000-2000}$ | 0,24 | -0,05 | -0,37 | 0,32 | -0,39 | 0,30 |
| $C_{>2000}$ | -0,18 | **-0,61** | -0,10 | -0,02 | **-0,53** | -0,10 |
| total biomass | **-0,58** | **-0,62** | 0,24 | -0,43 | -0,27 | 0,08 |


**Table 4: Results of the one-way ANOVAs performed to test differences between areas (PB, AB, IB and TB) in abundance and biomass data for the different zooplankton size classes, for total zooplankton (cumulative of all size classes) and for mesozooplankton (ESD between 200 and 2000 μm) between the areas. Significant p-value <0.05 are marked in bold. *ns*= non-significant difference. Values of F and p in italic mark where Dunnett's test was used. In the post-hoc analysis homogeneous group with the lowest and highest values are noted with "a" and "b" respectively. PB= Provencal Basin, AB= Algerian Basin, TB= Tyrrhenian Basin, IB= Ionian Basin.**


| | Abundance | | | | | | Biomass | | | | | |
| | | | Sheffé post-hoc | | | | | | Sheffé post-hoc | | | |
| Size class | f | P | PB | AB | TB | IB | F | p | PB | AB | TB | IB |
|---|---|---|---|---|---|---|---|---|---|---|---|---|
| C200 | 3.19 | 0.067 | *ns* | *ns* | *ns* | *ns* | 3.64 | **0.048** | *ns* | *ns* | *ns* | *ns* |
| C200-300 | 3.46 | 0.055 | *ns* | *ns* | *ns* | *ns* | 2.55 | 0.109 | *ns* | *ns* | *ns* | *ns* |
| C300-500 | 4.4 | **0.029** | b | ab | ab | a | 5.03 | **0.020** | b | a | ab | a |
| C500-1000 | 3.01 | 0.076 | *ns* | *ns* | *ns* | *ns* | 1.75 | 0.214 | *ns* | *ns* | *ns* | *ns* |
| C1000-2000 | 14.77 | **0.000** | a | b | ab | b | 17.87 | **0.000** | a | b | ab | b |
| C>2000 | 9.25 | **0.002** | a | b | ab | ab | 11.63 | **0.001** | a | b | a | a |
| Total | 5.51 | **0.015** | b | ab | ab | a | 3.2 | 0.066 | *ns* | *ns* | *ns* | *ns* |
| Total mesozooplankton (200-2000 μm) | 5.03 | **0.020** | b | ab | ab | a | 1.06 | 0.405 | *ns* | *ns* | *ns* | *ns* |


**Table 5. PERMANOVA analysis on the environmental variables and on zooplankton taxa abundances: Pair-wise tests with unrestricted permutation of raw data (number of permutations: 999) for the comparison between the zones. Resemblance worksheets are based on Euclidean distance. Significant p-value <0.05 are marked in bold**


| Groups | Environmental variables | | | Zooplankton taxa abundances | | |
|---|---|---|---|---|---|---|
| | t | P(perm) | Unique Perms | t | P(perm) | Unique Perms |
| PB, AB | 3,78 | **0,044** | 28 | 2,08 | **0,049** | 28 |
| PB, TB | 3,24 | 0,101 | 10 | 2,01 | 0,094 | 10 |
| PB, IB | 5,65 | **0,043** | 21 | 2,47 | 0,056 | 21 |
| AB, TB | 1,79 | **0,014** | 84 | 1,65 | **0,008** | 84 |
| AB, IB | 5,91 | **0,001** | 400 | 1,67 | **0,004** | 404 |
| TB, IB | 4,59 | **0,016** | 56 | 1,57 | **0,045** | 56 |
| TB+st7 and 8, ION ST | 1.65 | 0.159 | 56 | 1,90 | **0,019** | 56 |

**Table 6. Estimated grazing, respiration and excretion rates of zooplankton based on allometric models (see methods) and their impact on the phytoplankton stock and production along the PEACETIME survey transect.**

| | Provencal basin | | Algerian basin | | | | | | Tyrrhenian Basin | | Ionian Basin | | | | |
|---|---|---|---|---|---|---|---|---|---|---|---|---|---|---|---|
| | st1 | st2 | fast1 | fast2 | fast3 | st9 | st3 | st4 | st5 | st6 | st8 | st7 | ion1 | ion2 | ion3 |
| **Grazing impact** | | | | | | | | | | | | | | | |
| Phytoplankton stock (mg C m$^{-2}$) | 1749 | 1632 | 1554 | 1691 | 1412 | 1805 | 1161 | 1458 | 1526 | 933 | 1582 | 1212 | 1376 | 1587 | 1587 |
| Primary Production (mgC m$^{-2}$d$^{-1}$) | 295 | 155 | 229 | 184 | 297 | 303 | 165 | 225 | 197 | 190 | 289 | 187 | 266 | 279 | 304 |
| ZCD (mgC m$^{-2}$ d$^{-1}$) | 280 | 155 | 274 | 263 | 249 | 228 | 224 | 278 | 202 | 145 | 195 | 205 | 204 | 244 | 177 |
| Grazing impact on Phyto. stock (%) | 16,0 | 9,5 | 17,7 | 15,6 | 17,7 | 12,7 | 19,3 | 19,1 | 13,3 | 15,6 | 12,4 | 17,0 | 14,8 | 15,4 | 11,2 |
| Grazing impact on PP (%) | 94,8 | 99,9 | 119,7 | 143,3 | 83,9 | 75,4 | 135,6 | 123,7 | 102,5 | 76,7 | 67,6 | 109,7 | 76,5 | 87,6 | 58,3 |
| **Respiration** | | | | | | | | | | | | | | | |
| Respiration (mg C m$^{-2}$ d$^{-1}$) | 112,2 | 64,3 | 95,3 | 90,1 | 86,2 | 81,3 | 83,8 | 100,2 | 78,7 | 62,9 | 75,6 | 77,0 | 72,4 | 94,7 | 71,6 |
| % of Primary production respired by zooplankton | 38,0 | 41,4 | 41,5 | 49,0 | 29,0 | 26,8 | 50,6 | 44,5 | 39,8 | 33,1 | 26,1 | 41,0 | 27,1 | 33,9 | 23,5 |
| **NH4 zooplankton contribution** | | | | | | | | | | | | | | | |
| Excretion (mg N-NH4 m$^{-2}$ d$^{-1}$) | 17,7 | 9,2 | 13,6 | 12,9 | 12,3 | 16,2 | 12,0 | 14,3 | 11,3 | 9,1 | 10,9 | 11,0 | 10,4 | 13,6 | 10,3 |
| Phytoplankton needs (mgN m$^{-2}$ d$^{-1}$) | 50,2 | 26,4 | 39,0 | 31,3 | 50,6 | 51,6 | 28,2 | 38,3 | 33,6 | 32,4 | 49,2 | 31,9 | 45,3 | 47,4 | 51,8 |
| N demand (%) | 35,2 | 34,9 | 34,9 | 41,1 | 24,3 | 31,5 | 42,6 | 37,4 | 33,6 | 28,0 | 22,1 | 34,6 | 22,9 | 28,8 | 19,9 |
| **PO4 zooplankton contribution** | | | | | | | | | | | | | | | |
| Excretion (mg P-PO4 m$^{-2}$ d$^{-1}$) | 2,3 | 1,3 | 2,0 | 1,9 | 1,8 | 1,7 | 1,8 | 2,1 | 1,6 | 1,3 | 1,6 | 1,6 | 1,5 | 2,0 | 1,5 |
| Phytoplankton needs (mg P m$^{-2}$d$^{-1}$) | 8,6 | 4,5 | 6,7 | 5,3 | 8,6 | 8,8 | 4,8 | 6,5 | 5,7 | 5,5 | 8,4 | 5,4 | 7,7 | 8,1 | 8,8 |
| P demand (%) | 27,3 | 29,7 | 30,4 | 35,9 | 21,3 | 19,5 | 36,8 | 32,5 | 28,6 | 23,5 | 18,6 | 29,6 | 19,7 | 24,1 | 16,6 |


**Table 7 : Comparison of zooplankton biomass and abundance in different areas of the Mediterranean Sea. ** wet weight**

| Area | Sampling period | Net mesh size (µm) | Layer (m) | Biomass (mg m$^{-3}$) | Abundance (ind m$^{-3}$) | Reference |
|---|---|---|---|---|---|---|
| NWMS - Provencal and Ligurian Seas | Feb 2013 | 120 | 0-250 | 12.3 (1.9-42.3) | 608 (21-2548) | Donoso et al. (2017) |
| NWMS - Provencal and Ligurian Seas | Apr 2013 | 120 | 0-250 | 64.5 (13.9-197.8) | 3668 (850-7205) | Donoso et al. (2017) |
| NWMS - Gulf of Lions shelf | Mar/Apr 1998 | 80-200 | 0-200 | 9.56 ± 4.73 | | Gaudy et al. (2003) |
| NWMS - Gulf of Lions shelf | Jan 1999 | 80-200 | 0-200 | 4.73 ± 2.53 | | Gaudy et al. (2003) |
| NWMS - Provencal sea | Mar 1969 | 200 | 0-200 | 0.4 - 53 | | Nival et al. ( 1975) |
| NWMS - Provencal sea | Apr 1969 | 200 | 0-200 | 10 - 210 | | Nival et al. ( 1975) |
| NWMS - Provencal sea | Spring 2008 | 200 | 0-200 | 13.15 ± 2.5 | 1731 | Mazzocchi et al. (2014) |
| NWMS - Provencal sea | Jul 1999 | 200 | 0-300 | | 383 | Siokou et al. (2019) |
| NWMS - Provencal sea | May/Jun 2017 | 100-200 | 0-300 | 5.5 ± 2.1 | 1638 ± 433 | this study |
| SWMS - Algerian sea | Jul-Aug 1997 | 200 | 0-200 | 8.2 (2.1-34.5) | 370 (36-844) | Riandey et al. (2005) |
| SWMS - Algerian sea | Jul 1999 | 200 | 0-300 | | 197 | Siokou et al. (2019 |
| SWMS- Algero Provencal sea | Jun/Jul 2008 | 200 | 0-200 | 5.4 | 1561 ± 205 | Nowaczyk et al. 2011 |
| SWMS- Algerian sea | May/Jun 2017 | 100-200 | 0-300 | 6.6 ± 0.6 | 1254 ± 191 | This study |
| Tyrrhenian Sea | Autumn 1986 | 200 | 0-50 | 3.6 - 32 | | Fonda Umani and de Olazábal (1988) |
| Coastal Tyrrhenian sea | 1984-2006 | 200 | 0-50 | | 1708 | Mazzocchi et al. (2011) |
| Tyrrhenian sea | Sep/Oct 1963 | 60-300 | 0-700 | 0.15-0.3 | | Cited from Champalbert, (1996) |
| Tyrrhenian sea | Jun/Jul 2008 | 200 | 0-200 | 3.2 | 1250 | Nowaczyk et al. 2011 |
| Tyrrhenian sea | Jun 1968 | Not specify | 0-200 | 5.8** | | Cited from Kovalev et al. (2003) |
| Tyrrhenian sea | May/Jun 2017 | 100-200 | 0-300 | 4.8 ± 1.1 | 1398 ± 108 | This study |
| Ionian sea | Apr/May 1999 | 200 | 0-100 | 6.0 ± 0.8 (eastern) | | Mazzochi et al. (2003) |
| Ionian sea | Apr/May 1999 | 200 | 0-100 | 8.2 to 13.4 (western) | | Mazzochi et al. (2003) |
| Ionian sea | Spring 1992 | 200 | 0-300 | | 219 | Mazzochi et al. (2003) |
| Ionian sea | Spring 1999 | 200 | 0-300 | | 193 | Mazzochi et al. (2003) |
| Ionian sea | Spring 2008 | 200 | 0-200 | 2.73 | 213 | Mazzocchi et al. (2014) |
| Ionian sea | Autumn 2008 | 200 | 0-200 | 3.25 | 338 | Mazzocchi et al. (2014) |
| Ionian sea | Jun/Jul 2008 | 200 | 0-200 | 8 | 1181 ± 630 | Nowaczyk et al. 2011 |
| Ionian sea | Jul 1999 | 200 | 0-300 | | 146 | Siokou et al. (2019 |
| Ionian sea | May/Jun 2017 | 100-200 | 0-300 | 5.1 ± 0.5 | 1003± 76 | This study |