# Peer review of "Structure and functioning of epipelagic mesozooplankton and response to dust deposition events during the spring PEACETIME cruise in the Mediterranean Sea"

_Biogeosciences, 2020_

## Referee Comment (RC1) · Tamar Guy-Haim (Referee) · 25 May 2020

The manuscript by Feliú et al. presents a study of the zooplankton community structure at 12 stations along four Mediterranean Sea basins: Provencal, Algerian, Tyrrhenian, and Ionian basins during the PEACETIME cruise. This cruise was set to study the effect of Saharan dust depositions on processes at the air-sea interface. The authors sampled the epipelagic layer (0-300m) using two mesh-size nets during/after dust events. Their major finding is that following Saharan dust deposition, the observed changes between basins are masked and that there is a larger contribution of small size fraction

(<500 $\mu$m) to the mesozooplankton community. The authors also performed indirect (allometry-based) assessments of zooplankton metabolism and showed that this fraction also contributes significantly to the N and P fluxes in the epipelagic layer.

While this ms provides important information for the methodology of zooplankton sampling in the Mediterranean Sea, I had major concerns with incompatibility between its main aims as declared in the title, abstract and introduction, and the data presented. The most important environmental data for testing the research aim – the intensity and the spatiotemporal distribution of the Saharan dust deposition event – is missing. The ms lacks appropriate data on dust and nutrients. I understand that this information will be part of other/future publications, but I do think that it is mandatory to include it in this ms as well, and the final acceptance of this ms should be conditional on presenting these data.

Other concerns relate to the lack of hypothesis setting, sampling justification, and statistical analyses. Also, this paper would benefit from some closer proofreading. It includes misuse of wording that can be easily corrected. It may be useful to engage a professional English language editor.

Nevertheless, the presented study is unique and scientifically worthwhile. I, therefore, suggest accepting this manuscript after major revision, pending correction of all major issues.

Major issues that must be resolved:

[1] The ms title "Structure and functioning of epipelagic mesozooplankton and response to dust events during the spring PEACETIME cruise in the Mediterranean Sea" as well as the abstract and introduction suggest/state that this study estimates the effect of dust deposition on zooplankton structure and function. Nevertheless, the ms does not include any data on dust deposition (e.g., in terms of Aluminum concentrations – see Measures Vink 2000 or Titanium in Dammshäuser et al. 2011) and nutrients in the sampling stations. Without this information, it is impossible to provide a reliable

quantitative assessment of the effects of dust deposition on zooplankton communities. The only dust-related information given within the ms (as personal communication with C. Guiro) is that on May 10 there was "a quite important dust event". This, by all means, is not satisfactory for differentiating dust effects from natural variability.

Measures, C.I. and Vink, S., 2000. On the use of dissolved aluminum in surface waters to estimate dust deposition to the ocean. Global biogeochemical cycles, 14(1), pp.317-327. Dammshäuser, A., Wagener, T. and Croot, P.L., 2011. Surface water dissolved aluminum and titanium: Tracers for specific time scales of dust deposition to the Atlantic?. Geophysical Research Letters, 38(24).

[2] The ms includes data on zooplankton communities, but is mainly descriptive and include hypothesis-free statistical comparisons. While descriptive articles are of values, it would be useful to set working hypotheses in the introduction, and, along with environmental and biological data, use appropriate statistical methods for testing these hypotheses. Such hypotheses could be bottom-up enhancement of zooplankton communities triggered by dust deposition, west to east gradient, etc. It is unclear why multivariate data analysis was performed to explore spatial changes of environmental parameters – but not for the biological component. The relationship between environmental biological data could be examined, for example, using BIOENV/BVSTEP. The contributions of significant taxa to specific stations/basins could be tested using SIMPER procedure. In addition, two-way ANOVA should be performed to test difference of univariate parameters (abundance, biomass) between size class X ecoregions/basins.

Minor comments

Title

Add "deposition" to the title ("dust deposition events").

Abstract

L27: it is hard to understand the meaning of "long station". I suggest changing to

"long-duration sampling station".

Introduction

L41: For P limitation the following paper may be cited:

Thingstad, T.F. and Rassoulzadegan, F., 1995. Nutrient limitations, microbial food webs and 'biological C-pumps': suggested interactions in a P-limited Mediterranean. Marine Ecology Progress Series, pp.299-306.

Materials and Methods

L85: state whether the Bongo net was towed vertically or in oblique mode.

L83-85: was flow meter used to quantify the filtered volume? Or was it calculated based on net dimensions?

L83-87: clarify why was the entire epipelagic depth strata (0-300m) chosen for sampling. It can be hypothesized that dust deposition will mostly affect the uppermost strata that is more affected by atmospheric deposition, thus sampling all along 0-300m may mask such possible effect. Nevertheless, diel migration and fecal pellet deposition may affect the whole epipelagic community.

L89: which model of FlowCAM was used? Were the flow cells of type FOV (field of view)? Non-FOV flow cells are not efficient for quantitative measurements, and are best used for qualitative-only assessment of plankton as the transport of particles via the flow cell is not constrained (see Detmer et al 2019).

Detmer, T.M., Broadway, K.J., Potter, C.G., Collins, S.F., Parkos, J.J. and Wahl, D.H., 2019. Comparison of microscopy to a semi-automated method (FlowCAM$^{®}$) for characterization of individual-, population-, and community-level measurements of zooplankton. Hydrobiologia, 838(1), pp.99-110.

L128: detail ZooProcess version, add citation.

L128-129: which software was used for automatic classification of vignettes?

L135: ESD size categories 0.2 to 2.0 $\mu$m does not make sense (likely mm). Also, results show also 0-100 $\mu$m and 100-200 $\mu$m categories (Fig. 2).

L155-156: shortly describe the model for assessment of oxygen consumption, and excretion of NH3 and PO4.

L164: add citation for PRIMER software.

L161-177: multivariate analysis was done separately for environmental data (PCA) and biological data (nMDS). These datasets should be correlated using BIOENV/BVSTEP or RELATE procedures. In addition, PERMANOVA can be used to statistically test the differences between basins. Also, the contribution of certain species for the dis/similarities between stations and basins should be statistically explored using SIM-PER.

L168-169: RFD analysis here is only descriptive. To test differences between RFDs, RAD analysis (max rank normalization method) can be used: Saeedghalati, M., Farah-pour, F., Budeus, B., Lange, A., Westendorf, A.M., Seifert, M., Küppers, R. and Hoff-mann, D., 2017. Quantitative comparison of abundance structures of generalized communities: from B-cell receptor repertoires to microbiomes. PLoS computational biology, 13(1), p.e1005362.

L168-171: it would be more convenient to summarize all Spearman rank correlations and t-tests in two tables. I could not find t-test results except one testing the difference between N100 and N200 mesh (L107).

Results

L176-178: "mostly influenced by. . ." how was this determined? Visually? This can be tested statistically using correlation between the environmental factors and the 1st axis (or also the 2nd axis) of the PCA.
L179-184: also here seems that non-statistical assessment was done to relate biological data to environmental parameters. See above comment on the use of statistical procedures (BIOENV, RELATE, PERMANOVA).

L186-L261 and throughout the text: abundance of zooplankton is presented in ind.m-2 instead of ind.m-3. Similarly biomass, mgDW.m-2 instead of mgDW.m-3.

L188-198: no statistics! See above comment re ANOVA.

L199: "C300-500 biomass is positively correlated with Chl-a (r=-0.54, p=0.024)". r is negative so it is negatively and not positively correlated.

L211-223: missing statistics – see comment above re SIMPER.

L229: change sub-header to "zooplankton community changes at long-duration sampling stations".

Discussion

L265-278: the methodological concerns are presented as a major outcome of this study although they are mostly a by-product. Nevertheless, they provide important conclusions. I would reorder the sections, and put this as the second or third section.

L293-310: the authors compare the zooplankton abundances and biomass measured in this study (372000 ind m-3 and 1707 mgDW m-3 respectively reported in L186-187 and in figure 4) to other studies (table 3). Unlike their statement that the results of this study are in the same order of magnitude as previous studies – it seems that the changes are enormous! For example, Donoso et al 2017 measured 608 ind m-3 and 64 mgDW m-3. This is a change of 3 orders of magnitude in abundance and 2 in biomass! Note that the reported values are not the same in figure 4 and table 3. Please explain these differences.

L293-310: additional comparison should be performed to Mediterranean studies that used 100 $\mu$m mesh in addition to a larger mesh size and measured abundance and

biomass. For example:

Koppelmann, R., Böttger-Schnack, R., Möbius, J. and Weikert, H., 2009. Trophic relationships of zooplankton in the eastern Mediterranean based on stable isotope measurements. Journal of Plankton Research, 31(6), pp.669-686.

L389-410: phytoplankton biomass and production assessment were calculated from Chl-a rather than being measured directly. Similarly, zooplankton carbon demand, oxygen consumption and excretion was based on multiple assumptions, including the use of constant biomass to carbon conversion factor, carbon to Chl-a ratio, Redfield ratio and respiratory quotient. While these are all legitimate methods, it must be discussed that over large geographical scales that include environmental gradient – these factors may vary and thus these assessments may be inaccurate. For example, Minutoli Guglielmo 2009 showed an increasing trend in ETS activity from west to east:

Minutoli, R. and Guglielmo, L., 2009. Zooplankton respiratory Electron Transport System (ETS) activity in the Mediterranean Sea: spatial and diel variability. Marine Ecology Progress Series, 381, pp.199-211.

Figures and Tables

Fig. 2: in the Zooscan and FlowCAM analyses, there is a small overlap in the fractions 300-400 and 400-500. Nevertheless, it seems that the combined data (fig 2c) include these fractions from the flowCAM only. Shouldn't these fractions be accumulative?

Fig. 3: show the

Fig. 4: correct Chla-a to Chl-a

Table 3: is biomass presented in all mentioned studies as dry weight?

Please also note the supplement to this comment:
https://www.biogeosciences-discuss.net/bg-2020-126/bg-2020-126-RC1-supplement.pdf

---

## Referee Comment (RC2) · Anonymous Referee #2 · 27 May 2020

The manuscript presents a complete set of multidisciplinary data of zooplankton community structure and functioning from an oceanographic cruise in the Western Mediterranean Sea in late spring-early summer (May to June 2017) during two major dust events in the Algerian and Tyrrhenian basins. Investigating mesozooplankton structure in the western Mediterranean Sea is a classic but necessary marine science research approach to improve our knowledge on the mesozooplankton community and estimate the responses of this key trophic group in the pelagic ecosystem linking small primary and secondary producers to higher nektonic trophic levels. Furthermore

this study significantly contributes on the effect of the Saharan dust deposition on the zooplankton community structure. Several papers have been produced in the scientific world and at the Mediterranean Sea level, dealing with mesozooplankton composition, distribution and structure. Nevertheless, this study pushes the analysis deeper up to mesozooplankton functioning and estimate zooplankton growth, ingestion and metabolism using allometric relationships. This paper is very well written and organized, but a clear scientific question or at least two hypotheses should appear in the introduction instead of a description of the objectives as general scientific tasks. For instance what do you expect during the Sahara dust deposition, what should be the effect of the Sahara dust to the zooplankton community and how your data can show this influence. Also, the major part of the discussion is mainly based on regional differences and comparisons and less discussion has been made on the effect of Sahara dust on zooplankton estimated vital rates. Are there any differences of the vital rates before and after the Sahara dust events? What is the response of the zooplankton community after the Sahara dust deposition besides the changes in community structure? Please find below the detailed comments on the manuscript. Line 47: Add also the reference Siokou et al., 2019.( Deep-Sea Research Part II 164 (2019) 170–189). Line 58: Explain better why this cruise should be "flexible". The manuscript Guieu et al., 2020 has not been published so it is difficult to understand the design of the oceanographic cruise. Line 60: Add also the main questions and hypotheses of this study. Line 80: Explain why did you calculate the depth of the Mixed Layer. In this study there is no information at all about the hydrology of the area so it is difficult to follow. Only at the end in the discussion chapter the authors clarified the hydrological features existing in the area. Lines 85-86: Explain why do you perform the zooplankton sampling from the surface until 300 m? Is it due to the euphotic zone? Is it due to the hydrological features of the area? Line 140: Explain what negative and positive values of the NBSS slopes means. Line 153: Explain why did you use the conversion factor of C:Chla=50 and add the reference. For oligorophic waters are more suitable to use the conversion factor/equation of Malone et al 1993, which
is used the different Chla values according to depth. Line 157: Add equation model for ammonium, phosphorus excretion and oxygen consumption rates as you did with the other relationships. Line 200: Throughout the text, there are several definitions for the small zooplankton, sometimes<1mm, <300, <500 $\mu$m. Please clarify in order to avoid any confusion. Line 221: you wrote "due to higher relative abundance of small copepods" please specify what species and which size. Also ostracods are not copepod species. Pontellidae family is written twice, and some species in Pontellidae are not small. Please specify if possible which species. Line 225-229. The explanation for the strong variations of the NBSS that is due to the migration of the larger species is not very clear, According to the Fig.4 the abundances of large species are quite similar between day and night sampling. Unless, you will approve that there is statistical significant differences between the night and day samples. Line 234: Clausocalanus and Oithona species according to the literature are not herbivorous species. Line 253: Add in your Methods how or who did the Primary Production measurements. Line 316: Specify the hydrological features of the area. A short chapter for the study area could be very helpful. Line 370: In methodology you wrote that the contribution to nutrient regeneration by zooplankton was estimated using the values of primary production and converted to nitrogen and phosphorus requirement using Redfield ratio. However, the calculation doesn't follow the Redfield ratio of C:N:P = 106:16:1. Have you used this, or did you use the ratio that you found during the study? Please clarify. Line 340-375: In this chapter the authors are reported several times the different vital rates of the zooplankton size fractions. However, these data are not provided in Table 2. I could be useful to add biomass data of the total zooplankton as well as of the different size fraction in the Table 2. Line 392: Delete "species composition" since no data is shown in Table 2. Line 410: Please add also the taxa that correspond to each stage of succession. Lines 417-420: Explain how do you know that at the beginning you had small phytoplankton and then large? Because, according to Line 396 (pers. comm. J. Uitz), size and species composition of the phytoplankton community in FAST did not show any change after the dust. Lines 410-425: This

paragraph could be the main hypothesis of your study. Table 1: Add information about the dust events to follow better in the text. Table 2: Add biomass data of total zooplankton and size fractions. Table 3: Add Siokou et al., 2019 and make the comparison.

Please also note the supplement to this comment:
https://www.biogeosciences-discuss.net/bg-2020-126/bg-2020-126-RC2-supplement.pdf

---

## Author Comment (AC1) · 24 Jul 2020

| :--- |
| **Referee 1: Tamar Guy-Haim (Referee) (**tamar.guy-haim@ocean.org.il) |
| |

*Note: All the frames contain the comments of referee R1, and have been identified by a number. Our answers are below each frame, and the underlined paragraphs in yellow correspond to new or modified paragraphs in the manuscript. R2 refers to answers common to comments of referee 2.*

| Comment R1.1A |
| :--- |
| The manuscript by Feliú et al. presents a study of the zooplankton community structure at 12 stations along four Mediterranean Sea basins: Provencal, Algerian, Tyrrhenian, and Ionian basins during the PEACETIME cruise. This cruise was set to study the effect of Saharan dust depositions on processes at the air-sea interface. The authors sampled the epipelagic layer (0-300m) using two mesh-size nets during/after dust events. |
| Their major finding is that following Saharan dust deposition, the observed changes between basins are masked and that there is a larger contribution of small size fraction (<500 _m) to the mesozooplankton community. The authors also performed indirect (allometry-based) assessments of zooplankton metabolism and showed that this fraction also contributes significantly to the N and P fluxes in the epipelagic layer. |
| While this ms provides important information for the methodology of zooplankton sampling in the Mediterranean Sea, I had major concerns with incompatibility between its main aims as declared in the title, abstract and introduction, and the data presented. |

*Answer to comment R1.1A.*
*We have tried to answer all your comments and have substantially rewritten several paragraphs of the manuscript. We appreciate all your comments and we acknowledge that they have encouraged us to explore further our data. We hope that now the revised version of the manuscript shows more consistency between the main aims and the presented results.*

| Comment R1.1B |
| :--- |
| The most important environmental data for testing the research aim – the intensity and the spatiotemporal distribution of the Saharan dust deposition event – is missing. The ms lacks appropriate data on dust and nutrients. I understand that this information will be part of other/future publications, but I do think that it is mandatory to include it in this ms as well, and the final acceptance of this ms should be conditional on presenting these data. |

*Answer to comment R1.1B:*
*We have produced a full synthesis of the Saharan dust deposition events encountered during PEACETIME (See paragraph and table below). A lot of information is now presented in accepted and submitted papers of the special issue including the introduction paper by Cecil Guieu. We propose to include a synoptic table in our revised manuscript to summarize and comment on this information*

| Comment R1.1C |
| :--- |
| Other concerns relate to the lack of hypothesis setting, sampling justification, and statistical analyses. |

*Answer to comment R1.1C:*
*Hypotheses have been better formulated and new statistical analyses have been performed (See detailed answers below)*

Comment R1.1D:
Also, this paper would benefit from some closer proofreading. It includes misuse of wording that can be easily corrected. It may be useful to engage a professional English language editor.

*Answer to comment R1.1D:*
*The revised paper has been proofread by a native English speaker specialized in scientific publications.*

Comment R1.1E: Nevertheless, the presented study is unique and scientifically worthwhile. I, therefore, suggest accepting this manuscript after major revision, pending correction of all major issue

Comment R1.2 :
Major issues that must be resolved
 [1] The ms title "Structure and functioning of epipelagic mesozooplankton and response to dust events during the spring PEACETIME cruise in the Mediterranean Sea" as well as the abstract and introduction suggest/state that this study estimates the effect of dust deposition on zooplankton structure and function. Nevertheless, the ms does not include any data on dust deposition (e.g., in terms of Aluminum concentrations – see Measures Vink 2000 or Titanium in Dammshäuser et al. 2011) and nutrients in the sampling stations. Without this information, it is impossible to provide a reliable quantitative assessment of the effects of dust deposition on zooplankton communities.
The only dust-related information given within the ms (as personal communication with C. Guieu) is that on May 10 there was "a quite important dust event". This, by all means, is not satisfactory for differentiating dust effects from natural variability.
Measures, C.I. and Vink, S., 2000. On the use of dissolved aluminum in surface waters to estimate dust deposition to the ocean. Global biogeochemical cycles, 14(1), pp.317-327. Dammshäuser, A., Wagener, T. and Croot, P.L., 2011. Surface water dissolved aluminum and titanium: Tracers for specific time scales of dust deposition to the Atlantic?. Geophysical Research Letters, 38(24).

*Answer to comment R.1.2:*
*As the other referee raised a similar question (see comment R2.20), this answer is common to both.*

*As already mentioned in the general comment R1B, we have now produced a full synopsis of the Saharan dust deposition events encountered during PEACETIME. A lot of information is now presented in accepted and submitted papers of the special issue including the introduction paper by Cecile Guieu.*
*We present this new information in an additional paragraph in the Methods section.*

[revised manuscript text omitted]

Comment R1.3A:
Major issues that must be resolved ...
[2] The ms includes data on zooplankton communities, but is mainly descriptive and include hypothesis-free statistical comparisons. While descriptive articles are of values, it would be useful to set working hypotheses in the introduction, and, along with environmental and biological data, use appropriate statistical methods for testing these hypotheses. **H1** Such hypotheses could be bottom-up enhancement of zooplankton communities triggered by dust deposition, **H2** west to east gradient, etc.

*Answer to comment R1.3A: As the other referee raised a similar question (see comment R2.1A), this answer is common to both.*

*We agree with the referee regarding this comment, which has been very helpful in stimulating a redefinition of certain aspects of the focus of the paper.*

*We have now stated these general hypotheses:*

***H1: Is zooplankton structure impacted by dust deposition?.*** *We hypothesize that Saharan dust deposition events had an impact on the zooplankton community in the Mediterranean Sea, modifying its abundance, biomass, metabolic rates (no observed changes except in in $C_{500-1000}$) and diversity (The changes we observed are mostly related to community structure and taxonomic diversity (RFD) and to size distribution) .*

***H2: Hydrodynamical regions versus dust impact region***

*- It is difficult to relate zooplankton response to regional differences (lack of stations). Response is due to dust deposition and mainly affect the species diversity (RFD) and size structure (p1/p2 ratio)*

***H3:*** *How long should we observe the zooplankton structure in a given area to clearly observe the dust impact (p1/p2 ratio).*

*The results providing a basis for validating these hypotheses have been analyzed following several statistical treatments as detailed in following answers. (See below, comments R1.2B, R1.2C, R1.2D)*

*The following paragraph was added at the end of the Introduction:*

These objectives will serve to test the following hypotheses: whether the Saharan dust events impact the zooplankton community structure following deposition (H1), and if so, whether the effect would be immediately observable or after a lag time (H2). Finally, whether changes in zooplankton structure driven by dust deposition exceed regional differences under oligothropic conditions (H3).
* * *
Comment R1.3B
Major issues that must be resolved ...
It is unclear why multivariate data analysis was performed to explore spatial changes of environmental parameters – but not for the biological component. The relationship between environmental biological data could be examined, for example, using BIOENV/BVSTEP.
* * *
*Answer to Comment R1.3B*

*We agree with the referee's suggestion regarding the lack of more in-depth investigation of the relationships between environmental and biological data. We have decided to add a new paragraph in the Results section*
"3.3 Relationship between environmental variables and zooplankton community".

*To complement our previous results according to the referee's suggestion, we have used the BIOENV algorithm to select the environmental variables best explaining the community pattern. The BIOENV results show that salinity and chlorophyll were the environmental variables best explaining the overall spatial distribution of zooplankton community (BIOENV; Rs = 0.657).*

*To take this analysis further and to better explore the relationships between the zooplankton community and the environmental conditions defining their habitats , we have also performed a Co-inertia analysis, which allows comparison of spatial trends in the two data sets.*

*Accordingly, we propose these new paragraphs:*

Methods
The relationships between the biological and the environmental variables were also studied by coupling multivariate analyses of two datasets. The first dataset featured the abundances of all the zooplankton taxa identified from the 200µm net samples, and the second recorded environmental variables (the same as for the PCA analysis). A factorial correspondence analysis (FCA) and a principal

component analysis (PCA) were performed on these two data sets, respectively. Then the results of the two analyses were associated through a co-inertia analysis (Doledec and Chessel, 1994) performed using ADE-4software (Thioulouse et al., 1997). Prior to the analyses, the data were log-transformed to tend towards the normality of the distributions.

Results

The first factorial plane of the Co-inertia analysis (Figure 9) explained 96% of the total variance, with 79 % due to the first axis. On both spaces ('Environment' and 'Zooplankton'), the first axis opposes the IB stations associated with high temperature and salinity values and several zooplankton taxa (namely Echinoderm larvae and some copepod taxa, ie Pontellidae, *Rhincalanus* spp., *Haloptilus* spp. and *Phaena* spp.) to the PB and AB stations characterized by higher chlorophyll concentrations and by some copepod taxa (mainly *Pseudodiaptomus* spp., *Tortanus* spp. and *Pleuromama* spp.). On this axis, TB stations have an intermediate position, close to the coordinate zero. The second axis opposes northern (st1 and 2 of PB) and southern (AB) stations sampled in the Western Mediterranean basin. On this axis, PB stations are characterized by higher chlorophyll and salinity and deeper MLD, compared to AB and by the association with *Pseudodiaptomus* spp., whereas southern AB stations are associated with the copepods *Heterorhabdus* spp., *Labidocera* spp. and *Euterpina* spp. As in the preceding multivariate analyses, we note that St 8 and 9 from the IB tend to be closer to the TB stations than to the Ion station on the first factorial plane, particularly in the 'Zooplankton system'. The association between the environmental context and the zooplankton community is high with good correlation between the normalized scores of the stations (R2=0.844 and R2=0.820 for X1 and X2 axes, respectively), and by the positions of the plots of these stations close to the equality lines (i.e. X1 zooplankton = X1 Environment or X2 zooplankton = X2 Environment).

New figure 9: Co-inertia analysis. Ordination on the plans (1, 2) of the environmental variables (A) and the abundance of the zooplankton taxa (B) and of the stations in the 'Environment system' (C) and in the 'Zooplankton system' and plots of the stations on the first (C) and second (D) axes of the two systems. The line represents the equality between the coordinates on the two systems. Coloured squares identify the different regions: green = PB, black = AB, yellow = TB and red = IB.

[Figure]

Discussion
Interestingly, in the Co-inertia analysis the stations impacted by the dust (FAST and TB stations) are grouped on the left side of the relationship between X2 axis of environment and zooplankton. In addition, their succession in this graph is consistent with the sequence observed in the virtual time series of RFD (with FAST1 as the initial station before the dust deposition and TYR and St6 corresponding to day 9 and 12 after the dust event) showing the coupled impact of dust on both environment and zooplankton.

Comment R1.3C
The contributions of significant taxa to specific stations/basins could be tested using SIMPER procedure.

*Answer to Comment R1.3C*
*We agree with the referee and we have done the test. These new paragraphs have been added to the manuscript*

In section 3.2 of the manuscript, we added this new paragraph:
"The SIMPER analysis shows that the lower average similarity between the stations is in PB (64.79 %) mainly due to *Para/Clausocalanus* spp. The rest of the basins share a higher internal similarity 78.43 %, 79.79 % and 78.03 % for AB, TB and IB respectively. Another interesting point highlighted in the SIMPER analysis is the lower average dissimilarity between TB and ST7 and ST8 from (20.25 %), this dissimilarity increases when the comparison is made between TB and the rest of the stations included in IB (29.04 %); this is in agreement with the NMDS analysis (Figure 8) that related ST7 and ST8 with TB rather than with the stations in their basin."

In section 3.3 of the manuscript, we have added this new paragraph:
"Results of the PERMANOVA analysis on the environmental variables and on diversity on taxa are summarized on the following in Table 5. Interestingly, based on the zooplankton diversity of TB and IB, their difference is more significant when ST7 and ST8 are removed from IB and placed on TB (based on the NMDS cluster, Figure 8), whereas it is not the case when considering environmental variables (see Table below). This suggests that the similarity between st7 and st8 and the TB stations is not linked to the environmental context."

New Table 5. PERMANOVA analysis on the environmental variables and on zooplankton taxa abundances: Pair-wise tests with unrestricted permutation of raw data (number of permutations: 999) for the comparison between the zones. Resemblance worksheets are based on Euclidean distance.

| Groups | Environmental variables | | | Zooplankton taxa abundances | | |
|---|---|---|---|---|---|---|
| | t | P(perm) | Unique Perms | t | P(perm) | Unique Perms |
| PB, AB | 3,78 | 0,044 | 28 | 2,08 | 0,049 | 28 |
| PB, TB | 3,24 | 0,101 | 10 | 2,01 | 0,094 | 10 |
| PB, IB | 5,65 | 0,043 | 21 | 2,47 | 0,056 | 21 |
| AB, TB | 1,79 | 0,014 | 84 | 1,65 | 0,008 | 84 |
| AB, IB | 5,91 | 0,001 | 400 | 1,67 | 0,004 | 404 |
| TB, IB | 4,59 | 0,016 | 56 | 1,57 | 0,045 | 56 |
| TB+st7 and 8, ION ST | 1.65 | 0.159 | 56 | 1,90 | 0,019 | 56 |

Comment R1.3D
In addition, two-way ANOVA should be performed to test difference of univariate parameters (abundance, biomass) between size class X ecoregions/basins.

Answer to comment R1.3D
*Following the referee's suggestion, differences in the abundance and biomass of the zooplankton between size classes and basins were tested using two-way ANOVA. One-way ANOVA with Scheffé post-hoc analysis was also applied for comparison of the each size class between basins. Prior to analyses, data were log-transformed and tested for homogeneity. Dunnett's test was used in case of non-homogeneity.*

*This new paragraph has been added to the manuscript*

The two-way ANOVA shows that the PB basin is characterized by significantly lower abundance and biomass in the upper size classes (1000-2000µm and >2000µm) compared to the other areas (p<0.05). One-way ANOVA results show that both total zooplankton and mesozooplankton present significantly higher abundance in PB than in IB, whereas their total biomass was not significantly different between the areas (p>0.05). Significant differences in abundance and biomass between areas were found in the size classes C300-500, C1000-2000 and C>2000 and the biomass of C<200 (P<0.05) (Table 4).

New Table 4: Results of the one-way ANOVAs performed to test differences between areas (PB, AB, IB and TB) in abundance and biomass data for the different zooplankton size classes, for total zooplankton (cumulative of all size classes) and for mesozooplankton (ESD between 200 and 2000 µm) between the areas. Significant p-value <0.05 are marked in bold. ns= non-significant difference. Values of F and p in italic mark where Dunnett's test was used. In the post-hoc analysis homogeneous group with the lowest and highest values are noted with "a" and "b" respectively. PB= Provencal basin, AB= Algerian basin, TB= Tyrrhenian basin, IB= Ionian basin.

| | Abundance | | | | | | Biomass | | | | | |
| | | | Sheffé post-hoc | | | | | | Sheffé post-hoc | | | |
| Size class | f | P | PB | AB | TB | IB | F | p | PB | AB | TB | IB |
|---|---|---|---|---|---|---|---|---|---|---|---|---|
| C200 | 3.19 | 0.067 | *ns* | *ns* | *ns* | *ns* | 3.64 | **0.048** | a | a | a | a |
| C200-300 | 3.46 | 0.055 | *ns* | *ns* | *ns* | *ns* | 2.55 | 0.109 | *ns* | *ns* | *ns* | *ns* |
| C300-500 | 4.4 | **0.029** | b | ab | ab | a | 5.03 | **0.020** | b | a | ab | a |
| C500-1000 | *3.01* | *0.076* | *ns* | *ns* | *ns* | *ns* | 1.75 | 0.214 | *ns* | *ns* | *ns* | *ns* |
| C1000-2000 | 14.77 | **0.000** | a | b | ab | b | 17.87 | **0.000** | a | b | ab | b |
| C>2000 | 9.25 | **0.002** | a | b | ab | ab | 11.63 | **0.001** | a | b | a | a |
| Total | 5.51 | **0.015** | b | ab | ab | a | *3.2* | *0.066* | *ns* | *ns* | *ns* | *ns* |
| Total mesozooplankton (200-2000 µm) | 5.03 | **0.020** | b | ab | ab | a | *1.06* | *0.405* | *ns* | *ns* | *ns* | *ns* |

*Figure: two-way ANOVA to test difference of abundance (left) and biomass (right) between size class X ecoregions. (This figure is only displayed in this answer but not shown in the revised version of the manuscript)*

[Figure]

**Minor comments**

**Comment R1.4**
Title Add "deposition" to the title ("dust deposition events"). ☑

*Answer to comment R1.4*
*We agree to change the title following the referee's suggestion and according to the new results in the manuscript .*
*We propose this new title:* "Structure and functioning of epipelagic mesozooplankton and response to dust deposition events during the spring PEACETIME cruise in the Mediterranean Sea."

**Comment R1.5**
Abstract L27: it is hard to understand the meaning of "long station". I suggest changing to "long-duration sampling station". ☑

*Answer to comment R1.5*
*We agree to change the term 'long station' to "long-duration sampling station" following the referee's suggestion.*
*We have rewritten these sentences as follows:*

L27 "Whereas in the Algerian basin (long-duration station FAST)"
L70 "the short-duration stations ST1 to ST9, and the long-duration station TYR"
L71 "whereas two long-duration stations ION and FAST, lasting 3 and 5 days respectively"
L86 "night tows were also performed for the long-duration stations FAST and ION"
L225 At the long-duration stations FAST and ION, strong variations in slope

**Comment R1.6**
Introduction L41: For P limitation the following paper may be cited:
Thingstad, T.F. and Rassoulzadegan, F., 1995. Nutrient limitations, microbial food webs and 'biological C-pumps': suggested interactions in a P-limited Mediterranean. Marine Ecology Progress Series, pp.299-306.

*Answer to comment R1.6*
*We agree to add the citation following the referee's suggestion, considering that this is a specific paper rather than a review.*

**Comment R1.7**
Materials and Methods L85: state whether the Bongo net was towed vertically or in oblique mode

*Answer to comment R1.7*
*We agree to specify how the net was towed following the referee's suggestion.*
*We have rewritten this sentence as follows*: "The Bongo frame was vertically towed from 300 m depth to the surface at a constant speed of $1ms^{-1}$".

**Comment R1.8**
Materials and Methods L83-85: was flow meter used to quantify the filtered volume? Or was it calculated based on net dimensions?

*Answer to comment R1.8*

*We agree to specify how we quantify the filtered volume, following the referee's suggestion. The volume was calculated based on the net ring diameter and the length of the towed cable.*
*We have rewritten this sentence as follows:*
"Sample volume was estimated based on the ring diameter and the towed cable length".

**Comment R1.9**
Materials and Methods L83-87: clarify why was the entire epipelagic depth strata (0-300m) chosen for sampling. It can be hypothesized that dust deposition will mostly affect the uppermost strata that is more affected by atmospheric deposition, thus sampling all along 0-300m may mask such possible effect. Nevertheless, diel migration and fecal pellet deposition may affect the whole epipelagic community.

*Answer to comment R1.9: As the other referee raised a similar question (see comment R2.6), this answer is common to both.*

*During the PEACETIME cruise, there were no zooplankton specialists on board due to the high pressure of other tasks (Atmospheric and oceanic sampling, on-board mesocosm studies).*
*The zooplankton net sampling strategy had to be defined before the cruise and the time devoted to zooplankton sampling was short, only for one tow between CTD casts.*
*Under these conditions, the best compromise based on previous studies (See table 3 in the manuscript) in these regions was to sample in the epipelagic water column. This depth was chosen to be sure that the zooplankton community in the epipelagic layer was collected.*
*Considering that bad meteorological conditions could affect the verticality of the net, using a 200 m cable length could not sample the whole epipelagic layer.*
*Also note that the observed impact on zooplankton is more significant because it integrates the whole water column.*

Comment R1.10
Materials and Methods L89: which model of FlowCAM was used? Were the flow cells of type FOV (field of view)?  Non-FOV flow cells are not efficient for quantitative measurements, and are best used for qualitative-only assessment of plankton as the transport of particles via the flow cell is not constrained (see Detmer et al 2019).
Detmer, T.M., Broadway, K.J., Potter, C.G., Collins, S.F., Parkos, J.J. and Wahl, D.H., 2019. Comparison of microscopy to a semi-automated method (FlowCAM®) for characterization of individual-, population-, and community-level measurements of zooplankton. Hydrobiologia, 838(1), pp.99-110.

*Answer to comment R1.10*
*We agree to add the FlowCAM model to the manuscript and to specify that we use a FOV flow cell. We consider that it is a necessary addition to the manuscript.*

*We propose these new sentences:*
-"The samples were processed using FlowCAM® (Fluid Imaging Technologies Inc. Series VS-IV, Benchtop model)"
-"For the fraction $N_{100F<200}$, a 4X magnification and 300 µm FOV flow cell were used and the analysis was carried out up to 3000 counted particles. For the fraction $N_{100F200-1000}$ a 2X magnification and 800 µm FOV flow cell were used and the analysis was carried out up to 1500 counted particles

Comment R1.11
Materials and Methods L128: detail ZooProcess version, add citation.

*Answer to comment R1.11*
*We agree to change paragraph following the referee's suggestion. We propose this new sentence:*
"After scanning, the  images were processed with ZooProcess (version 7.32) using  the  image analysis  software  Image  J (Grosjean  et  al.,2004; Gorsky et al., 2010)"

**Comment R1.12**
Materials and Methods L128-129: which software was used for automatic classification of vignettes?

Answer to comment R1.12
We agree to mention the software used for the vignettes classification, and we propose this new sentence: "the Plankton Identifier software (http://www.obs-vlfr.fr/~gaspari/Plankton_Identifier /index.php,  last  access: November 2019) was used for automatic classification of zooplankton".

**Comment R1.13**
Materials and Methods L135: ESD size categories 0.2 to 2.0 µm does not make sense (likely mm). Also, results show also 0-100 _m and 100-200 µm categories (Fig. 2).

*Answer to comment R1.13*
*We have noted the typing mistake in the sentence: µm will be changed to mm. Thesentence is rewritten as:* "The data were classified in size categories of 0.1 mm of ESD from 0.2 to 2.0 mm".

**Comment R1.14**
Materials and Methods L155-156: shortly describe the model for assessment of oxygen consumption, and excretion of NH3 and PO4

*Answer to comment R1.14: As the other referee raised a similar question (see comment R.2.28), this answer is common to both.*

*We agree to better describe the model used following the referee's suggestion. To do that, we now show the explanatory equation:*
*We propose this new paragraph*:

"Ammonium and phosphorus excretion and oxygen consumption rates were estimated using the multiple regression model by Ikeda et al. (1985) with carbon body weight and temperature as independent variables.

$$\ln Y = a_0 + a_1 \ln X_1 + a_2 X_2$$

Where ln*Y* represent the ammonium excretion, phosphorus excretion or oxygen consumption. $a_0$, $a_1$ and $a_2$ are constant (see Ikeda et al. 1985), $X_1$ is the body mass (dry weight, carbon, nitrogen or phosphorus weight) and $X_2$ is the habitat temperature (°C)".
* * *
**Comment R1.15**
Materials and Methods L164: add citation for PRIMER software.
* * *
*Answer to comment R1.15*
*We agree that this citation is necessary.*
*The added reference will be:* "Anderson M.J., Gorley R.N. & Clarke K.R. 2008. PERMANOVA+ for PRIMER: Guide to Software and Statistical Methods. PRIMER-E: Plymouth, UK."
* * *
Comment R1.16A
Materials and Methods L161-177: multivariate analysis was done separately for environmental data (PCA) and biological data (nMDS). These datasets should be correlated using BIOENV/BVSTEP or RELATE procedures.
Comment R1.16B
In addition, PERMANOVA can be used to statistically test the differences between basins.
Comment R1.16C
Also, the contribution of certain species for the dis/similarities between stations and basins should be statistically explored using SIMPER.
* * *
*Answer to comment R1.16A: See answer to major comment R1.3B*
*Answer to comment R1.16B: See answer to major comment R1.3C*
*Answer to comment R1.16C: See answer to major comment R1.3C*
* * *
Comment R1.17
Materials and Methods L168-169: RFD analysis here is only descriptive.
To test differences between RFDs, RAD analysis (max rank normalization method) can be used: Saeedghalati, M., Farahpour, F., Budeus, B., Lange, A., Westendorf, A.M., Seifert, M., Küppers, R. and Hoffmann, D., 2017. Quantitative comparison of abundance structures of generalized communities: from B-cell receptor repertoires to microbiomes. PLoS computational biology, 13(1), p.e1005362.
* * *
*Answer to comment R1.17*
*We agree that our previous description was mainly descriptive and following the referee's suggestion, we propose this new text and figures on the RFD comparison to be added in the manuscript.*

*In section 2.6 of the manuscript, we have added this new paragraph:*
"In order to improve the interpretation of the RFDs, first we used a method derived from Saeedghalati et al. (2017) based on the ordination of normalized rank abundance distribution. Rank-abundance matrix was created with the data standardized by the total abundance. Resemblance was

measured with Bray-Curtis similarity and a cluster was created using the complete linkage criterion. Secondly, a rank abundance distribution index was estimated following Mouillot and Lepretre (2000). The RFD for each station was separated into three portions: first the ranks with relative abundance <0.5 % were discarded (rare taxa, between 0 and 30% of the taxa according to all stations; by taking <1% we would discard between 18 and 49% of the taxa) and then the two parts were fitted with a linear regressions. One part with 4 highest ranks (see Mouillot and Lepretre for the justification) and the remaining portion with the following ranks (between 15 and 23 taxa, depending on the station). The slope for both upper and lower RFD portion was calculated (p1 and p2 respectively), then the p1/p2 ratios were estimated to quantify the differences between the RFDs of all the stations."

*In section 4.4 of the manuscript, we have added this new paragraph:*

The cluster analysis on the RFDs (Figure 11A)  is in agreement with this succession of the time series (Figure 10F) by grouping the stations according to impact level of the wet dust deposition. It separates the initial condition (FAST1) from the most disturbed state (stations FAST3 and ST6) and identifies a transition phase before (FAST2) and after (TYR and ST6) the peak disturbance.

The changing trends in p1/p2 ratios (Figure 11B) show an interesting development, with a sharp increase until day 5 after the dust deposition and a progressive decrease towards the end of the virtual time series. The linear regression suggests that the community structure will deliver a p1/p2 ratio value similar to the initial value of the time series after 22 days. Is interesting to note that this delay corresponds to an average generation time of zooplankton organisms for this region.

[Figure]

New figure 11: Cluster analysis on rank frequency diagrams (A) and changing trends in the p1/p2 ratio (B) on the stations impacted by wet dust deposition.

*In the revised version of the manuscript, we have focused only on the information concerning the virtual time series (see paragraph 3.4 Zooplankton community changes linked to dust deposition events during the PEACETIME survey and figures 10 and 11). But below we present the whole analysis available for the referee and any reader of the comments.*

*Figure: Rank frequency diagram for all stations of the PEACETIME cruise. Ac: Acartia spp.; Cal: Calanoid copepods; Cala: Calanus spp.; Cent: Centropages spp.; Cor: Corycaeus spp.; Euc: Eucalanus spp.; Halop: Haloptilusspp; Luci: Lucicutia spp.; Mecy: Mecynocera spp.; On: Oncaea spp.; Ot: Oithona spp.; P/Cla: Para/Clausocalanus spp.; Pleu: Pleuromamma spp.; Pont:Pontellidae; Tem: Temora spp.; App: Appendicularia; Cha: Chaetognatha; Dec: Decapods; Hydro: Hydrozoans; Ich: Ichtyoplankton; Ost: Ostracods; Poly: Polychaeta; Pte: Pteropods; Siph: Siphonophores; Thal: Thaliaceans. (This figure is only displayed in this answer but not shown in the revised version of the manuscript)*

[Figure]

*Figure: Cluster analysis on rank frequency diagrams of all the stations in the PEACETIME survey. (This figure is only displayed in this answer but not shown in the revised version of the manuscript)*

[Figure]

*Figure: Changing trends in the p1/p2 ratio of all the stations in the PEACETIME survey. (This figure is only displayed in this answer but not shown in the revised version of the manuscript)*

[Figure]

Comment R1.18
L168-171: it would be more convenient to summarize all Spearman rank correlations and t-tests in two tables. I could not find t-test results except one testing the difference between N100 and N200 mesh (L107).

*Answer to comment R1.18*
*Following the referee suggestion we propose this new table to be added on the manuscript.*

New Table 3. Summary table of Spearman rank correlations. T°= temperature; Sal= salinity; Chl-a= Chlorophyll; MLD= Mix layer depth; pp= primary production. Bold characters indicate significant Rs value (p<0.05)

| | Correlation coeff. | | | | | |
|---|---|---|---|---|---|---|
| Abundance | T° | Sal | Chl-a | DO | MLD | PP |
| $C_{<200}$ | -0,49 | -0,43 | 0,32 | **-0,61** | -0,16 | -0,37 |
| $C_{200-300}$ | **-0,58** | -0,37 | 0,48 | **-0,58** | 0,08 | -0,24 |
| $C_{300-500}$ | -0,51 | -0,19 | **0,52** | -0,45 | 0,21 | -2,28 |
| $C_{500-1000}$ | **-0,56** | -0,50 | 0,23 | -0,49 | -0,06 | 0,05 |
| $C_{1000-2000}$ | 0,29 | 0,01 | -0,28 | 0,33 | -0,34 | 0,35 |
| $C_{>2000}$ | **-0,12** | **-0,53** | -0,15 | 0,08 | -0,50 | -0,16 |
| Total abundance | **-0,67** | -0,44 | **0,56** | **-0,68** | 0,08 | -0,28 |
| Biomass | | | | | | |
| $C_{<200}$ | **-0,61** | -0,48 | 0,42 | **-0,71** | -0,08 | -0,36 |
| $C_{200-300}$ | **-0,52** | -0,29 | 0,52 | -0,51 | 0,12 | -0,14 |
| $C_{300-500}$ | -0,49 | -0,18 | **-0,53** | -0,46 | 0,19 | -0,27 |
| $C_{500-1000}$ | -0,45 | -0,43 | 0,17 | -0,41 | -0,11 | 0,14 |
| $C_{1000-2000}$ | 0,24 | -0,05 | -0,37 | 0,32 | -0,39 | 0,30 |
| $C_{>2000}$ | -0,18 | **-0,61** | -0,10 | -0,02 | **-0,53** | -0,10 |
| total biomass | **-0,58** | **-0,62** | 0,24 | -0,43 | -0,27 | 0,08 |

*T-test was changed for ANOVA and the table is now included. See answer to comment R1.3D*

Comment R1.19
Results L176-178: "mostly influenced by: how was this determined? Visually? This can be tested statistically using correlation between the environmental factors and the 1st axis (or also the 2nd axis) of the PCA.

*Answer to comment R1.19*
*We agree with the referee's suggestion, and have performed a correlation analysis between the environmental variables and the scores of the stations on the two axes of the PCA. We propose the following sentence.*
"The first axis (62 % of the variance) is mostly influenced by temperature and dissolved oxygen, as shown by their high correlations with the scores of the sampling points on this axis (r= 0.95 with p=0.000 and r=0.92 with p=0.000, respectively), whereas the second axis (28.3 %) is mostly influenced by MLD (r=-0.75, p=0.01), salinity (r=-0.75, p=0.001) and Chl-a (r=-0.57, p=0.022)."

*Table. Correlations between environmental variables and the axes of the PCA. (This table is only displayed in this answer but not shown in the revised version of the manuscript)*

|  | X1 | X2 |
|---|---|---|
| Temperature | .9564 | -.2174 |
|  | p=.000 | p=.419 |
| Salinity | .6294 | -.7476 |
|  | p=.009 | p=.001 |
| Dissolved Oxigen | .9213 | -.1154 |
|  | p=.000 | p=.670 |
| Chlorophyll-a | -.7572 | -.5680 |
|  | p=.001 | p=.022 |
| MLD | -.3880 | -.7460 |
|  | p=.137 | p=.001 |

Comment R1.20
Results L179-184: also here seems that non-statistical assessment was done to relate biological data to environmental parameters. See above comment on the use of statistical procedures (BIOENV, RELATE, PERMANOVA).

*Answer to comment R1.20: See answer to Comment R1.3C*

Results Comment R1.21
L186-L261 and throughout the text: abundance of zooplankton is presented in ind.m-2 instead of ind.m-3. Similarly biomass, mgDW.m-2 instead of mgDW.m-3.

*Answer to comment R1.21*
*Throughout the text, the data was presented in $m^{-2}$, because we estimate metabolic rates in the integrated water column. In the comparative table (New Table 7), the data was presented in $m^{-3}$ to be able to compare them with published data.*

Comment R1.22
Results L188-198: no statistics! See above comment re ANOVA.

*Answer to comment R1.22: See answer to comment R1.3D*

Comment R1.23
Results L199: "C300-500 biomass is positively correlated with Chl-a (r=-0.52, p=0.042)". r is negative so it is negatively and not positively correlated.

*Answer to comment R1.23*
*Following the referee's suggestion, we noted the typing error. We propose this new sentence:*
"$C_{300-500}$ biomass is negatively correlated with Chl-a (r=-0.52, p= 0.042)".

Comment R1.24
Results L211-223: missing statistics – see comment above re SIMPER.

*Answer to comment R1.24: See Answer comment R1.3C*

Comment R1.25
Results L229: change sub-header to "zooplankton community changes at long-duration sampling stations".

*Answer to comment R1.25 : In the new version of the manuscript, we have changed the name of this section to:*
3.4 Zooplankton community changes linked to dust deposition events during the PEACETIME survey

Comment R1.26
Discussion L265-278: the methodological concerns are presented as a major outcome of this study although they are mostly a by-product. Nevertheless, they provide important conclusions. I would reorder the sections, and put this as the second or third section.

*Answer to comment R1.26*
*We agree that this methodological aspect is a by-product of the paper, but we think that it is better for the reader to be aware early in the Discussion of the argumentation about the adequacy (and limitations) of our sampling and analytical strategy. Consequently we prefer to keep this paragraph at the beginning of the Discussion section.*

Comment R1.27
Discussion L293-310: the authors compare the zooplankton abundances and biomass measured in this study (372000 ind m-3 and 1707 mgDW m-3 respectively reported in L186-187 and in figure 4) to other studies (table 3). Unlike their statement that the results of this study are in the same order of magnitude as previous studies – it seems that the changes are enormous! For example, Donoso et al 2017 measured 608 ind m-3 and 64 mgDW m-3. This is a change of 3 orders of magnitude in abundance and 2 in biomass! Note that the reported values are not the same in figure 4 and table 3. Please explain these differences.

*Answer to comment R1.27*
*Throughout the text, all estimations (abundance, biomass and metabolic rates) are in m$^{-2}$ to express those values as integrated in the water column. In table 3 the abundance and biomass from other*

*papers were expressed in m$^{-3}$ in order to use the same values as they appear in the respective publications.*

*Tables 3 and Figure 4 had the same values, the differences are in the unit and also because in table 3 the abundances and biomass were expressed as mean values ± Standard Deviation of the respective basin*
* * *
Comment R1.28

Discussion L293-310: additional comparison should be performed to Mediterranean studies that used 100 _m mesh in addition to a larger mesh size and measured abundance and biomass. For example: Koppelmann, R., Böttger-Schnack, R., Möbius, J. and Weikert, H., 2009. Trophic relationships of zooplankton in the eastern Mediterranean based on stable isotope measurements. Journal of Plankton Research, 31(6), pp.669-686.
* * *
*Answer comment R1.28*

*All the papers cited for comparison use samples from areas close to the PEACETIME track, there are more studies in Mediterranean Sea that use 100 μm nets but since they were sampled in areas that are far away, those data are not comparable with our study*
* * *
Comment R1.29

Discussion L389-410: phytoplankton biomass and production assessment were calculated from Chl-a rather than being measured directly. Similarly, zooplankton carbon demand, oxygen consumption and excretion was based on multiple assumptions, including the use of constant biomass to carbon conversion factor, carbon to Chl-a ratio, Redfield ratio and respiratory quotient.

While these are all legitimate methods, it must be discussed that over large geographical scales that include environmental gradient – these factors may vary and thus these assessments may be inaccurate.

For example, Minutoli Guglielmo 2009 showed an increasing trend in ETS activity from west to east: Minutoli, R. and Guglielmo, L., 2009. Zooplankton respiratory Electron Transport System (ETS) activity in the Mediterranean Sea: spatial and diel variability. Marine Ecology Progress Series, 381, pp.199-211.
* * *
Answer to comment R1.29

*We agree with the referee's comment and we propose to add this new sentence in section 4.3:*

"By using allometric relationships relating zooplankton grazing and metabolic rates to size structure, zooplankton impacts (top-down vs. bottom-up) on primary production have been investigated. We are aware that using constant conversion factors may limit the analysis of the spatial variation, since these factors may display temporal and geographical variations (Minutoli and Guglielmo, 2009). However, our sampling strategy based on a limited number of stations sampled did not enable us to consider temporal and spatial variations accurately and our main goal was to have rough estimations of the epipelagic zooplankton mediated fluxes at the scale of the PEACETIME cruise."
* * *
comment R1.30

Figures and Tables. Fig. 2: in the Zooscan and FlowCAM analyses, there is a small overlap in the fractions 300-400 and 400-500. Nevertheless, it seems that the combined data (fig 2c) include these fractions from the flowCAM only. Shouldn't these fractions be accumulative?
* * *
*Answer to comment R1.30:*

*Classes 300-400 and 400-500 µm should not be cumulative because it is the same part of the sample analyzed twice. If we accumulate them, we will count the same size class twice.*
* * *
Comment R1.31
Figures and Tables. Fig. 3: show the % of total variance in the labels of each of the axis (PCs). If possible overly the environmental vectors (fig 3B) on the PCA (fig 3A) instead of showing them in two different images.
* * *
*Answer comment R1.31:*
*We agree and the correction will be done in the figure.*
* * *
Comment R1.32
Figures and Tables. Fig. 4: correct Chla-a to Chl-a
* * *
*Answer comment R1.32:*
*We agree and the correction will be done in the figure.*
* * *
Comment R1.33
Figures and Tables. Table 3: is biomass presented in all mentioned studies as dry weight?
* * *
*Answer to comment R1.33: Throughout the manuscript, we use the term dry weight, but in table 3 one of the cited papers publishes the data in wet weight.*
*In table 3, all values are expressed in dry weight except one marked with ***. We will mention this in the legend of the table in the revised version.*

\*\*\*\*\*\*\*\*\*\*\*\*\*\*\*\*\*\*\*\*\*\*\*\*\*\*\*\*\*\*\*\*\*\*\*\*\*\*\*\*\*\*\*\*\*\*\*\*\*\*\*\*\*\*\*\*\*\*\*\*\*\*\*\*\*\*\*\*\*\*\*\*\*\*\*\*\*\*\*\*\*\*\*\*\*

---

## Author Comment (AC2) · 24 Jul 2020

*Note: All the frames contain the comments of referee R2, and have been identified by a number. Our answers are below each frame, and the underlined paragraphs in yellow correspond to new or modified paragraphs in the manuscript. R1 refers to answers common to comments by referee 1.*

Major Comment 2.1
The manuscript presents a complete set of multidisciplinary data of zooplankton community structure and functioning from an oceanographic cruise in the Western Mediterranean Sea in late spring-early summer (May to June 2017) during two major dust events in the Algerian and Tyrrhenian basins. Investigating mesozooplankton structure in the western Mediterranean Sea is a classic but necessary marine science research approach to improve our knowledge on the mesozooplankton community and estimate the responses of this key trophic group in the pelagic ecosystem linking small primary and secondary producers to higher nektonic trophic levels. Furthermore this study significantly contributes on the effect of the Saharan dust deposition on the zooplankton community structure. Several papers have been produced in the scientific world and at the Mediterranean Sea level, dealing with mesozooplankton composition, distribution and structure. Nevertheless, this study pushes the analysis deeper up to mesozooplankton functioning and estimate zooplankton growth, ingestion and metabolism using allometric relationships. This paper is very well written and organized, but

*We have tried to answer all your comments and have substantially rewritten several paragraphs of the manuscript. We appreciate all your comments and we acknowledge that they have encouraged us to explore further our data. We hope that now the revised version of the manuscript shows more consistency between the main aims and the presented results.*

(Comment 2.1A) a clear scientific question or at least two hypotheses should appear in the introduction instead of a description of the objectives as general scientific tasks. For instance what do you expect during the Sahara dust deposition, what should be the effect of the Sahara dust to the zooplankton community and how your data can show this influence.

*Answer to comment 2.1 A.*
*As the other referee raised a similar question (see comment R1.3A), this answer is common to both.*

*We agree with the referee about this comment which has been very stimulating in helping us to redefine various aspects of the focus of the paper.*

*We have now stated these general hypotheses:*
***H1: Is zooplankton structure impacted by dust deposition?.*** *We hypothesize that Saharan dust deposition events had an impact on the zooplankton community in the Mediterranean Sea, modifying its abundance, biomass, metabolic rates (no observed changes except in in $C_{500-1000}$) and diversity (The changes we observed are mostly related to community structure and taxonomic diversity (RFD) and to size distribution) .*

***H2: Hydrodynamical regions versus dust impact region***

*- It is difficult to relate zooplankton response to regional differences (lack of stations). Response is due to dust deposition and mainly affect the species diversity (RFD) and size structure (p1/p2 ratio)*

***H3 -*** *How long should we observe the zooplankton structure in a given area to clearly observe the dust impact (p1/p2 ratio)?.*

*The results providing a basis to validate these hypotheses have been analyzed following several statistical treatments as detailed in the following answers. (See below comment R1.2B, R1.2C, R1.2D)*

*The following paragraph has been added at the end of the Introduction:*
"These objectives will serve to test the following hypotheses: whether the Saharan dust events impact the zooplankton community structure following deposition (H1), and if so, whether the effect would be immediately observable or after a lag time (H2). Finally, whether changes in zooplankton structure driven by dust deposition exceed regional differences under oligothropic conditions (H3)."
* * *
Major Comment 2.1 (continue)
(Comment 2.1B) Also, the major part of the discussion is mainly based on regional differences and comparisons and less discussion has been made on the effect of Sahara dust on zooplankton estimated vital rates. Are there any differences of the vital rates before and after the Sahara dust events? What is the response of the zooplankton community after the Sahara dust deposition besides the changes in community structure?
* * *
*Answer to Comment 2.1B*
*Concerning the potential changes in metabolic rates, our calculation based on empirical models did not show increased metabolic rates which could be linked to the dust events. Planned dedicated measurements should have been implemented to observe potential changes in metabolic rates.*
*With regard to the response of the zooplankton community after the Sahara dust deposition, we have developed in the revised version a more complete analysis of the RFD changes following the dust events.*
*This is detailed in the answer to the referee 1 (see answer to comment R1.17).*

**Please find below the detailed comments on the manuscript.**
* * *
Comment R2.2
Introduction Line 47: Add also the reference Siokou et al., 2019.( Deep-Sea Research Part II 164 (2019) 170–189).
* * *
*Answer to comment R2.2*
*we agree with the referee's suggestion and the citation will be added. We propose this new sentence:*
"... a succession of oceanographic surveys covering wide transects at different time periods of the year (Kimor and Wood, 1975; Nowaczyk et al., 2011; Donoso et al., 2017; Siokou et al., 2019).

New reference
Siokou, I., Zervoudaki, S., Velaoras, D., Theocharis, A., Christou, E. D., Protopapa, M. and Pantazi, M.: Mesozooplankton vertical patterns along an east-west transect in the oligotrophic Mediterranean sea during early summer, Deep. Res. Part II Top. Stud. Oceanogr., 164, 170–189, doi:10.1016/j.dsr2.2019.02.006, 2019.

Comment R2.3
Introduction Line 58: Explain better why this cruise should be "flexible". The manuscript Guieu et al., 2020 has not been published so it is difficult to understand the design of the oceanographic cruise.

*Answer to comment R2.3*

*To address this comment, we have selected some paragraphs from the work done by Guieu et al. (2020) where the flexibility of the cruise is explained. This is now explained in the new table 2 in the manuscript.*

*"Based on the experience of the ChArMEx airborne campaigns (Mallet et al., 2016) and of previous oceanographic cruises needing an adaptive planning strategy based on observations and short-term forecasts (see section "Satellite monitoring of the ocean"), an operational server named the PEACETIME Operation Center (POC; http://poc.sedoo.fr/; last access 9 Feb. 2020) was set-up by the Service de Données de l'Observatoire Midi-Pyrénées (OMP/SEDOO, Toulouse, France) for the cruise. Guieu et al., 2020*

*"The actual positions of stations were discussed and determined on the basis of near-real time satellite data analysis (SPASSO) in order to account for local oceanic conditions (i.e. presence or not of mesoscale structures). In parallel, short- and middle-term forecast models of weather conditions and of dust transport and deposition were systematically analyzed to verify the conditions, and eventually start the Fast Action. The Fast Action strategy consisted in routing the ship towards an area of forecasted dust deposition event in order to tentatively document the respective roles of dynamics and deposition on marine biogeochemical conditions. The goal was to position the ship in the center of the area of dust deposition, at least one day (24 hours) before the event in order to sample the water column before, during and after the deposition, and collect and characterize the rain event. Several constraints had to be considered for the Fast Action decision." (Guieu et al., 2020)*

*"All these elements were simultaneously analyzed during a daily meeting between scientists involved on land and on ship, as well as with the crew. Each day, the initial plan was confirmed for the next 48 h or, eventually, modified" (Guieu et al., 2020).*

Comment R2.4
Line 60: Add also the main questions and hypotheses of this study.

*See Answer to comment 2.1A*

Comment R2.5
Materials and Methods Line 80: Explain why did you calculate the depth of the Mixed Layer. In this study there is no information at all about the hydrology of the area so it is difficult to follow. Only at the end in the discussion chapter the authors clarified the hydrological features existing in the area.

*Answer to comment R2.5*
*Mixed layer depth is an important parameter that defines quasi-homogenous regions of the ocean (Swain et al. 2006). We think that this provides valuable information to determine differences in the hydrology of the basins.*
*In previous works, MLD was also used (among other hydrological and trophic parameters) to define the habitat of epipelagic zooplankton (Donoso et al 2017, GJR 122): Illustrating physical separation between highly stratified/nutrient-poor and well-mixed/nutrient-rich areas.*
*To add more detail on the hydrological features of the area, we cite a key reference in the Discussion (Millot and Taupier-Letage. (2005). See answer to comment R2.14*

*Reference:*

*Donoso, K., Carlotti, F., Pagano, M., Hunt, B. P. V., Escribano, R. and Berline, L.: Zooplankton community response to the winter 2013 deep convection process in the NW Mediterranean Sea, J. Geophys. Res. Ocean., 122(3), 2319–2338, doi:10.1002/2016JC012176, 2017.*

*Millot, C., Taupier-Letage, I., 2005. Circulation in the Mediterranean Sea. In: (Ed.), The Handbook of Environ-mental Chemistry, vol. 1. The Natural Environment and the Biological Cycles, Springer, pp. 29-66.*

*Swain, D., Ali, M. M. and Weller, R. A.: Estimation of mixed-layer depth from surface parameters, J. Mar. Res., 64(5), 745–758, doi:10.1357/002224006779367285, 2006.*
* * *
**Comment R2.6**

Materials and Methods Lines 85-86: Explain why do you perform. the zooplankton sampling from the surface until 300 m? Is it due to the euphotic zone? Is it due to the hydrological features of the area?
* * *
*Answer to comment R2.6: As the other referee raised a similar question (see comment R.1.9) this answer is common to both.*

*During the PEACETIME cruise there was no zooplankton specialist on board due to the high pressure of other tasks (Atmospheric and oceanic sampling, on board mesocosm studies).*

*The zooplankton net sampling strategy had to be defined before the cruise and the time devoted to zooplankton sampling was short, only for one tow between CTD casts.*

*Under these conditions, the best compromise based on previous studies (See table 3 in the manuscript) in these regions was to sample in the epipelagic water column.*

*This depth was chosen to be sure that the zooplankton community in the epipelagic layer was collected.*

*Considering that bad meteorological conditions could affect the verticality of the net, using a 200 m cable length could not sample the whole epipelagic layer.*

*Also note that the observed impact on zooplankton is more significant because it integrates the whole water column*
* * *
**Comment R2.7A**

Materials and Methods Line 140: Explain what negative and positive values of the NBSS slopes means.
* * *
*Answer to comment R2.7A*

*We agree with the referee's suggestion and to explain the values of the NBSS slope, we propose this new sentence in section 2.4:*

"The slope of the NBSS reflects the balance between small and large individuals, a steeper slope corresponding to a higher proportion of small individuals (bottom-up control) and a flatter slope corresponding to a higher proportion of large individuals (top down control) (Donoso et al., 2017; Naito et al., 2019)".
* * *
**Comment R2.7B**

Materials and Methods Line 153: Explain why did you use the conversion factor of C:Chla=50 and add the reference. For oligorophic waters are more suitable to use the conversion factor/equation of Malone et al 1993, which is used the different Chl-a values according to depth.
* * *
*The ratio C:Chla varies between 20 and 100 (Malone et al 2013; Marañón 2005) and 50 is often taken as an average value (Romero et al., 2011; Gomez et al. 2015).*

*Many factors influence the value of C:Chla ratio (light, depth, nutrients status community composition, etc., and finally regions and seasons), resulting in lower values than 50 in the productive zone or productive seasons, and higher values in oligothropic conditions (Malone et al 2013; Marañón et al., 2015).*

*Concerning the Mediterranean sea, values of C:Chla ratio in the Eastern basin are usually around 60 and above (Lagaria et al, 2016), and in the Western basin they vary between 20 and 50 (Delgado et al 1992; Van Wambeke et al., 2002). Several works in the Med. Sea also used the C:Chla ratio of 50 as conversion factor (e.g. Christou., et al 2017).*

*The equation of Malone et al. (Malone et al., 1993) is an empirical formulation adapted from observed data in the Sargasso sea, we have no available data to calibrate it for the PEACETIME stations. Moreover we mainly need to have a rough estimation of zooplankton carbon demand in the water column.*

*References.*

*Christou, E. D., Zervoudaki, S., Fernandez De Puelles, M. L., Protopapa, M., Varkitzi, I., Pitta, P., Tsagaraki, T. M. and Herut, B.: Response of the Calanoid Copepod Clausocalanus furcatus, to Atmospheric Deposition Events: Outcomes from a Mesocosm Study, Front. Mar. Sci., 4, 35, doi:10.3389/fmars.2017.00035, 2017.*

*Gomes, A., Gasol, J. M., Estrada, M., Franco-Vidal, L., Díaz-Pérez, L., Ferrera, I. and Morán, X. A. G.: Heterotrophic bacterial responses to the winter-spring phytoplankton bloom in open waters of the NW Mediterranean, Deep. Res. Part I Oceanogr. Res. Pap., 96, 59–68, doi:10.1016/j.dsr.2014.11.007, 2015.*

*Lagaria, A., Mandalakis, M., Mara, P., Papageorgiou, N., Pitta, P., Tsiola, A., Kagiorgi, M. and Psarra, S.: Phytoplankton response to Saharan dust depositions in the Eastern Mediterranean Sea: A mesocosm study, Front. Mar. Sci., 3(JAN), 287, doi:10.3389/FMARS.2016.00287, 2017.*

*Malone, T., Pike, S. E. and Conley, D. J.: Transient variations in phytoplankton productivity at the JGOFS Bermuda time series station, Deep. Res. Part I, 40(5), 903–924, doi:10.1016/0967-0637(93)90080-M, 1993.*

*Marañón,E. PhytoplantongrowthratesintheAtlanticsubtropicalgyres. Limnol. Oceanogr. 50,299–310.doi:10.4319/lo.2005.50.1.0299. 2005.*

*Romero, E., Peters, F., Marrasé, C., Guadayol,  scar, Gasol, J. M. and Weinbauer, M. G.: Coastal Mediterranean plankton stimulation dynamics through a dust storm event: An experimental simulation, Estuar. Coast. Shelf Sci., 93(1), 27–39, doi:10.1016/j.ecss.2011.03.019, 2011.*

*Van Wambeke, F., Heussner, S., Diaz, F., Raimbault, P. and Conan, P.: Small-scale variability in the coupling/uncoupling of bacteria, phytoplankton and organic carbon fluxes along the continental margin of the Gulf of Lions, Northwestern Mediterranean Sea, J. Mar. Syst., 33–34, 411–429, doi:10.1016/S0924-7963(02)00069-6, 2002.*

Comment R2.8
Line 157: Add equation model for ammonium, phosphorus excretion and oxygen consumption rates as you did with the other relationships.

*Answer to comment R2.28: As the other referee raised a similar question (see comment R.1.14) this answer is common to both.*

*We agree to better describe the model used following the referee suggestion. To do that, we show the explanatory equation:*
*We propose this new paragraph:*

"Ammonium and phosphorus excretion and oxygen consumption rates were estimated using the multiple regression model by Ikeda et al. (1985) with carbon body weight and temperature as independent variables.

$$\ln Y = a_0 + a_1 \ln X_1 + a_2 X_2$$

Where ln*Y* represent the ammonium excretion, phosphorus excretion or oxygen consumption. $a_0$, $a_1$ and $a_2$ are constant (see Ikeda et al. 1985), $X_1$ is the body mass (dry weight, carbon, nitrogen or phosphorus weight) and $X_2$ is the habitat temperature (°C)".
* * *
Comment R2.9
Line 200: Throughout the text, there are several definitions for the small zooplankton, sometimes<1mm, <300, <500 _m. Please clarify in order to avoid any confusion.
* * *
*Answer to comment R2.9*
*We agree with the referee's suggestion and have defined small to zooplankton as the organisms < 500 µm*

Line 16: "with a noticeable contribution of the small-size fraction (< 500 µm) of up to 50 % in abundance and 25 % in biomass"

Line 202 "Abundance of zooplankton smaller than 300 µm is dominated by cyclopoid and calanoid copepodites"
(this sentence has been rewritten in order to talk about the fraction <300 µm but now we don't refer to it as "the small zooplankton size class")

Line 208 "The ratio between copepods with length smaller than 1 mm and larger than 1mm ranges from 2.8 to 8.3 (5.1 on average)"

Line 287 "the small size classes ($C_{200-300}$ and $C_{300-500}$) of mesozooplankton have been optimally sampled using a 100 µm mesh size net ($N_{100}$)"

Line 435: "In our study, this strategy also enabled us to show the importance of small forms (< 500 µm of ESD) both in terms of stocks and fluxes."
* * *
Comment R2.10
Line 221: you wrote "due to higher relative abundance of small copepods" please specify what species and which size. Also ostracods are not copepod species. Pontellidae family is written twice, and some species in Pontellidae are not small. Please specify if possible which species.
* * *
*Answer to comment R2.10*
*We agree with the comment of the referee and we have replaced the original sentence:*
*"This differentiation of ST7 and 8 from the ION sampling dates in the NMDS analysis is mainly due to higher relative abundance of small copepods (Figure 5), and specifically to several taxa such as*

*Mesocalanus spp. (more abundant), Pontellidae spp. and ostracoda (less abundant), Clytemnestra spp. (absent in ION) and Pontellidae spp. (absent atST7 and 8)"*

… by this new sentence:

"This differentiation of ST7 and 8 from the ION sampling dates in the NMDS analysis is mainly due to differences in relative abundance of *Mesocalanus* spp. (more abundant), ostracoda (less abundant), *Clytemnestra* spp. (absent in ION) and *Pontellidae* spp. (absent at ST7 and 8)".
* * *
Comment R2.11
Line 225-229. The explanation for the strong variations of the NBSS that is due to the migration of the larger species is not very clear, According to the Fig.4 the abundances of large species are quite similar between day and night sampling. Unless, you will approve that there is statistical significant differences between the night and day samples.
* * *
*Answer comment R2.11: Yes. that is true. But the NBSS was performed only on mesozooplankton. Zooplankton higher >2000 µm was removed in order to have a continuous set of data, so when we explain that "the variation of the NBSS that is due to the migration of the larger species" is larger within the mesozooplankton (200-2000 µm).*
*This information has now been added in the legend of the relevant figures (6 and 7).*
* * *
Comment R2.12
Line 234: Clausocalanus and Oithona species according to the literature are not herbivorous species.
Answer to comment R2.10
* * *
*Answer to comment R2.12*
*We agree with the referee's remark and propose these new sentences.*

Line 233. At all three TB stations, RFDs are characterized by high dominance of filter-feeding zooplankton Para/Clausocalanus spp. and Oithona spp. in 1st and 2nd position with a strong drop in abundance for the following ranked taxa.

Line 311 ST1 and ST2 are clearly differentiated from all others with deeper MLD, higher chlorophyll-a concentrations and a zooplankton community dominated by typical filter-feeding copepods of PB (Centropages, Para/Clausocalanus, Acartia, etc), as mentioned by Gaudy et al. (2003) and Donoso et al. (2017)
* * *
Comment R2.13
Line 253: Add in your Methods how or who did the Primary Production measurements.
* * *
*Answer to comment R2.13*
*We propose to add the following sentence in line 75.*

"The primary production was measured with the14C-uptake technique, following the methods detailed in (Marañón et al., 2000)".
* * *
Comment R2.14
Line 316:Specify the hydrological features of the area. A short chapter for the study area could be very helpful.
* * *
Answer to comment R2.14
*We agree to add a key reference which details all the hydrological features in the area.*
*We propose this new sentence:*

"AB and TB are very closely related to each other in terms of hydrological features and chlorophyll-a, but slightly differentiated in salinity and zooplankton taxonomy, probably because they are both strongly influenced by the Modified Atlantic Water (MAW) and its associated mesoscale features (Millot and Taupier-Letage., 2005)."

Reference: Millot, C., Taupier-Letage, I., 2005. Circulation in the Mediterranean Sea. In: (Ed.), The Handbook of Environ-mental Chemistry, vol. 1. The Natural Environment and the Biological Cycles, Springer, pp. 29-66
* * *
Comment R2.15
Line 370: In methodology you wrote that the contribution to nutrient regeneration by zooplankton was estimated using the values of primary production and converted to nitrogen and phosphorus requirement using Redfield ratio. However, the calculation doesn't follow the Redfield ratio of C:N:P = 106:16:1. Have you used this, or did you use the ratio that you found during the study? Please clarify.
* * *
*Answer to comment R2.15*

*We use Redfield ratio as used by Alcaraz et al. (2010) on their Table 5.*

*Reference: Alcaraz, M., Almeda, R., Calbet, A., Saiz, E., Duarte, C. M., Lasternas, S., Agustí, S., Santiago, R., Movilla, J. and Alonso, A.: The role of arctic zooplankton in biogeochemical cycles: Respiration and excretion of ammonia and phosphate during summer, Polar Biol., 33(12), 1719–1731, doi:10.1007/s00300-010-0789-9, 2010.*
* * *
Comment R2.16
Line 392: Delete "species composition" since no data is shown in Table 2.
* * *
*Answer to comment R2.16*
*We agree with the referee's suggestion and we propose this new sentence:*
"... an increase in primary production from FAST1 to FAST3, but with no visible changes in phytoplankton biomass (see Table 2)."
* * *
Comment R2.17
Line 410: Please add also the taxa that correspond to each stage of succession.
* * *
*Answer to comment R2.17 The RFD presented synoptically in figure 10F are the same as those presented in the previous scatted panels of figure 10, with the indication of the taxa.*
* * *
Comment R2.18
Lines 417-420: Explain how do you know that at the beginning you had small phytoplankton and then large? Because, according to Line 396 (pers. comm. J. Uitz), size and species composition of the phytoplankton community in FAST did not show any change after the dust.
* * *
*Answer to comment R2.18*
*We have now added more detail regarding the phytoplankton changes after the two dust events in the table dedicated to detail the information on the dust event (see comment R2.20). This table shows an increase of micro- and nano-plankton after the dust events.*

*Consequently the sentence in line 395-398 reads:*
"Size and species composition of the phytoplankton community in FAST did not show any change after the dust (pers. comm. J. Uitz), but there were potential increases in food competition with *Para/Clausocalanus* spp. (Lombard et al., 2010) and/or in predation by chaetognaths and siphonophores (Purcell et al., 2005)"

*... was changed by the new sentence:*

"Size and species composition of the phytoplankton community in FAST suggest a change toward larger cells (see supplementary table S1) poorly edible by appendicularians and inducing filter clogging. There were also potential increases in food competition with *Para/Clausocalanus* spp. (Lombard et al., 2010) and/or in predation by chaetognaths and siphonophores (Purcell et al., 2005)."

*Which is consistent with the content of Line 417 (unchanged): "State 1 before the dust event is characterized by oligothropic conditions with low nutrients, low phytoplankton concentration dominated by small size cells and their typical zooplankton grazers (e.g. appendicularians and thaliaceans), leading to a convex RFD shape (like FAST1 Figure 9F) reflecting a mature community (sensu Frontier, 1976). State 2 is characterized by a nutrient input linked to the dust event stimulating larger phytoplankton cells and their herbivorous grazers (copepods) and attracting carnivorous migrants leading to a more concave RFD shape (like FAST3, ST5 and TYR Figure 9F) typical of a disturbed community (sensu Frontier, 1976)."*

Comment R2.19
Lines 410-425: This paragraph could be the main hypothesis of your study.

*Answer to comment R2.19: see answer to comment 2.1A.*

Comment R2.20
Table 1: Add information about the dust events to follow better in the text.

*Answer to comment R2.20: As the other referee raised a similar question (see comment R.1.2) this answer is common to both.*
*We produce now a full synopsis of the Saharan dust deposition events encountered during PEACETIME. A lot of information are now presented in accepted and submitted papers of the special issue including the introduction paper by Cecile Guieu.*
*We present this new information in an additional paragraph in the Methods section.*

Guieu et al., introductory paper) detailed how they used three regional dust transport models to identify major dust events during the PEACETIME cruise. Two major wet dust events occurred during the period (Table 2). The first concerned the whole southern Tyrrhenian basin, with predicted flux > 1g m-2 (Desboeufs et al. in prep.), and started on May 10, several days before the arrival of the vessel in this area. The dust event was confirmed by aluminium, iron and lithogenic Si measured in sediment traps at TYR with a lithogenic flux between 200 and 1000m of 150-200 mg m-2 (Bressac et al., in prep.). The second was located in the area between the Balearics and the Algerian coast and occurred from 3 to 5 June, with predicted flux of 0.5 g m-2 (Guieu et al., accepted) after the arrival of the vessel in this area (station FAST). The dust event was confirmed by on-board atmospheric dust deposition samples (Desboeufs, in preparation this special issue), water column observations (nutrients, trace metals) (Tovar-Sánchez et al. 2020) and tracers of dust deposition in sediment traps (pers. comm. C. Guieu). The highest aerosol mass concentrations (around 25 µg m-3) with the highest iron content (245 ng m-3) were measured at FAST between 1 and 5 June, and subsequently the highest trace metal concentrations in the surface micro-layer were measured on 4 June (Co: 773.6 pM; Cu: 20.1 nM; Fe: 1433.3 nM; and Pb: 1294.7 pM) (Tovar-Sánchez et al 2020). The chemical composition of rain samples at FAST confirmed wet deposition of dust reaching a total particulate flux of 0.012 g m-2 (Fu et al., in prep.). The Ionian basin was the only southern area not impacted by dust deposition during the PEACETIME cruise, and results obtained at the long-duration station ION will be used for comparison.

*More details about the total trace metal concentrations in the dusty rain collected by Fu et al., (in preparation) were presented in (Tovar-Sánchez et al 2020): Trace metals values ranged from 180pM for Cd to 343nM for Fe (Cd: 180 pM; Co: 1380 pM; Cu:18.1 nM; Fe: 343 nM; Ni: 9.9 nM; Mo: 875 pM; V: 26.9 nM; Zn: 345 nM; and Pb: 788 pM)" (Tovar-Sánchez et al 2020).*

*This information is summarized in the table below.*

New Table 2. Overview of the main characteristics of the wet dust events occurring during PEACETIME. Zooplankton sampling was carried out very close to a CTD cast except at FAST2 where the sampling was done between two casts respectively 9 hours after the first cast (a) and 16 hours before the second (b) .

| Stations impacted by dust and cruise visit duration | Crusie strategy with regard to dust events | Dates, geographical characteristics and intensity of the dust events predicted by the model and by observations | Zooplankton sampling Date | Iron in aerosol ng m$^{-3}$ | Nutrients below the nutricline NO3 (n mol/l) PO$_4$ (n mol/l) | Surface Primary production mg C m$^{-3}$ d$^{-1}$ | Water column (0-250) average Chl-a concentration mg m$^{-3}$ | Depth range of the DCM strata (m) | Mean concentration of Chl-a on DCM strata mg m$^{-3}$ | Ratio fluorecence phytoplankton $F_{micro}$:$F_{nano}$:$F_{pico}$ within the DCM strata |
|---|---|---|---|---|---|---|---|---|---|---|
| Wet dust event FAST 02 to 07 June | Station FAST schedule and position detemined on board acording to metereologial event | From 3 to 5 June; Impacted area: Betwen Baleares and Algerian coast; Predicted flux from models: 0.5 g m$^{-2}$ (Guieu et al., accepted, Supp Info figure SI5); On-board atmospheric dust deposition observations confirmed a weak wet dust deposition of 0.012 g m$^{-2}$ (Guieu et al., accepted). In sediment traps lithogenic flux was 40-60 mg m$^{-2}$ between 200 and 1000m. Water column observations (nutrients, trace metals) (van Wambeke et al., in prep, Tovar-Sánchez et al. 2020, Bressac et al., in prep) show a clear imprint of the atmospheric deposition. | FAST1: 04-06-2017 | 245.3 | 224 246 | 2.44 | 0,12 | 60-90 | 0,42 | 27:45:28 |
| | | | FAST2: 06-06-2017 | 266.0 | 808 239 | 2.85 | 0,14[(a)] 0,18 [(b)] | 60-100 [(a)] 70-90 [(b)] | 0,38[(a)] 0,86 [(b)] | 25:43:32 [(a)] 50:30:20 [(b)] |
| | | | FAST3: 08-06-2017 | 44.9 | 135 113 | 2.04 | 0,10 | 70- 90 | 0,42 | 20:49:31 |
| Wet dust event Tyrrhenian 16 to 22May | TB stations schedule before the cruise. Model predicted a dust event 6 days before the arrival | From 10 to 12 may; Impacted area: whole southern Tyrrhenian sea; Predicted flux from models: >1 g m$^{-2}$ (Desboeufs et al. in prep) Dust event was confirmed by alumimium, iron and Lithogenic Si measured in sediment tramps at TYR with a lithogenic flux between 200 and 1000m was 150-200 mg m$^{-2}$ (Bressac et al., in prep.) | ST5: 16-05-2017 | 57.3 | 841 148 | 1.68 | 0,12 | 70-80 | 0,55 | 21:48:30 |
| | | | TYR: 19-05-2017 | 162.3 | n.d 127 | 1.77 | 0,11 | 70-80 | 0,61 | 33:40:27 |
| | | | ST6: 22-05-2017 | 189.8 | 488 136 | 1.66 | 0,07 | 70-80 | 0,36 | 7:44:49 |
| References of the data | Dulac (pers.com) Desboeufs et al. (in prep) Guieu et al. (accepted) . | Desboeufs et al. (in prep) Guieu et al. (accepted) Bressac et al. (in prep) Tovar-Sánchez et al. (2020) van Wambeke et al. (in prep) | Tovar-Sánchez et al. (2020) | van Wambeke et al. (in prep) | | E. Maranon and M. Perez-Lorenzo | J.Uitz, C. Dimier | J.Uitz, C. Dimier | J.Uitz, C. Dimier | J.Uitz, C. Dimierl |

Comment R2.21A
Line340-375: In this chapter the authors are reported several times the different vital rates of the zooplankton size fractions. However, these data are not provided in Table2.
Comment R2.21B
It could be useful to add biomass data of the total zooplankton as well as of the different size fraction in the Table 2.
Table 2: Add biomass data of total zooplankton and size fractions.

*Answer to comment R2.21A.*
*We propose to put the value by size fractions as supplementary material. See supplementary table S1*
*Answer to comment R2.21B*
*Biomass per class is already shown in figure 4b and now is also available as supplementary material. See supplementary table S1*

Comment R2.22
Table 3: Add Siokou et al., 2019 and make the comparison

*Answer to comment R2.22*
*We agree with the referee's suggestion and propose this new addition to table 3*

| Area | Sampling period | Net mesh size (µm) | Layer (m) | Biomass (mg m$^{-3}$) | Abundance (ind m$^{-3}$) | Reference |
|---|---|---|---|---|---|---|
| NWMS - Provencal sea | Jul 1999 | 200 | 0-300 | | 383 | Siokou et al. (2019) |
| SWMS- Algerian sea | Jun 1999 | 200 | 0-300 | | 197 | Siokou et al. (2019) |
| Ionian sea | Jun 1999 | 200 | 0-300 | | 146 | Siokou et al. (2019) |

\*\*\*\*\*\*\*\*\*\*\*\*\*\*\*\*\*\*\*\*\*\*\*\*\*\*\*\*\*\*\*\*\*\*\*\*\*\*\*\*\*\*\*\*\*\*\*\*\*\*\*\*\*\*\*\*\*\*\*\*\*\*\*\*\*\*\*\*\*\*\*\*\*\*\*\*\*\*\*\*\*\*\*\*

---

## Author Response (AR2)

**This document presents the answers to the editor, plus the version of the manuscript with the changed paragraphs underlined in green following editor's comments**

Comments to the Author:

**MC1:** Thank you for your important revisions. The main three issues raised by both reviewers concerned the lack of strong hypothesis to introduce the paper, the lack of evidence that dust deposition did occur and important issues regarding the statistical analyses. In addition, reviewers also made many minor comments. Following the very detailed report of both referees, the authors have carefully answered point by point to major and minor issues raised. Consequently, authors have substantially rewritten some sections, new sections have been added, new figures and tables proposed and finally the references have been updated. The paper reads very well now also because a native English speaker proofread the manuscript.

I therefore consider that your paper can be now published after some minor revisions.

**MC2:** I made many edits and comments directly on your text

All this edits and comments were considered. We assigned a code to each and the answers can be found in the text comment section from TC1 to TC22.

*in addition, I recommend that you answer the following points:*

*Abstract:*

**MC3:**- I wonder if updates could be done considering that several important points and conclusions are now supported using different statistical tools: I find it a pity that the efforts made on the statistical aspects to consolidate the interpretations of the results are not at all reflected in the abstract and the conclusion. I strongly encourage the authors to do so.

We fully agree with this comment and the abstract is now completely rewritten.

The PEACETIME cruise (May-June 2017) was a basin scale survey covering the Provencal, Algerian, Tyrrhenian and Ionian basins during the post-spring bloom period and was dedicated to track the impact of Saharan dust deposition events on the Mediterranean Sea pelagic ecosystem. Two such

events occurred during this period, and the cruise strategy allowed to study the initial phase of the ecosystem response to one dust event in the Algerian basin (during 5 days at the so-called 'FAST long-duration station'), and a latter response to another dust event in the Tyrrhenian basin (by sampling from 5 to 12 days after the deposition). The present paper documents the structural and functioning patterns of the zooplankton component during this survey, including their responses to these two dust events. The mesozooplankon was sampled at 12 stations by combining nets with 2 mesh sizes (100 and 200 µm) mounted on a Bongo frame for vertical hauls within the 0-300 meter-depth layer.

Algerian and Tyrrhenian basins were found quite similar in terms of hydrological and biological variables, which clearly differentiated them from the northern Provencal Basin and the eastern Ionian Basin. In general, total mesozooplankton showed reduced variations in abundance and biomass values over the whole area, with a noticeable contribution of the small size fraction (< 500 µm) of up to 50 % in abundance and 25 % in biomass. This small-size fraction makes a significant contribution (15 to 21 %) to the mesozooplankton fluxes (carbon demand, grazing pressure, respiration and excretion) estimated using allometric relationships to the mesozooplankton size spectrum at all stations. The taxonomic structure was dominated by copepods, mainly cyclopoids and calanoids, and completed by appendicularians, ostracods and chaetognaths. Zooplankton taxa assemblages, analyzed using multivariate analysis and rank frequency diagrams, slightly differed between basins in agreement with recently proposed Mediterranean regional patterns.

However, the strongest changes in zooplankton community were linked to the dust deposition events. A synoptic analysis of the two dust events observed in the Tyrrhenian and Algerian basins and based on the rank frequency diagrams and a derived index proposed by Mouillot and Lepretre (2000) delivered a conceptual model of a virtual time series of zooplankton community responses after a dust deposition event. The initial phase before the deposition event (state 0) was dominated by small-size cells consumed by their typical zooplankton filter feeders (small copepods and appendicularians). Then, the disturbed phase during the first five days after the deposition event (state 1) induced a strong increase of filter-feeders and grazers of larger cells and the progressive attraction of carnivorous species, leading to a sharp increase of the zooplankton distribution index. Afterward, this index progressively decreased from day 5 to day 12 highlighting a diversification of the community (state 2). A three weeks delay was estimated to get the index returned to its initial value, potentially indicating the recovery time of a Mediterranean zooplankton community after a dust event.

To our knowledge, PEACETIME is the first in situ study allowing observation of mesozooplankton responses before and soon after natural Saharan dust depositions. The change in rank-frequency diagrams of the zooplankton taxonomic structure is an interesting tool to highlight short-term responses of zooplankton to episodic dust deposition events.

**MC4:**- I found the section starting L18 quite confusing as it somehow contradicts the observations analyzed as a 'virtual time series' under the influence of dust deposition with different time lag. I suggest removing that sentence and to focus on a clear statement about the 'virtual time series concept' results that I think are very interesting.

See comment MC3.

Results and Table 1.

**MC5:** I would be more precise concerning the sediment traps data and I made suggestions directly in the text. Please report in the text the corrections that I suggest for the lithogenic cumulated fluxes. The point here is that (1) at TYR the sediment trap sampling likely missed a part of the export as it started 6 days after the event occurred and (2) at FAST, the last trap represented 24 hours collection between June 5 and 6, so the collection likely missed the main lithogenic export that likely occurred after. Lithogenic fluxes at both sites are thus likely well below the actual export following the events.

Done in table 2 and in the text paragraph 2.2

**MC6:** There is one remark from Referee#2 concerning the sampling 0-300m: I agree with you answer but I think your final statement "Also note that the observed impact on zooplankton is more significant because it integrates the whole water column" should be somewhere in the text, conclusion maybe (need to rephrase the sentence).

Done in paragraph 4.4

**MC7:** There are a number of additional figures and table that are presented in response to reviewers comments. I would definitely recommend adding this pertinent material to the Supp. Info.

Done

**Text Comments (TC)**

Answers to your comments on the draft

**TC1:** I would remove 'major' as the one at FAST was rather modest

The following change was made in the sentence: "including their responses to two dust events."

**TC2:** I found this section quite confusing as it contradicts the observations analysed as a 'virtual time series' under the influence of dust deposition with different time lag. I suggest to remove that sentence and to focus on a clear statment about the 'virtual time series' results.

The abstract is now rewritten and that sentence was removed

**TC3:** I think it is too early to give such information

The following change was made in the sentence: "A dust event occurred over a large area"

**TC4:** be carefull order of station was wrong St 5, TYR then ST6

The following change was made in the sentence: "samples at ST5, TYR and ST6"

**TC5:** parenthese

The parenthesis was added

**TC6:** collected at TYR between 6 and 9 days after the event (only a part of the exported lithogenic material could thus have been collected) => the deposition that occured 11-12 May was thus higher than what was collected several days after.

The following sentence was added in the draft : " Lithogenic flux values at TYR and FAST are likely underestimated considering that traps were placed with a time delay after the dust event (6 and 1 days respectively), thus the reported values could represent only a fraction of the total fluxes."

**TC7:** replace by "Bressac et al., in prep"

The citation was changed

**TC8:** Paragraph 2.2 Line 104: "considered as a 'non recently impacted site' (please add something like that, otherwise 'for comparison' alone is not clear enough)"

Answer: "station ION will be considered (for comparison) as a non-recently impacted area."

**TC9-TC10:** those need to be defined also in the text

All the names of the basins are now defined in the text paragraph 3.1

**TC11:** please, add: Central and Western (as a rev asked for additional ref in the eastern Med that you didn't add for that reason.

The following change was made in the sentence: "in different regions of the Central and Western Mediterranean Sea"

**TC12:** maybe good place here to indicate that biomass in that table is an average over -250 m, likely diluting any possible effect on the surface mixed layer for ex.

The following change was made in paragraph 4.4 : Thus, the PEACETIME survey dedicated to the tracking of such events was an opportunity to observe real in situ zooplankton responses in the epipelagic layer (0-300 m).

**TC13:** see my remark about traps at FAST: only the last sample is likely affected by the dust deposition

In the following sentence below we use the difference in swimmers in traps between the two dates with the first one as a non impacted as reference:

"The daily observation of sediment traps at 200 and 500 meters over five days between FAST1 and FAST3 (pers.comm. C. Guieu) shows a relative increase of swimmers collected at 500 m versus those collected at 200 m, also suggesting increasing numbers of migrants.

Thus, there is no reason to modify this sentence.

**TC14:** Nagib did all the sampling

Change done

**TC15:** the new citation is:Van Wambeke F., Taillandier V., Desboeufs K., Pulido-Villena E., Dinasquet J., Engel A., Maranon E., Guieu C., Influence of atmospheric deposition on biogeochemical cycles in an oligotrophic ocean system, in preparation for Biogeosciences, (this special issue).

Change done

**Figure 1- TC16**:these should be also defined in the text

This is defined in the section 3.1 see comments TC9-TC10

**Table 2- TC17:** est ce que tu as bien pris les dernières données ds le draft de France et al.?

Answer: these data were taken from a  Excel file send by France. But the data is not presented in her paper, so we change the reference as personal communication.

**Table 2- TC18:** il y a des données de NO3 à TYR

There is data of NO3 at TYR but not for the cast close to the zooplankton sample. Data for NO3 was taken 2 day before the zooplanton sample. So we put this value and explain in the legend that it was taken two days before.

Table 2- **TC19:**Please report in the text the corrections that I made for the lithogenic cumulated fluxes. The point here is that (1) at TYR we likely missed a part of the export as we started sample 6 days after the event occurred and (2) at FAST, the last trap represented 24 hours collection between June 5 and 6, so we also likely missed the main lithogenic export.

This correction was made in the text paragraph 2.2

Table 2- **TC20:** FAST cumulated (5 days) lithogenic export was 50 mg.m-2 (at 200 m) and 70 mg.m-2 (at 1000 m) (Bressac et al., in prep.).

The sediment trap was place from June 5 to June 6 so it should be a 24 hrs collection as you said before. **We understood that the 5 cumulated days you say in this point probably is a mistake. Are we right?** **Please answer to be changed in the text accordingly in Line 106**

Table 2- **TC21:** Quantifying the dust deposition from sediment traps data is difficult as sediment traps were recovered on June 6 whilst the dust deposition occured on June 5. So this cumulated flux is a minimum value for the dust deposition in the area.

This correction was made in the text paragraph 2.2

Table 2- **TC22:** TYR cumulated (4 days) lithogenic export was 153 mg.m-2 (at 200 m) and 207 mg.m-2 (at 1000 m) (Bressac et al., in prep.). Note that this is a minimum dust deposiition since the sampling started 6 days after the event

This correction was made in table 2 and in the text paragraph 2.2

[revised manuscript text omitted]